# OTX2 controls chromatin accessibility to direct somatic versus germline differentiation

Elisa Barbieri [1,2] & Ian Chambers [1,2]

## Abstract

The choice between somatic and germline fates is essential for species survival. This choice occurs in embryonic epiblast cells, as these cells are competent for both somatic and germline differentiation. The transcription factor OTX2 regulates this process, as *Otx2*-null epiblast-like cells (EpiLCs) form primordial germ cell-like cells (PGCLCs) with enhanced efficiency. Yet, how OTX2 achieves this function is not fully characterised. Here we show that OTX2 controls chromatin accessibility at specific chromatin loci to enable somatic differentiation. CUT&RUN for OTX2 and ATAC-seq in wild-type and *Otx2*-null embryonic stem cells and EpiLCs identifies regions where OTX2 binds and opens chromatin. Enforced OTX2 expression maintains accessibility at these regions and also induces opening of ~4000 somatic-associated regions in cells differentiating in the presence of PGC-inducing cytokines. Once cells have acquired germline identity, these additional regions no longer respond to OTX2 and remain closed. Our results indicate that OTX2 works in cells with dual competence for somatic and germline differentiation to increase accessibility of somatic regulatory regions and induce the somatic fate at the expense of the germline.

**Keywords** Formative Pluripotency; OTX2; Germline; Chromatin Accessibility
**Subject Categories** Chromatin, Transcription & Genomics; Development

## Introduction

During embryonic development, cells are faced with choices that determine their fate on multiple occasions. A critical step is the choice between somatic and germline differentiation which occurs early in mammalian development shortly after implantation. In mouse embryos at day 6.5, cells in the embryo proper express OTX2, a transcription factor associated with epiblast identity both in vivo and in vitro and with a critical function in neural development (Acampora et al, 1995, 2013; Simeone et al, 1992; Simeone, 1998; Iwafuchi-Doi et al, 2012). Cells in the posterior proximal region of the epiblast respond to external signals, in particular bone morphogenic factor (BMP) 4, secreted from the

adjacent extraembryonic tissue (Hayashi et al, 2007; Lawson et al, 1999). At this time, while the rest of the epiblast continues to express OTX2, this group of cells downregulate OTX2. Subsequently, these cells go on to express primordial germ cell (PGC)-associated transcription factors BLIMP1, PRDM14 and AP2γ (Lawson et al, 1999; Zhang et al, 2018a; Ohinata et al, 2009; Vincent et al, 2005; Yamaji et al, 2008; Weber et al, 2010; Kurimoto et al, 2008; Ohinata et al, 2005). This process can be recapitulated in vitro. Naive embryonic stem cells (ESCs) can be differentiated into PGC-like cells (PGCLCs) via a transient population of formative pluripotent cells called epiblast-like cells (EpiLCs). EpiLCs are able to respond to BMP4 and are therefore considered competent for germline development (Hayashi et al, 2011; Hayashi and Saitou, 2013). These cells downregulate OTX2 and subsequently induce expression of PGC-associated TFs (Zhang et al, 2018a; Hayashi et al, 2011).

The differential expression of OTX2, which is high in the epiblast but becomes repressed as cells enter the germline, is suggestive of a role for OTX2 in the choice between somatic and germline differentiation. Indeed, in vivo, embryos lacking *Otx2* show an increase in PGC numbers at embryonic day 7.5 (Zhang et al, 2018a). In addition, *Otx2*-null ESCs can generate PGCLCs from EpiLCs with higher efficiency than wild-type cells (Zhang et al, 2018a).

Cells in the posterior epiblast, as well as EpiLCs in vitro, possess a dual competence for both somatic and germline differentiation. The choice of which fate to follow is based on several factors, among which the reactivation of the naive gene regulatory network plays a pivotal role. OTX2 has been associated with the ability to repress the naive gene regulatory network. Indeed, OTX2 antagonises the activity of NANOG in ESCs (Acampora et al, 2017) and OTX2 binding sequences have been identified in the regulatory regions of *Nanog*, *Oct4* and *Sox2* (Acampora et al, 2016). Deletion of OTX2 binding elements in the *Nanog* and *Oct4* regulatory regions leads to the increased expression of *Nanog* and *Oct4* in EpiLCs and an increased yield of PGCLCs (Di Giovannantonio et al, 2021).

OTX2 not only represses naive pluripotency. OTX2 overexpression in ESCs induces the exit from naive pluripotency and the expression of formative and primed pluripotency-associated genes (Acampora et al, 2013). This suggests that OTX2 can actively instruct the cells towards a more differentiated cell state. Moreover, the increase in OTX2 expression during the transition of ESCs to EpiLCs leads to the redistribution of OCT4 on chromatin,

[1]Centre for Regenerative Medicine, Institute for Regeneration and Repair, 5 Little France Drive, Edinburgh EH16 4UU, Scotland. [2]Institute for Stem Cell Research, School of Biological Sciences, University of Edinburgh, 5 Little France Drive, Edinburgh EH16 4UU, Scotland. ✉E-mail: Elisa.Barbieri@ed.ac.uk

contributing to the establishment of the formative and primed gene regulatory network (Buecker et al, 2014; Yang et al, 2014). Subsequently, *Otx2* is rapidly downregulated as EpiLCs transition to PGCLCs, but is maintained initially during somatic differentiation (Zhang et al, 2018a). Together, these observations suggest that OTX2 has a more prominent role in determining the somatic fate of cells with competence for both germline and somatic differentiation.

In this work, we show that OTX2 is able to open chromatin at specific somatic regulatory regions, for example, at the *Fgf5* enhancers, priming the cells towards the somatic fate. OTX2 induces chromatin accessibility early during the EpiLC to PGCLC transition, when cells possess dual competence, instructing cells towards the somatic fate at the expense of the germline.

# Results

## OTX2 localisation at chromatin

To investigate how OTX2 acts on chromatin to determine differentiation choice, the chromatin binding profile of OTX2 was analysed in cells competent for germline differentiation. Wild-type ESCs were differentiated into EpiLCs for 44 h and Cleavage Under Targets & Release Using Nuclease (CUT&RUN (Skene et al, 2018)) was performed in both ESCs and EpiLCs (Fig. 1A). A total of 7136 OTX2-bound regions were identified (Fig. 1B). The majority of OTX2-bound regions (4443) are EpiLC-specific but 1429 regions are ESC-specific and 1264 regions are bound by OTX2 in both ESCs and EpiLCs. Heatmaps show the distribution of OTX2 binding to these 7136 regions (Fig. 1C). While ESC-specific and EpiLC-specific regions had higher OTX2 occupancy in the respective cell type (Fig. 1C,D), common regions showed higher occupancy by OTX2 in EpiLCs (Fig. 1C), in line with the higher OTX2 protein expression in EpiLCs (Buecker et al, 2014). This was borne out in the analysis of the average signal of OTX2 binding to these regions which is also slightly higher in EpiLCs (Fig. 1D). Indeed, common regions showed the highest average signal of all sites in both ESCs and EpiLCs, suggesting that although already present in ESCs, OTX2 binds strongly or more frequently to these regions in EpiLCs compared to ESCs (Fig. 1D). Examples of ESC-specific, common and EpiLC-specific OTX2 peaks are enhancers of *Tet2, Mycn* and *Fgf5*, respectively (Whyte et al, 2013; Buecker et al, 2014; Thomas et al, 2021). *Tet2* intragenic enhancer shows strong binding of OTX2 in wild-type ESCs but not in EpiLCs. OTX2 binds *Mycn* enhancer in both cell types with higher intensity in EpiLCs compared to ESCs. In contrast, the *Fgf5* downstream enhancers E1, E2 and E3 are bound by OTX2 only in EpiLCs (Figs. 1E and EV1A).

Most OTX2-bound regions are located distal to the transcription start sites in both ESCs and EpiLCs (Fig. EV1B). Although the binding of OTX2 to promoters increases in common and EpiLC-specific regions compared to ESC-specific regions, the majority of OTX2-bound regions remain distally located (Fig. EV1B). Comparison of these regions with published ESC and EpiLC enhancer lists showed that 49.6% of OTX2-bound common distal and 53.4% of OTX2-bound ESC-specific regions overlap with ESC enhancers previously defined by co-localisation of OCT4, SOX2 and NANOG (Whyte et al, 2013; Chen et al, 2008). In addition, 77.6% of OTX2-bound common distal regions and 52.1% OTX2-bound EpiLC-

specific distal regions overlap with EpiLC enhancers previously defined by the presence of H3K27Ac, H3K4me1 and p300 (Buecker et al, 2014; Sankar et al, 2022). Moreover, ESC-specific and EpiLC-specific distal regions show high levels of H3K4me1 (a marker of enhancer regions), in ESCs and EpiLCs, respectively (data from (Bleckwehl et al, 2021)), with common regions having similar levels of H3K4me1 in the two cell types (Fig. EV1C). While most ESC-specific distal regions also have an H3K27ac signal in ESCs (a marker of active regions), only a subset of EpiLC-specific regions have an H3K27ac signal in EpiLCs (Fig. EV1C). This suggests that OTX2 binds mostly to active putative enhancers in ESCs and to active and primed putative enhancers in EpiLCs.

To investigate differences between the OTX2-bound sites, motif analysis on ESC-specific, common and EpiLC-specific OTX2 peaks was performed. As expected, the OTX2 motif was enriched in all subsets (Fig. EV1D). Together with OTX2-like motifs recognised by GSC and CRX, these were among the most enriched in all three subsets. Consistent with the known interaction of OCT4 with OTX2 (Buecker et al, 2014), the OCT/SOX motif was also enriched in all three datasets (Fig. EV1D). Differences between the subsets emerge when analysing other enriched motifs. ESC-specific and common regions are enriched for SOX and KLF family motifs and EpiLC-specific regions are enriched for ZIC family motifs (Fig. EV1D). Gene ontology analysis of the genes closest to OTX2-bound regions, and that are therefore likely to be targets of OTX2, reveals that both ESC-specific and EpiLC-specific regions associate with genes involved in transcriptional regulation. ESC-specific regions are also closely located to genes involved in stem cell maintenance and development, while EpiLC-specific regions are associated with genes involved in differentiation (axon guidance and neural system development) (Fig. EV1E). Indeed, the largest change in probability is the increase in association of the term 'multicellular organism development' in EpiLC-specific regions. Taken together, these results suggest that OTX2 may have different roles in naive and formative pluripotency, acting near SOX motifs in ESCs at genes regulating stem cell maintenance and potentially with ZIC proteins at genes that prepare for differentiation in EpiLCs.

## Lack of OTX2 leads to altered chromatin accessibility

Chromatin is remodelled during differentiation (Chen and Dent, 2014; Hota and Bruneau, 2016). This enables cells to acquire different cell identities by opening new regulatory regions and by closing regions that regulate states that cells have transitioned beyond. As OTX2 has an essential role in orchestrating the choice between somatic and germline fates, we asked whether OTX2 functions to modulate chromatin accessibility. The assay for transposase-accessible chromatin with sequencing (ATAC-seq (Buenrostro et al, 2013)) was used in wild-type and *Otx2*$^{-/-}$ ESCs, EpiLCs, and early during differentiation (summarised in Fig. 2A). To clearly distinguish between chromatin changes associated with early PGCLC differentiation and those associated with somatic cell differentiation, we compared cells at day 2 of differentiation. PGCLCs can be distinguished from somatic cells by cell surface expression of CD61 and SSEA1 (Hayashi et al, 2011; Hayashi and Saitou, 2013). However, as CD61 and SSEA1 only become detectable at day 4 of PGCLC differentiation, we were unable to sort PGCLCs from somatic cells at day 2. To circumvent this

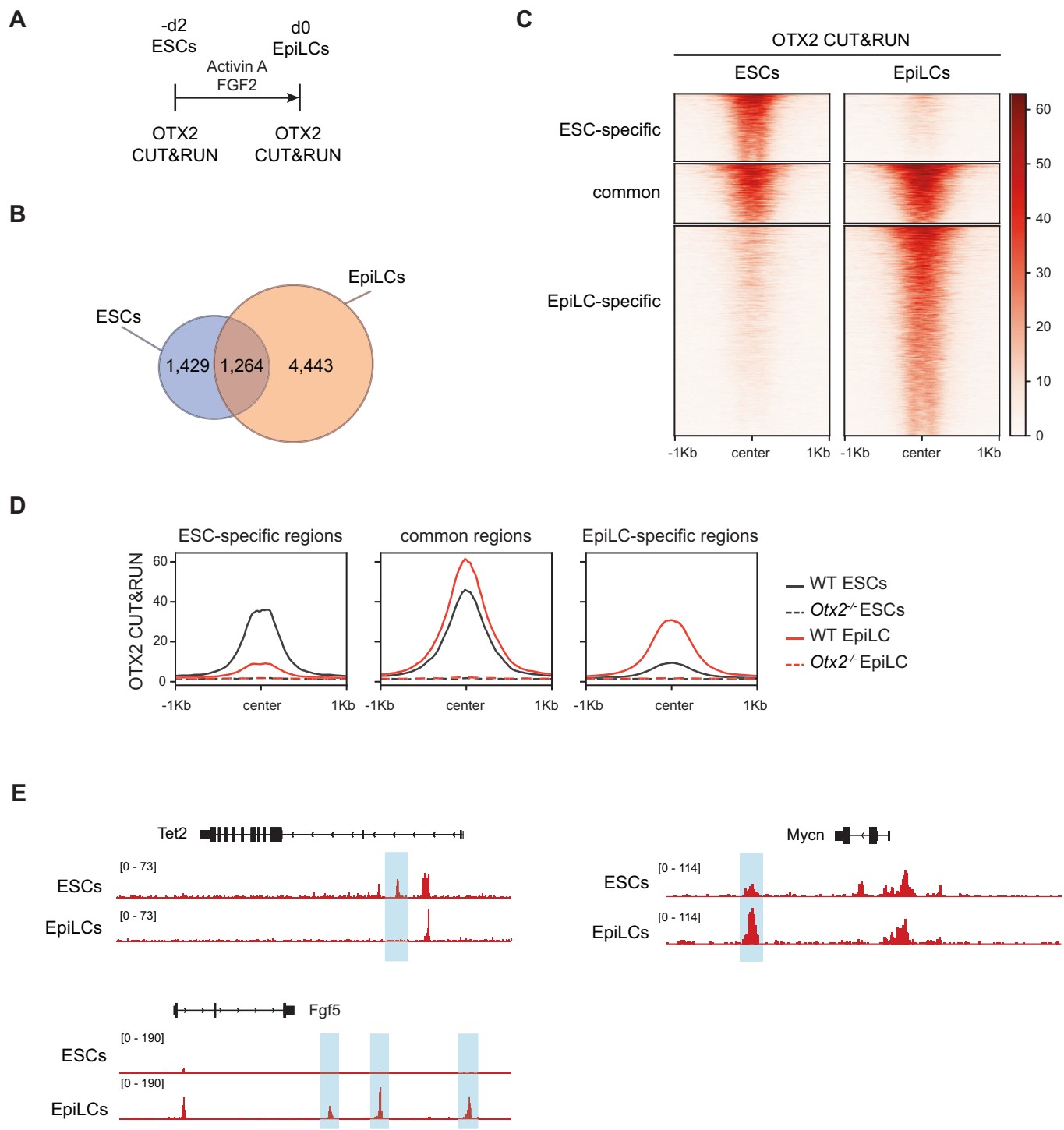

**Figure 1. OTX2 chromatin binding during the ESC to EpiLC transition.**

(**A**) Summary of OTX2 CUT&RUN samples analysed in this study. (**B**) Venn diagram of OTX2-bound regions in ESCs (blue) and EpiLCs (orange). (**C**) Heatmap of OTX2 CUT&RUN signal, showing ESC-specific, EpiLC-specific and common regions. (**D**) Average read density profiles of OTX2 CUT&RUN in wild-type and *Otx2*$^{-/-}$ ESCs, wild-type and *Otx2*$^{-/-}$ EpiLCs at ESC-specific, common and EpiLC-specific OTX2-bound regions. (**E**) OTX2 CUT&RUN tracks showing examples of ESC-specific (*Tet2* (Whyte et al, 2013)), common (*Mycn* (Whyte et al, 2013)) and EpiLC-specific (*Fgf5* (Buecker et al, 2014; Thomas et al, 2021)) OTX2-bound regions. Nomenclature of *Fgf5* enhancers from (Buecker et al, 2014).

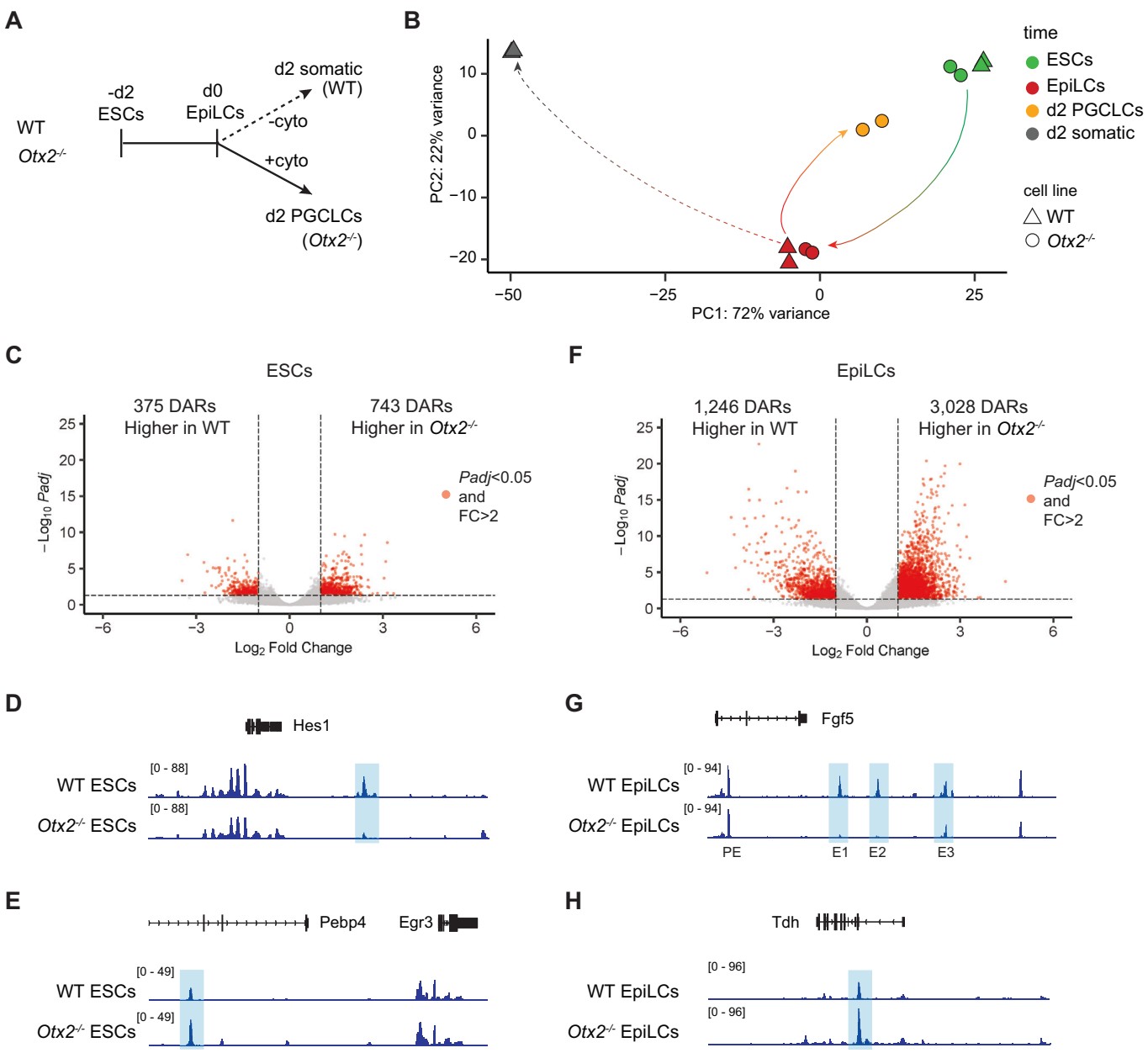

**Figure 2. Chromatin accessibility during ESC - EpiLC - PGCLC/somatic transitions.**

(A) Summary of ATAC-seq samples analysed in this study. To obtain a comparison of cell populations that were almost entirely PGCLCs, or from which PGCLCs were entirely lacking, we compared wild-type (WT) cells differentiated without cytokines with *Otx2*$^{-/-}$ cells differentiated with cytokines. At day 6 of differentiation, these populations either completely lack CD61 + SSEA1+ PGCLCs or are composed of >90% CD61 + SSEA1+ cells, respectively, as illustrated by flow cytometry plots in Fig. EV2. (B) Principal component analysis of ATAC-seq samples. Arrows show ESC → EpiLC, EpiLC → PGCLC and EpiLC → somatic cell transitions. (C) Volcano plot comparing accessible regions in wild-type (WT) and *Otx2*$^{-/-}$ ESCs. Analysis performed and plot generated by DESeq2 using Wald test with Benjamini–Hochberg correction for multiple testing from $n = 2$ biological replicates per condition. (D, E) ATAC-seq tracks showing chromatin accessibility in wild-type (WT) and *Otx2*$^{-/-}$ ESCs for *Hes1* (D) and *Pebp4* (E). (F) Volcano plot comparing accessible regions in wild-type (WT) and *Otx2*$^{-/-}$ EpiLCs. Analysis performed and plot generated by DESeq2 using Wald test with Benjamini–Hochberg correction for multiple testing from $n = 2$ biological replicates per condition. (G, H) ATAC-seq tracks showing chromatin accessibility in wild-type (WT) and *Otx2*$^{-/-}$ EpiLCs for *Fgf5* (G) and *Tdh* (H). *Fgf5* enhancer nomenclature from (Buecker et al, 2014). All DARs are highlighted (blue).

problem, we compared *Otx2*$^{-/-}$ cells cultured in the presence of PGC-promoting cytokines with wild-type cells cultured in the absence of PGC-promoting cytokines. Under these conditions *Otx2*$^{-/-}$ cells produce an essentially pure ( > 90%) CD61$^+$/SSEA1$^+$ population (Zhang et al, 2018a; Hayashi et al, 2011; Hayashi and

Saitou, 2013), while wild-type cells yield a cell population from which PGCLCs are absent (Figs. 2A and EV2). Therefore, we used *Otx2*$^{-/-}$ cells cultured in the presence of PGC-promoting cytokines to model PGCLCs and wild-type cells cultured in the absence of PGC-promoting cytokines to model somatic cells.

**A**

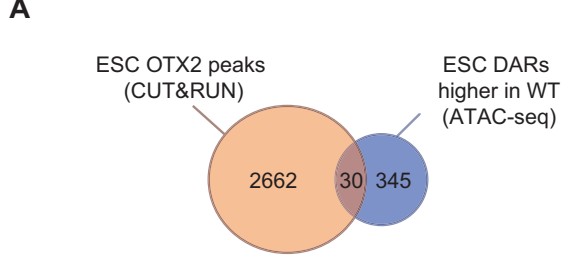

ESC OTX2 peaks
(CUT&RUN)

ESC DARs
higher in WT
(ATAC-seq)

2662   30   345

**B**

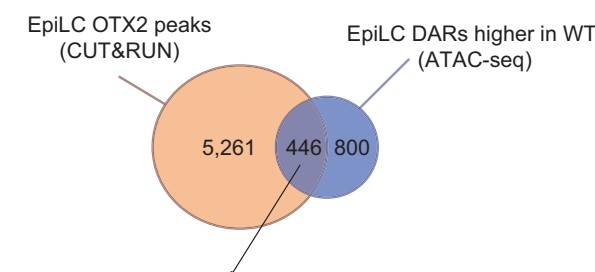

EpiLC OTX2 peaks
(CUT&RUN)

EpiLC DARs higher in WT
(ATAC-seq)

5,261   446   800

EpiLC DARs higher in WT with OTX2 binding

**C**

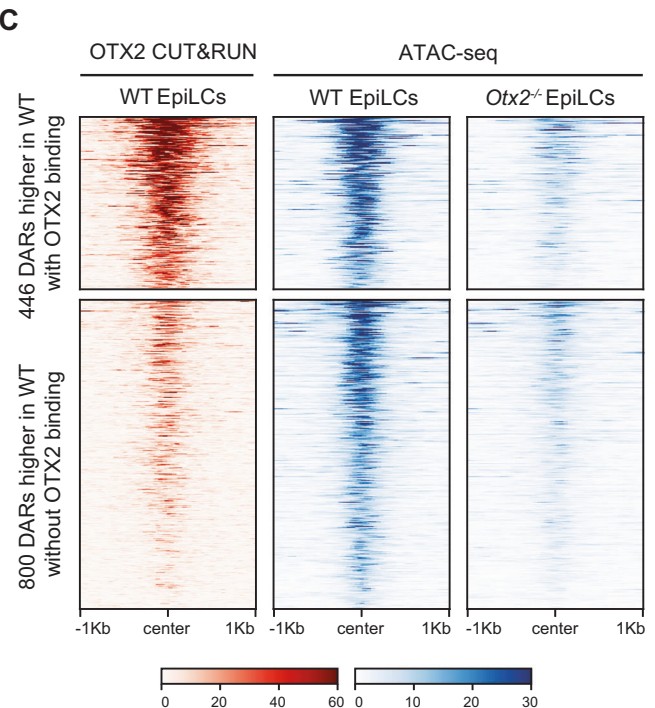

OTX2 CUT&RUN     ATAC-seq

WT EpiLCs     WT EpiLCs     *Otx2⁻/⁻* EpiLCs

446 DARs higher in WT with OTX2 binding

800 DARs higher in WT without OTX2 binding

-1Kb  center  1Kb   -1Kb  center  1Kb   -1Kb  center  1Kb

0  20  40  60   0  10  20  30

**D**

446 DARs higher in WT with OTX2 binding

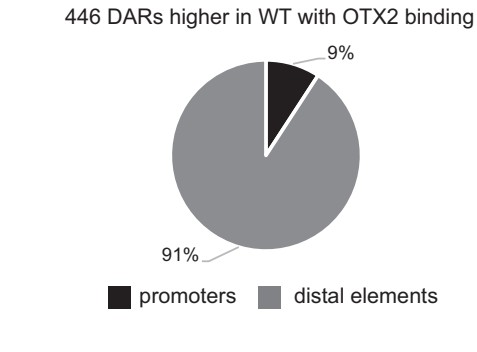

9%

91%

■ promoters  ■ distal elements

**E**

446 DARs higher in WT with OTX2 binding

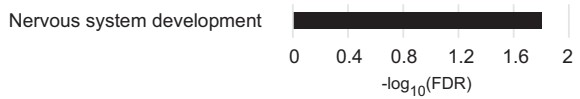

ESCs          EpiLCs

ATAC-seq

-1Kb  center  1Kb   -1Kb  center  1Kb

— WT   — *Otx2⁻/⁻*

**F**

HOMER motif analysis
446 DARs higher in WT with OTX2 binding

*p-value*

Otx2     1e-140
GSC      1e-125
CRX      1e-77
Pitx1    1e-27
Zic3     1e-24

HOMER motif analysis
800 DARs higher in WT without OTX2 binding

*p-value*

GSC      1e-67
Otx2     1e-65
Zic3     1e-63
Zic      1e-40
CRX      1e-39

**G**

Gene ontology of 446 DARs higher in WT
with OTX2 binding closest genes

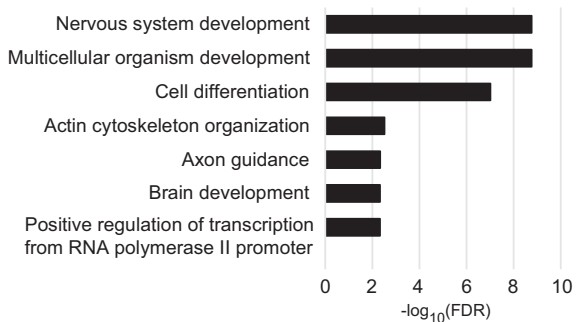

Nervous system development

0    0.4   0.8   1.2   1.6    2
-log₁₀(FDR)

**H**

Gene ontology of 1,246 EpiLC DARs higher in WT closest genes

Nervous system development
Multicellular organism development
Cell differentiation
Actin cytoskeleton organization
Axon guidance
Brain development
Positive regulation of transcription
from RNA polymerase II promoter

0    2    4    6    8    10
-log₁₀(FDR)

◄

**Figure 3.   OTX2 facilitates chromatin accessibility in EpiLCs.**

(A) Venn diagram of the overlap of OTX2-bound regions in ESCs (orange) and ESC DARs that are more accessible in wild-type ESCs (blue). (B) Venn diagram of the overlap of OTX2-bound regions in EpiLCs (orange) and EpiLC regions where accessibility is higher in the wild-type compared to $Otx2^{-/-}$ cells (blue) identifying two subsets of DARs, OTX2-bound (446 regions) and OTX2-unbound (800 regions). (C) Heatmap of OTX2 binding (CUT&RUN - red) and accessibility (ATAC-seq - blue) at 446 OTX2-bound regions and 800 OTX2-unbound regions that show increased accessibility in wild-type compared to $Otx2^{-/-}$ EpiLCs. (D) Genomic distribution of the 446 OTX2-bound regions. Promoters are $+/-$ 1 kb from any TSS. (E) Average read density profiles of ATAC-seq signal at the 446 OTX2-bound regions in wild-type and $Otx2^{-/-}$ ESCs and EpiLCs. (F) Motif analysis in 446 OTX2-bound and 800 OTX2-unbound EpiLC regions showing high enrichment of OTX2-like motifs in the 446 OTX2-bound regions and enrichment of motifs recognised by both OTX2 and ZIC family members in the 800 OTX2-unbound regions. (G) Gene ontology analysis of the closest genes to the 446 OTX2-bound DARs showing higher accessibility in wild-type EpiLCs. (H) Gene ontology analysis of the closest genes to the 1246 EpiLC regions more accessible in the wild-type.

ATAC-seq data from the above differentiated samples was compared to ESCs and EpiLCs (Fig. 2B). Principal component analysis (PCA) showed that loss of OTX2 does not dramatically alter the global chromatin accessibility in ESCs or EpiLCs, as wild-type and $Otx2^{-/-}$ cells cluster together at both stages. In contrast, after 2 days of differentiation, PGCLCs and somatic cells have drastically distinct chromatin accessibility landscapes (Fig. 2B). Interestingly, d2 PGCLCs show an intermediate position between ESCs and EpiLCs on both PC1 and PC2, suggesting that during differentiation into the germline, cells may restore some chromatin characteristics that previously defined ESCs. In contrast, wild-type differentiated cells (somatic cells) cluster far from other samples, suggesting that cells that adopt a somatic fate have drastic changes in chromatin accessibility compared to pluripotent cells and to PGCLCs (Fig. 2B).

Although the PCA suggests a high similarity between wild-type and $Otx2^{-/-}$ cells at both ESC and EpiLC stages, direct comparison shows that subsets of regions are differentially accessible. Comparing wild-type and $Otx2^{-/-}$ ESCs identified 375 differentially accessible regions (DARs) with increased accessibility in wild-type cells, and 743 regions with higher accessibility in $Otx2^{-/-}$ ESCs (Fig. 2C). Examples of DARs with increased accessibility in wild-type or $Otx2^{-/-}$ ESCs are present in the *Hes1* and *Pebp4* loci, respectively (Fig. 2D,E).

Compared to ESCs, a higher number of DARs were detected from the analysis of EpiLCs. Specifically, 1246 regions were more accessible in wild-type EpiLCs and 3028 regions were more accessible in $Otx2^{-/-}$ EpiLCs (Fig. 2F). An example of a gene with DARs that become accessible in the ESC to EpiLC transition is *Fgf5*. *Fgf5* DARs correspond to characterised EpiLC-specific enhancers (PE, E1, E2 and E3) (Buecker et al, 2014) and they remain comparatively inaccessible in $Otx2^{-/-}$ EpiLCs (Fig. 2G). An example of DAR that has increased accessibility in EpiLCs in the absence of OTX2 is an open region at the *Tdh* locus (Fig. 2H). This suggests that in the absence of OTX2, chromatin accessibility changes in both ESCs and EpiLCs.

## Cells lacking OTX2 show loss of accessibility in a subset of EpiLC regions

To investigate the hypothesis that changes in accessibility between wild-type and $Otx2^{-/-}$ cells are directly induced by OTX2, we analysed the overlap between OTX2 binding sites and regions that are more accessible in wild-type cells. In ESCs, OTX2 binds <10% (30 out of 375) of DARs that are more accessible in wild-type cells than in $Otx2^{-/-}$ cells (Fig. 3A), suggesting that accessibility of ESC

DARs is directly due to OTX2 in a small subset of DARs (*P* value of OTX2 motif enrichment by HOMER = 1e-16). In contrast, in EpiLCs, OTX2 binds 36% (446 out of 1246) of the DARs that are more accessible in wild-type than in $Otx2^{-/-}$ cells (Fig. 3B,C). Notably, these regions are mainly located distal to genes (91%, Fig. 3D), despite the increased fraction of promoter regions bound by OTX2 in EpiLCs (Fig. EV1B). These 446 regions are more accessible in EpiLCs than ESCs, underscoring their EpiLC-specificity (Fig. 3E). Without OTX2 these regions show the same low level of accessibility in both ESCs and EpiLCs (Fig. 3E). Consistent with this, motif analysis of these 446 regions shows high enrichment for OTX-like motifs (Fig. 3F).

The remaining 800 DARs that are more accessible in wild-type EpiLCs than in $Otx2^{-/-}$ EpiLCs have an OTX2 CUT&RUN signal that falls below the threshold applied in the bioinformatic analysis. However, CUT&RUN for OTX2 shows that these 800 DARs have low but detectable OTX2 binding (Fig. 3C). Consistent with this, motif analysis of these 800 DARs identified OTX2 motifs, but with a reduced *P* value compared to the 446 DARs mentioned above (Fig. 3F). Together with the increased *P* value of ZIC motifs at these 800 DARs (Fig. 3F), this may indicate that these sites are bound by OTX2 at reduced affinity and that accessibility of these sites may require the combined action of OTX2 and ZIC TFs. Gene ontology analysis of the genes closest to the 446 DARs increased by OTX2 binding reveals an association with nervous system development (Fig. 3G). Expanding the analysis to the genes closest to the 1246 DARs higher in the wild-type reveals that these regions are associated with terms of differentiation, including more mature neural features (Fig. 3H). These results suggest that OTX2 may contribute to control chromatin accessibility at somatic regulatory regions.

## OTX2 indirectly controls closure of naive-associated chromatin regions

The comparison between accessible regions in wild-type and $Otx2^{-/-}$ EpiLCs revealed 3028 DARs with higher accessibility in $Otx2^{-/-}$ EpiLCs. OTX2 binding at these regions is low or undetectable (Fig. EV3A) with only 28 EpiLC DARs overlapping an OTX2 peak (Fig. EV3B). Therefore, OTX2 may not have a direct role in inducing chromatin closure. Interestingly, when we analysed the accessibility level of these 3028 DARs in ESCs, we found that they are even more accessible in ESCs than in EpiLCs (Fig. EV3C). Together, these results suggest that these 3028 regions are accessible in ESCs, but that they close during the transition from ESCs to EpiLCs, due to the indirect action of OTX2.

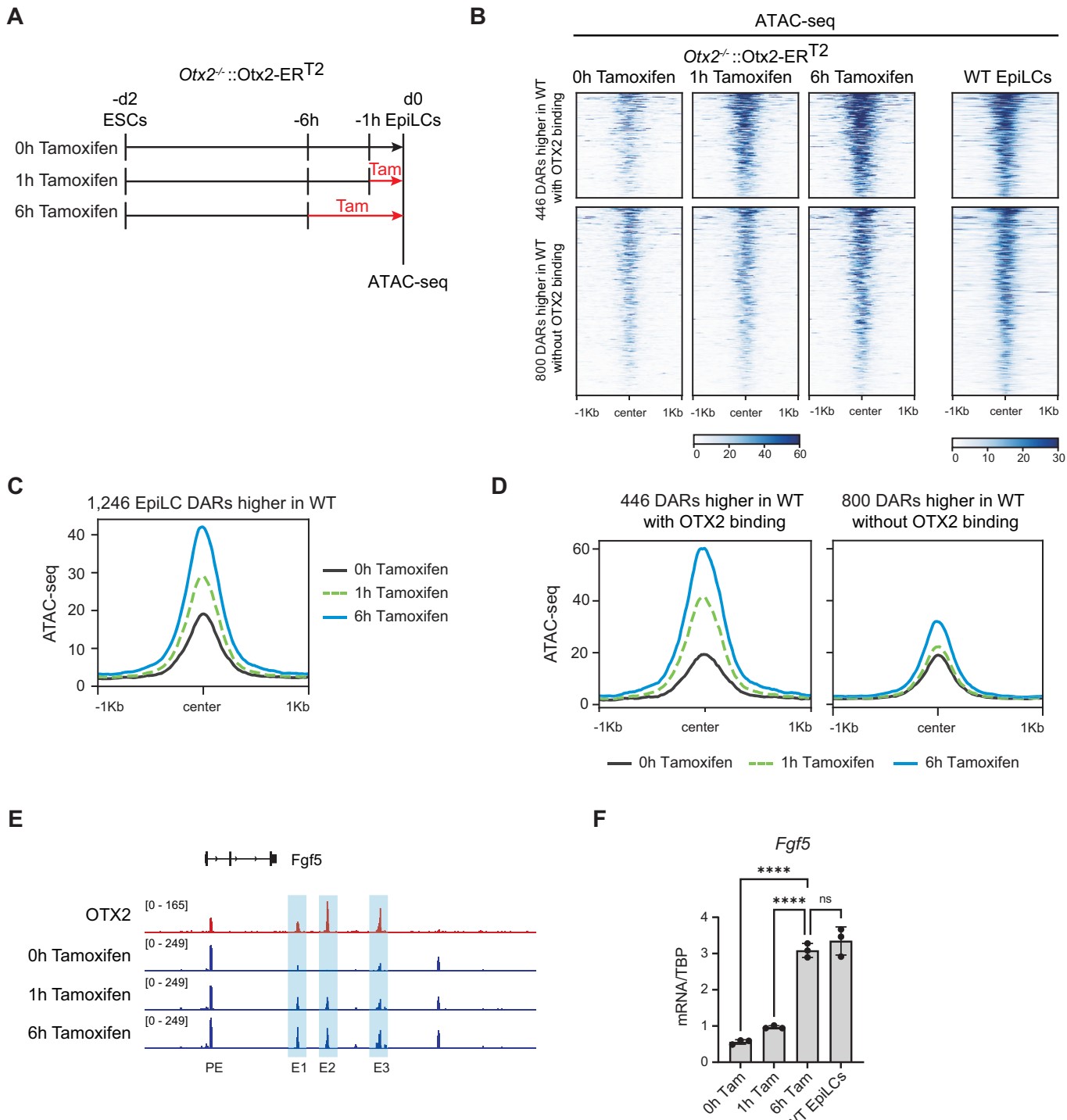

## OTX2 expression in Otx2−/− EpiLCs rescues chromatin accessibility

To determine whether OTX2 can directly alter the accessibility of the 1246 regions that are more accessible in wild-type EpiLCs than in Otx2−/− EpiLCs (Fig. 3B), we used an Otx2−/− cell line carrying an OTX2-ERT2 fusion protein to rapidly induce OTX2. In these cells, OTX2-ERT2 is relocated from the cytoplasm to the nucleus within

20 min of tamoxifen addition (Zhang et al, 2018a). To assess the effect of OTX2 on chromatin accessibility in EpiLCs, Otx2−/−::Otx2-ERT2 ESCs were differentiated to EpiLCs and tamoxifen was added either 1 or 6 h before the end of the differentiation (Fig. 4A). ATAC-seq showed that a 1-h treatment with tamoxifen is sufficient to induce an increase in the accessibility signal at these 1246 regions (Fig. 4B,C). The change in accessibility at 1 h is almost exclusively due to the 446 regions that are bound by OTX2 strongly, as the remaining 800 regions

◀ **Figure 4.  OTX2 expression rescues chromatin accessibility in EpiLCs.**

(A) Differentiation scheme for $Otx2^{-/-}$::Otx2-ER$^{T2}$ ESCs into EpiLCs +/− tamoxifen. (B) Heatmap of ATAC-seq signal at the 446 OTX2-bound (top) and 800 OTX2-unbound (bottom) EpiLC DARs in $Otx2^{-/-}$::Otx2-ER$^{T2}$ EpiLCs treated for 1 or 6 h with tamoxifen (left) and in wild-type EpiLCs (right). (C) Average read density profile of ATAC-seq signal at the 1246 DARs higher in wild-type than in $Otx2^{-/-}$::Otx2-ER$^{T2}$ EpiLCs, treated for 1 h (dashed green) or 6 h (blue) with tamoxifen compared to untreated cells (black). (D) Average read density profile of ATAC-seq signal at the 446 OTX2-bound and the 800 OTX2-unbound EpiLC DARs in $Otx2^{-/-}$::Otx2-ER$^{T2}$ EpiLCs treated for 1 h (dashed green) or 6 h (blue) with tamoxifen compared to untreated cells (black). (E) OTX2 CUT&RUN (red) and ATAC-seq (blue) tracks showing accessibility changes in response to OTX2 expression at the Fgf5 locus. DARs are highlighted in light blue. Fgf5 enhancer nomenclature from (Buecker et al, 2014). (F) Fgf5 mRNA levels in wild-type (WT) and $Otx2^{-/-}$::Otx2-ER$^{T2}$ EpiLCs treated with tamoxifen as indicated. mRNA levels were quantified by RT-qPCR and normalised to TBP mRNA levels. Data are from a representative of two independent experiments (centre: mean, data points: technical replicates, error bars: standard deviation, ns (not significant) = P ≥ 0.05, ****P < 0.0001). Statistical analysis (one-way ANOVA with Tukey's correction for multiple comparisons) was performed using the software GraphPad Prism version 10.5.0. Exact P values are: 0 h Tam vs 6 h Tam P < 0.0001; 1 h Tam vs 6 h Tam P < 0.0001; 6 h Tam vs WT EpiLCs P = 0.4976. Source data are available online for this figure.

showed little change in accessibility by 1 h (Fig. 4B,D). Although accessibility did increase further after 6 h of tamoxifen treatment (Fig. 4B,D), the fact that the major increase in accessibility occurred within 1 h at the 446 regions, together with the higher enrichment of OTX2 motifs at these regions (Fig. 3F) strongly suggests that OTX2 is directly and specifically required to open these chromatin regions.

Among these OTX2-controlled regions are regulatory elements of the OTX2 target gene Fgf5. The accessibility of these regions increases after relocation of OTX2-ER$^{T2}$, in the case of peak E2 from a baseline level (Fig. 4E). The increase in accessibility is accompanied by an increase in the level of Fgf5 mRNA expression (Fig. 4F). This suggests that OTX2 facilitates Fgf5 transcription not only by binding to enhancers but also by controlling the accessibility of these enhancers to the transcriptional machinery (Fig. 4E,F).

The majority of OTX2-bound sites in EpiLCs did not show any change in chromatin accessibility in $Otx2^{-/-}$ compared to wild-type EpiLCs (Figs. 3B and EV3D). To assess whether the accessibility of these sites is affected by OTX2, we analysed the ATAC-seq signal at these regions in tamoxifen-treated $Otx2^{-/-}$::OTX2-ER$^{T2}$ EpiLCs. Accessibility at the 5261 regions was not increased by tamoxifen treatment (Fig. EV4A,B). Nevertheless, motif analysis of the 5261 regions showed high enrichment for OTX2-like motifs, as previously seen for the EpiLC DARs where accessibility was induced by OTX2 binding (Fig. 3F). Interestingly, the OCT-SOX motif is only enriched in the 5261 regions (Fig. EV4C). Therefore, accessibility of these regions may rely on OCT4 and SOX2 rather than on OTX2.

Together, these results suggest that OTX2 controls accessibility of a subset of chromatin sites in cells competent for both somatic and germline differentiation.

## OTX2 retains the ability to open chromatin during the early stages of germline differentiation

In wild-type cells, OTX2 restricts the number of cells entering the germline (Zhang et al, 2018a). OTX2 can completely block entry of all cells into the germline when expression is enforced during the first 2 days of PGCLC differentiation (Zhang et al, 2018a).

To determine whether OTX2 retains the capacity to induce chromatin accessibility during PGCLC differentiation, $Otx2^{-/-}$::Otx2-ER$^{T2}$ EpiLCs were differentiated in the presence of PGC-inducing cytokines, either with or without tamoxifen. ATAC-seq and OTX2 CUT&RUN were performed at day 2 of differentiation (Fig. 5A). Enforced OTX2 expression resulted in an increased accessibility at the 1246 regions that were previously shown to be more accessible in wild-type EpiLCs than

in $Otx2^{-/-}$ EpiLCs (Fig. 5B). These regions showed low accessibility in the absence of tamoxifen but became open and bound by OTX2-ER$^{T2}$ in tamoxifen-treated cells (Fig. 5B,C). This suggests that OTX2 can increase accessibility of chromatin regulatory regions not only in EpiLCs but also during the early stages of differentiation in the presence of PGC-inducing cytokines.

To determine whether OTX2 expression is essential to maintain chromatin accessibility in cells differentiating in the presence of PGC-inducing cytokines after day 2, ATAC-seq was performed at day 4 of differentiation in PGCLC medium, in cells treated with tamoxifen for either the whole 4 days (Tam d0-d4) or just for the first 2 days (Tam d0-d2) (Fig. 5D). The 1246 DARs showed their highest accessibility at day 4 in cells treated throughout with tamoxifen (Fig. 5E,F). In contrast, when tamoxifen was withdrawn after 2 days, these regions closed by day 4 (Fig. 5E,F) despite being accessible at day 2 (Fig. 5C). Together, these results show that OTX2 can induce accessibility in 1246 chromatin regions in both EpiLCs and during differentiation in the presence of PGC-inducing cytokines. However, the continued presence of OTX2 is essential to maintain accessibility of these regions.

These changes at the chromatin level are reflected by changes in transcription in tamoxifen-treated cells compared to untreated PGCLCs. The prolonged expression of OTX2 inhibited the reactivation of pluripotency-associated genes Nanog, Pou5f1 and Sox2 (Fig. EV5A). Prolonged OTX2 expression also reduced expression of Prdm14, although Prdm1 and Tfap2c remained expressed (Fig. EV5B), potentially due to expression in the endoderm (Prdm1) and surface ectoderm (Prdm1, Tfap2c) (Zhang et al, 2018a; Pijuan-Sala et al, 2019). In addition, enforcing OTX2 expression resulted in restoration of diverse somatic cells, as indicated by expression of endoderm (Sox17), mesoderm (Kdr/Flk1) and surface ectoderm (Dlx5) transcripts (Fig. EV5C). While somatic cells differentiated in the absence of cytokines show high level of the neural ectoderm marker Sox1, the overexpression of OTX2 is not sufficient to induce Sox1 expression (Fig. EV5C), as the presence of BMP4 inhibits neural fate (Di-Gregorio et al, 2007; Ying et al, 2003). Therefore, OTX2 expression diverts cells into somatic lineages, while BMP may influence the specific (proximal epiblast) lineages formed.

## OTX2 does not bind to PGCLC-specific accessible regions

The foregoing results show that OTX2 binds to regions associated with somatic differentiation. Whether OTX2 also binds to PGCLC-specific regions to restrict germline differentiation by

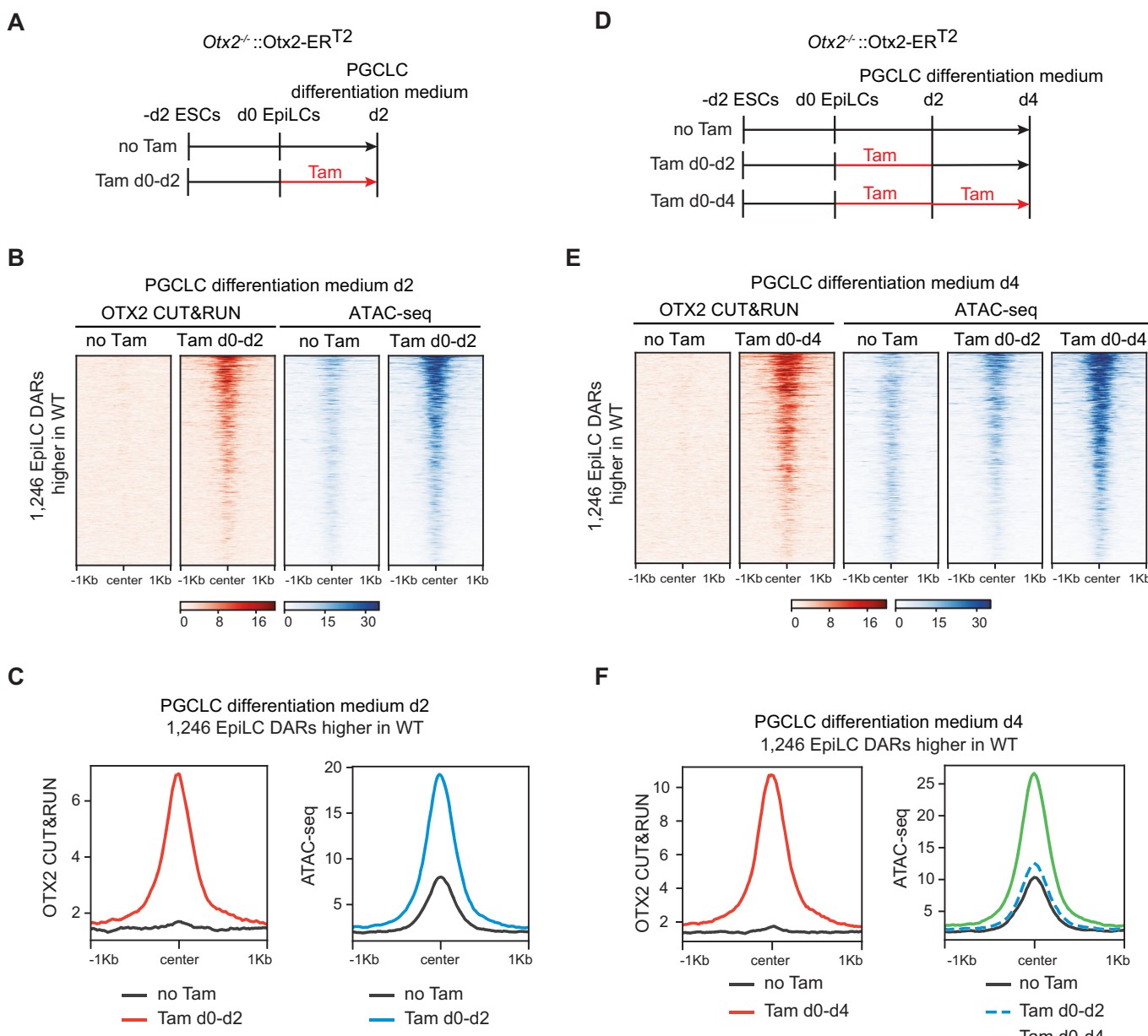

**Figure 5. Sustained OTX2 expression in early PGCLCs induces chromatin accessibility.**

(A, D) Differentiation schemes of *Otx2⁻/⁻*::Otx2-ER[T2] ESCs to PGCLCs +/− tamoxifen at day 2 (A) and day 4 (D). (B, E) Heatmaps of OTX2 CUT&RUN (red) and ATAC-seq (blue) signal in the 1246 EpiLC DARs where accessibility is increased by OTX2 in tamoxifen treated *Otx2⁻/⁻*::Otx2-ER[T2] PGCLCs at day 2 (B) and day 4 (E). (C, F) Average read density profiles of OTX2 CUT&RUN (red) and ATAC-seq (blue) at 1246 DARs where accessibility is increased by OTX2 in tamoxifen-treated *Otx2⁻/⁻*::Otx2-ER[T2] PGCLCs at day 2 (C) and day 4 (F).

inhibiting PGC-specific gene expression is unclear. To address this, we identified regions where chromatin is differentially accessible in cells that enter the germline (day 2 *Otx2⁻/⁻* PGCLCs) and in somatic cells (wild-type cells differentiated for 2 days in the absence of PGC-promoting cytokines). In total, 22,499 DARs are more accessible in somatic cells and 24,864 DARs are more accessible in PGCLCs (Fig. EV6A). To directly determine whether OTX2 binds to PGCLC DARs, we analysed the OTX2 CUT&RUN signal at these regions. In *Otx2⁻/⁻*::Otx2-ER[T2] PGCLCs treated with tamoxifen for 2 days, OTX2 is bound to EpiLC-specific

regions (Fig. 5B). However, in the same conditions, the vast majority of PGCLC DARs are not bound by OTX2-ER[T2] (Fig. EV6B). Therefore, OTX2 binding is not directly responsible for the accessibility of these DARs in PGCLCs. Moreover, we identified a subset of PGCLC-specific accessible regions that are only open in PGCLCs and closed in both ESCs and EpiLCs (Fig. EV6C,D). These 2617 PGCLC-specific regions are not bound by OTX2 in any of the analysed samples—wild-type EpiLC or tamoxifen-induced *Otx2⁻/⁻*::Otx2-ER[T2] d2 PGCLCs (Fig. EV6E). Together, these results suggest that OTX2 does not directly inhibit

PGCLC-specific regions prior to and during the first stage of PGCLC differentiation.

## OTX2 overexpression opens additional somatic regulatory regions

Since OTX2 can induce chromatin accessibility in EpiLCs, we speculated that OTX2 expression in somatic cells may also induce newly accessible regions. We compared tamoxifen-treated $Otx2^{-/-}$::Otx2-ER$^{T2}$ d2 aggregates, which are blocked for PGCLC differentiation due to OTX2 activity, with untreated $Otx2^{-/-}$::Otx2-ER$^{T2}$ d2 aggregates (PGCLCs) and with wild-type ESCs and EpiLCs (Fig. 2A). This identified 4221 regions with high accessibility only in tamoxifen-induced $Otx2^{-/-}$::Otx2-ER$^{T2}$ d2 aggregates (Fig. 6A). These regions are enriched for OTX2 motifs and, to a lesser extent, ZIC3 motifs (Fig. 6B). CUT&RUN indicates that only a subset of the additional 4221 accessible regions is bound by OTX2 (Fig. 6C). Gene ontology analysis of the genes closest to these additional accessible regions reveals an association with cell differentiation and multicellular organism development, in particular neural system development (Fig. 6D). Our results suggest that the enforced expression of OTX2 during the transition from EpiLCs to PGCLCs induces opening of additional somatic regions that may contribute to prevention of entry of cells into the germline.

Since OTX2 is not able to block germline differentiation when relocated to the nucleus after day 2 of PGCLC differentiation (Zhang et al, 2018a), we tested the ability of OTX2 to open chromatin at both the 1246 EpiLC DARs and the 4221 additional accessible regions. $Otx2^{-/-}$::Otx2-ER$^{T2}$ PGCLCs were differentiated for 4 days in the absence of tamoxifen and then treated for 1 h or 6 h at day 4 (Fig. 6E). The 1246 EpiLC regions that were previously shown to be more accessible in wild-type than $Otx2^{-/-}$ EpiLCs showed an increase in accessibility in response to a 1–6 h tamoxifen treatment at day 4 in PGCLC differentiation medium (Fig. 6F). In contrast, the 4221 accessible regions that were induced by enforced OTX2 expression at day 2 were unable to respond to OTX2 induced by a 1–6 h tamoxifen treatment after 4 days in PGCLC medium and remained closed (Fig. 6F). These results suggest that once cells have entered the germline, OTX2 loses the ability to open chromatin specifically at a subset of sites previously amenable to OTX2 action.

## Discussion

At the implantation stage, cells within the mouse epiblast undergo the important choice between somatic and germline differentiation. The transcription factor OTX2 plays a pivotal role in this choice, limiting the number of cells entering the germline, both in vivo and in vitro (Zhang et al, 2018a). In this work, we show how OTX2 operates globally on chromatin, in cells competent for both somatic and germline differentiation to promote a somatic chromatin environment that primes cells towards the somatic fate at the expense of the germline.

Using CUT&RUN we confirmed that OTX2 binds chromatin in both 2i/LIF ESCs and EpiLCs (Buecker et al, 2014). Motif analysis showed that these sites are, as expected, enriched for OTX2 binding motifs but also highlighted differences between the two cell types. In ESCs, sites bound by OTX2 are enriched for motifs of SOX and KLF TFs, are associated with genes involved in maintaining the stem cell population and are located distal to transcription units. In EpiLCs, the majority of sites bound by OTX2 are also located distally, but a higher proportion are closer to promoter regions than in ESCs. This may suggest that OTX2 is more frequently involved in transcriptional regulation by binding to promoter regions as cells exit naive pluripotency. Moreover, in EpiLC, OTX2-binding sites are enriched for ZIC motifs, rather than the KLF and SOX motifs found in ESC sites. The absence of the KLF and SOX2 motifs in OTX2 EpiLC-specific binding site can be attributed to the decreased expression of both *Klf4* and *Sox2* mRNAs after exit from the naive pluripotency (Hayashi et al, 2011; Boroviak et al, 2015; Yang et al, 2019a; Corsinotti et al, 2017). Therefore, while OTX2 is expressed in both ESCs and EpiLCs, we speculate that the partner proteins that OTX2 acts alongside differ in naive and formative pluripotent states.

The importance of protein partners in OTX2 function is also highlighted by the difference in the motif enrichment for ZIC family TF in subsets of differentially accessible regions observed in EpiLCs. In the 446 regions where OTX2 binds strongly, the only highly enriched motifs are OTX2-like. In contrast, in the remaining 800 EpiLC differentially accessible regions, OTX2 and ZIC motifs are equally enriched. *Zic2*, *Zic3* and *Zic5* are expressed in EpiLCs (Yang et al, 2019b, 2019a). These two subsets of regions show different OTX2 occupancy, implying differences in OTX2 affinity between the 446 and 800 DARs. This suggests that OTX2 requires a co-activator to open the 800 regions. In epiblast stem cells that have lost competence for the germline, OTX2 works with ZIC2 to establish a new regulatory network that controls primed pluripotency (Matsuda et al, 2017; Kondoh, 2024). OTX2 may therefore partner with ZIC2 or ZIC3 to control accessibility of somatic-associated regions that OTX2 cannot open on its own.

OTX2 is able to directly control accessibility at a subset of EpiLC regions, amongst which are regulatory regions of *Fgf5*, a well-known OTX2 target gene (Acampora et al, 2013; Buecker et al, 2014). This suggests that OTX2 acts as a pioneer TF to open chromatin during the exit from naive pluripotency. Moreover, this pioneering activity is not restricted to EpiLCs, as OTX2 also opens chromatin regions when its expression is enforced during the transition to PGCLCs. Interestingly, as reported for other TFs including OCT4 and SOX2 (Maresca et al, 2023; King and Klose, 2017), OTX2 is also required to maintain chromatin accessibility, as withdrawal of tamoxifen leads to closure of chromatin during PGCLC differentiation. An increase in chromatin accessibility can be achieved by binding of TFs to cognate motifs, followed by the action of chromatin remodellers that move or evict nucleosomes. Given the reported interactions between OTX2 and OCT4 (Buecker et al, 2014) and between OCT4 and BRG1 (King and Klose, 2017), one of the two mutually exclusive catalytic subunits of SWI/SNF remodelling complex, one hypothesis is that OTX2 binds to its motif, recruits OCT4 which in turn recruits BRG1 to induce opening of OTX2-bound DARs in EpiLCs. It will therefore be of interest in the future to determine the extent to which SWI/SNF, or other chromatin remodellers are required for OTX2 to open chromatin.

The ability of OTX2 to induce chromatin accessibility at somatic associated regulatory regions in cells that are competent for both somatic and germline fates suggests that the main action of OTX2 in the epiblast is to prime cells to respond to somatic-directing cues and differentiate towards the somatic fate. As the somatic fate

**A**

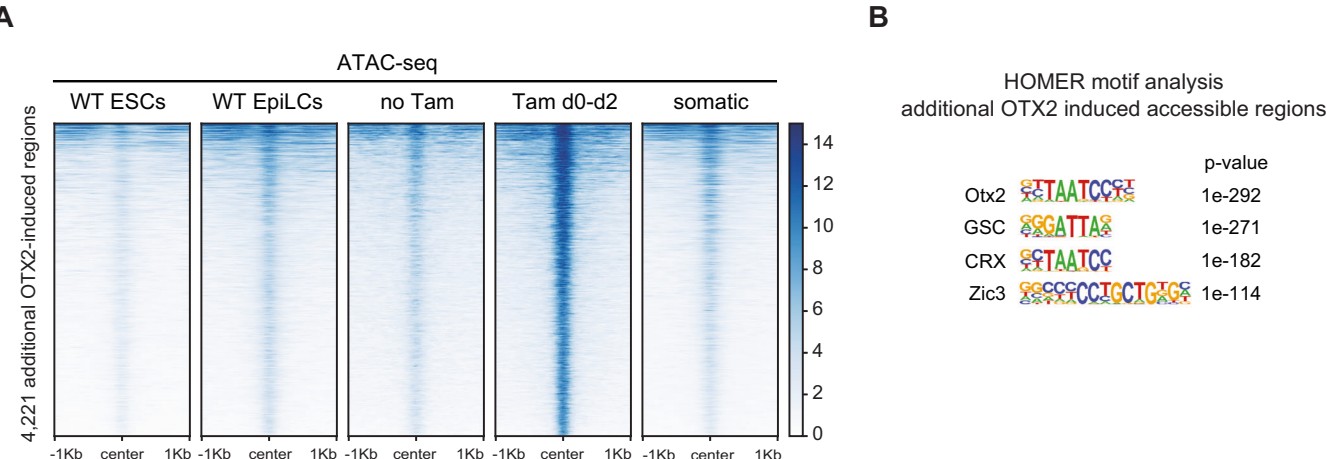

**B**

HOMER motif analysis
additional OTX2 induced accessible regions

**C**

**D**

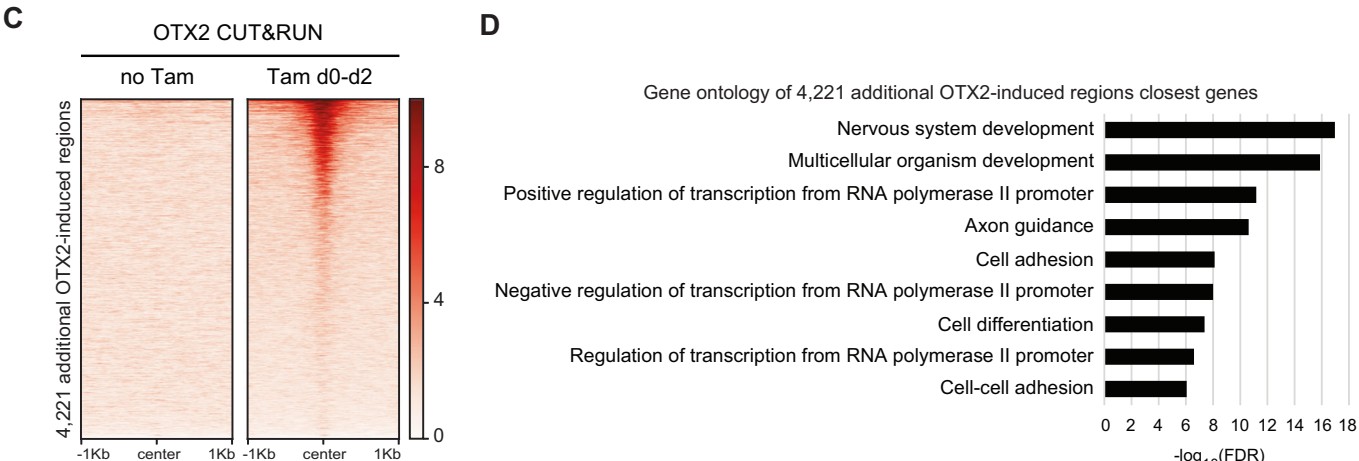

**E**

**F**

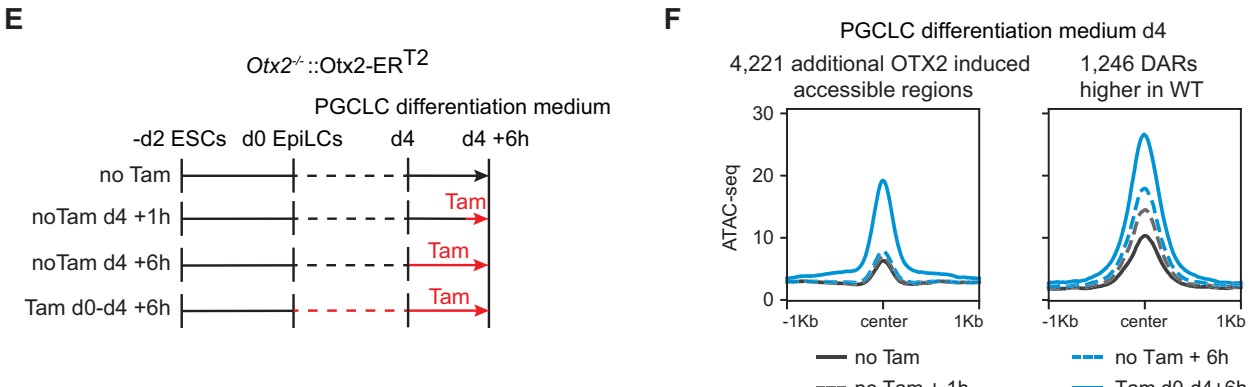

**Figure 6. OTX2 induces accessibility at additional somatic regions.**

(A) Heatmap of ATAC-seq signal in wild-type ESCs, wild-type EpiLCs, *Otx2*[−/−]::Otx2-ER[T2] d2 PGCLCs treated $+/-$ tamoxifen for 2 days and somatic cells at 4221 additional OTX2-induced regions. (B) Motif analysis in the additional OTX2-induced regions. (C) Heatmap of OTX2 CUT&RUN signal in tamoxifen-treated *Otx2*[−/−]::Otx2-ER[T2] d2 PGCLCs at 4221 OTX2-induced regions. (D) Gene ontology analysis of the closest genes to the newly accessible regions. (E) Differentiation scheme of *Otx2*[−/−]::Otx2-ER[T2] ESCs to day 4 PGCLCs with tamoxifen treatments for 4 days + 6 h, 6 h and 1 h only. (F) Average read density profiles of ATAC-seq signal in *Otx2*[−/−]::Otx2-ER[T2] PGCLCs treated with tamoxifen as shown in (E) at the 1246 EpiLC DARs higher in WT and at 4221 additional OTX2-induced PGCLC regions.

should be suppressed in developing PGCLCs to establish the PGC fate, somatic-associated regulatory regions should close. In this work, we showed that enforced expression of OTX2 during the transition to PGCLCs both maintains and induces opening of somatic-associated regulatory regions. OTX2 is able to open chromatin in only a subset of its regulated regions, as only regions that are already open in EpiLCs gain accessibility when OTX2 activity is induced in late PGCLCs. At this time, cells are already committed to germline differentiation and enforced expression of OTX2 cannot alter this cell fate. The additional regions where enforced expression of OTX2 induces accessibility during the initial 2 days of differentiation in the presence of PGC-inducing cytokines remain closed when OTX2 activity is induced at day 4 of PGCLC differentiation. This suggests that genes regulated by these regions may induce and establish somatic identity.

The OTX2-induced accessibility may arise from either increased accessibility in a subset of cells or more modest increase in all the cells within the population. In the first case, fluctuations in the levels of pluripotency-associated proteins, like NANOG, or somatic-inducing factors, like OTX2, in cells competent for both somatic and germline fates may explain how only a small subset of wild-type cells are able to respond to PGC-inducing cytokines and enter the germline. Cells with higher OTX2 levels in the population may be primed to commit to the somatic lineages, while cells with lower OTX2 levels may be prone to enter the germline. Investigating this hypothesis will require the future application of single-cell techniques to determine the level of heterogeneity in expression of somatic and pluripotency-associated factors in populations with dual competence, like EpiLCs.

The role of OTX2 in early development of mice and humans shows similarities. Downregulation of OTX2 via CRISPRi in hESCs leads to increased human PGCLC formation (Tang et al, 2022), while OTX2 overexpression inhibits PGCLC formation (Zhang et al, 2025). This suggests that, in humans and in mice OTX2 restricts germline entry. Despite this, the described mechanism of action of OTX2 appears to differ between the two species. A recent study showed that *OTX2* expression increases during the first day of PGCLC differentiation from primed human ESCs via incipient mesoderm-like cells (Zhang et al, 2025), while mouse *Otx2* is rapidly downregulated during differentiation of EpiLCs by PGC-inducing cytokines (Zhang et al, 2018a). Another difference is revealed when cells are differentiated in the absence of cytokines: while mouse Otx2-null EpiLCs differentiate to PGCLCs (Zhang et al, 2018a), human OTX2-null cells do not (Zhang et al, 2025). Moreover, in human ESCs OTX2 directly interacts with AP2γ and these TFs co-localise at human PGCLC-specific genes to suppress their precocious activation (Zhang et al, 2025). This suggests that, in contrast to our findings for mouse OTX2, human OTX2 blocks PGCLC differentiation by direct repression of germline genes. However, it is important to note that in vitro differentiation of PGCLCs in mouse and human cell lines starts from distinct states: naïve/formative in the mouse and primed in the human. It will therefore be interesting to learn in the future whether these species differences persist if differentiation of PGCLCs can be obtained from human naïve or formative, rather than primed pluripotent cells.

As OTX2 controls chromatin accessibility of somatic-associated regions, it seems likely that other TFs may act similarly to establish PGC identity. AP2 family proteins have an important role in determining PGC identity in humans; AP2α is expressed in a population of progenitors that give rise to PGC-like cells (Chen et al, 2019) while AP2γ functions as a pioneer TFs in naïve hESCs to

open chromatin at distal regulatory regions (Pastor et al, 2018). A recent paper showed that, in human cells, PRDM1 and PRDM14 each act alongside known pioneer TFs; PRDM1 with FOXA1 in definitive endoderm and PRDM14 with OCT4 in pluripotent stem cells (Matsui et al, 2024). It will be interesting to determine whether in human cells, AP2γ also has pioneering activity in PGCLCs and may therefore open PGC-specific regions together with PRDM1 and/or PRMD14.

In the absence of OTX2, chromatin regions that are accessible in ESCs fail to close during the transition to EpiLCs. Although we could not see a direct repressive action of OTX2 in these regions, the absence of OTX2 may indirectly influence the accessibility of pluripotency-associated regions. This could make such regions more prone to full reopening during the EpiLC to PGCLC transition, thereby supporting germline entry. A repressive role for OTX2 is already known for *Nanog* expression: mESCs cultured in medium containing serum and LIF express Nanog and Otx2 heterogeneously and reciprocally, with NANOG-high cells having a higher propensity to retain naïve pluripotency, and OTX2-high cells having a higher propensity to exit naïve pluripotency (Acampora et al, 2017). The OTX2-NANOG balance may similarly influence germline entry. Deletion of OTX2-specific binding sites in the proximal enhancer of NANOG leads to increased *Nanog* expression and PGCLC formation (Di Giovannantonio et al, 2021). In addition, Nanog overexpression is sufficient to induce PGCLC formation from EpiLCs (Murakami et al, 2016), with induction of NANOG repressing *Otx2* expression in the absence of cytokines (Vojtek et al, 2022). While the repressive action of OTX2 may also affect other pluripotency-associated regions, we could not identify OTX2 binding at PGCLC-specific regions, suggesting that OTX2 does not block germline differentiation by inhibiting opening of germline-specific regions. Further studies will be required to determine whether OTX2 represses additional pluripotency TF genes that become reactivated during germline entry. However, our model suggests that OTX2 plays a more active role in determining the choice between somatic and germline differentiation beyond the repression of NANOG alone.

In conclusion, in this work we generated novel information regarding the molecular mechanism of action of OTX2 in controlling the choice between somatic and germline differentiation. This shows that OTX2 can control accessibility of somatic-associated regulatory regions, preparing the cells to initiate the somatic fate at the expense of the germline. Regions that OTX2 opens in EpiLCs are always permissive to its action, while additional somatic-associated regulatory regions become refractory to the presence of OTX2 once the germline fate is already established. Whether regulatory regions controlling genes that can block germline differentiation are among the latter subset is an interesting question for future analysis.

## Methods

**Reagents and tools table**

| Reagent/resource | Reference or source | Identifier or catalog number |
| --- | --- | --- |
| **Experimental models** | | |
| E14Tg2a (*M. musculus*) | Hooper et al, 1987 | N/A |
| Otx2<sup>−/−</sup> (*M. musculus*) | Acampora et al, 2013 | N/A |

| Reagent/resource | Reference or source | Identifier or catalog number |
|---|---|---|
| Otx2<sup>−/−</sup>::Otx2-ER<sup>T2</sup> (M. musculus) | Zhang et al, 2018a | N/A |
| **Antibodies** | | |
| Goat anti-human OTX2 | R&D Systems | AF1979 |
| Alexa Fluor 647 mouse anti-CD15 (SSEA-1) | Biolegend | 125607 |
| Phycoerythrin (PE) hamster anti-CD61 | Biolegend | 104307 |
| **Oligonucleotides and other sequence-based reagents** | | |
| RT-qPCR primers | This study | See list of RT-qPCR primers in "Methods" |
| **Chemicals, enzymes and other reagents** | | |
| Glasgow minimum essential medium (GMEM) | Sigma | G5154 |
| Fetal bovine serum | Thermo Fisher Scientific | 10270106 |
| L-glutamine | Invitrogen | 25030-024 |
| Pyruvate solution | Invitrogen | 11360-039 |
| MEM non-essential amino acids | Invitrogen | 11140-035 |
| 2-Mercaptoethanol | Gibco | 31350010 |
| Gelatine | Sigma | G1890-500G |
| Leukemia Inhibitory Factor (LIF) | Homemade | N/A |
| Trypsin | Thermo Fisher Scientific | 15090046 |
| 1X DPBS | Gibco | 14190094 |
| DMEM/F-12 | Thermo Fisher Scientific/Gibco | 21041025 |
| Neurobasal | Thermo Fisher Scientific/Gibco | 12348017 |
| Apo-transferrin | Sigma | T1147-100MG |
| Bovine Albumin Fraction V | Thermo Fisher Scientific | 15260037 |
| Progesterone | Sigma | P8783 |
| Putrescine | Sigma | P5780 |
| Sodium selenite | Sigma | S5261 |
| Insulin | Sigma | I1882-100MG |
| B27 without vitamin A | Thermo Fisher Scientific | 12587010 |
| Penicillin-streptomycin | Thermo Fisher Scientific | 15140122 |
| PD0325901 | Stem Cell Technologies | 72182 |
| CHIR99021 | APExBIO | B5779 |
| ESGRO LIF | Sigma/Millipore | ESG1106 |

| Reagent/resource | Reference or source | Identifier or catalog number |
|---|---|---|
| Poly-L-ornithine | Sigma | P3655 |
| Laminin | Corning | 354232 |
| Fibronectin | Sigma | FC010 |
| Fibronectin | Sigma | F1141 |
| Human Activin A | PeproTech | 120-14 |
| Human FGF-basic | Thermo Fisher Scientific | 13256029 |
| Knock-out serum replacement (KOSR) | Thermo Fisher Scientific/Gibco | 10828028 |
| TrypLE Express Enzyme | Thermo Fisher Scientific | 12604021 |
| Bone morphogenetic protein (BMP) 4 | Qkine | Qk038_BMP4_25 µg |
| Bone morphogenetic protein (BMP) 8a | R&D Systems | 1073-BP-010 |
| Stem cell factor (SCF) | R&D Systems | 455-MC-010 |
| Epidermal growth factor (EGF) | R&D Systems | 2028-EG-200 |
| 4-hydroxytamoxifen | Sigma | H7904-5MG |
| EDTA/Trypsin | Thermo Fisher Scientific | 25200072 |
| Direct-zol RNA MiniPrep kit | Zymo Research | R2025 |
| High-Capacity cDNA Reverse Transcription kit | Applied Biosystem | 4368814 |
| Takyon SYBR MasterMix | Eurogentec | UF-NSMT-B0701 |
| pAG-MNase | Homemade | N/A |
| NEBNext Ultra II DNA Library prep kit | New England Biolabs | E7645S |
| NEBNext Multiplex Oligos for Illumina | New England Biolabs | E7335S, E7500S, E7710S, E7730S |
| High Sensitivity D1000 ScreenTapes | Agilent | 5067-5584 |
| High Sensitivity D1000 Reagents | Agilent | 5067-5585 |
| ATAC-seq kit | Active Motif | 53150 |
| DNA Clean and Concentration-5 kit | Zymo Research | D4013 |
| NEBNext High Fidelity 2x PCR Master Mix | New England Biolabs | M0541S |
| **Software** | | |
| FlowJo v10 | https://www.flowjo.com/ | |
| GraphPad Prism version 10.5.0 | https://www.graphpad.com/ | |

| Reagent/resource | Reference or source | Identifier or catalog number |
|---|---|---|
| FastQC v. 0.11.9 | Andrews https://www.bioinformatics.babraham.ac.uk/projects/fastqc/ | |
| TrimGalore v. 0.6.6 | Krueger et al, 2021 | |
| Cutadapt v. 1.9.1 | Martin, 2011 | |
| Burrows-Wheeler Alignment (BWA) v. 0.7.16 | Li, 2013 | |
| Samtools v. 1.6 | Li et al, 2009 | |
| Picard v. 2.23.3 | Broad Institute, 2019 | |
| Model-Based Analysis of ChIP-Seq (MACS2) v. 2.1.1 | Zhang et al, 2008 | |
| BEDTools v. 2.27.1 | Quinlan and Hall, 2010 | |
| Irreproducible Discovery Rate (IDR) v. 2.0.4.2 | Li et al, 2011 | |
| DeepTools v. 3.5.1 | Ramírez et al, 2016 | |
| DESeq2 v. 3.18 | Love et al, 2014 | |
| RStudio v.2024.12.1 | https://posit.co/download/rstudio-desktop/ | |
| ChIPseeker v 3.8 | Wang et al, 2022; Yu et al, 2015 | |
| Hypergeometric Optimization of Motif EnRichment (HOMER) v. 4.11 | Heinz et al, 2010 | |
| Database for Annotation, Visualization and Integrated Discovery (DAVID) database | Huang et al, 2009; Sherman et al, 2022 | |
| **Other** | | |
| Cell repellent U-bottom 96-well plates | Greiner Bio-one | 650970 |
| BD LSR Fortessa (Flow cytometry) | BD | |
| Roche LightCycler 480 (qPCR) | Roche | |
| Tapestation 2200 | Agilent | |
| NextSeq 500 | Illumina | |
| NextSeq 2000 | Illumina | |
| NovaSeq 6000 | Illumina | |

## Cell culture

E14Tg2a (Hooper et al, 1987), $Otx2^{-/-}$ (Acampora et al, 2013) and $Otx2^{-/-}$::Otx2-ER$^{T2}$ (Zhang et al, 2018a) cell lines were maintained in serum/LIF medium (Glasgow minimum essential medium (GMEM)

Sigma, cat. G5154), 10% fetal bovine serum (Thermo Fisher Scientific, cat. 10270106), 2 mM L-glutamine (Invitrogen, cat. 25030-024), 1 mM pyruvate solution (Invitrogen, cat. 11360-039), 1× MEM non-essential amino acids (Invitrogen, cat. 11140-035), 0.1 mM 2-mercaptoethanol (Gibco, cat. 31350010), 100 U/ml homemade LIF in gelatin-coated flasks in a 37 °C, 7.5% $CO_2$ humidified incubator. Cells were routinely tested for mycoplasma contamination.

EpiLC and PGCLC differentiation was performed as previously described (Hayashi and Saitou, 2013) with a few modifications. First, ESC grown in serum/LIF were dissociated with Trypsin (Thermo Fisher Scientific, cat. 15090046), washed with PBS to remove any residual medium and resuspended in N2B27 medium (DMEM/F-12 (Thermo Fisher Scientific/Gibco cat. 21041025) supplemented with homemade N2 (DMEM/F-12, 11.1111 mg/ml apo-transferrin (Sigma, cat. T1147-100MG), 0.55% (w/v) Bovine Albumin Fraction V (Thermo Fisher Scientific, cat. 15260037), 2.2 µg/ml progesterone (Sigma, cat. P8783), 1.778 mg/ml putrescine (Sigma, cat. P5780), 3 mM sodium selenite (Sigma, cat. S5261) and 12.5 µg/ml insulin (Sigma, cat. I1882-100MG)) mixed 1:1 with Neurobasal (Thermo Fisher Scientific/Gibco cat. 12348017) supplemented with B27 without vitamin A (Thermo Fisher Scientific cat. 12587010), L-glutamine and penicillin-streptomycin (Thermo Fisher Scientific, cat. 15140122), followed by addition of 1.8 ml of 50 mM 2-Mercaptoethanol) supplemented with 0.4 µM PD0325901 (Stem Cell Technologies, cat. 72182), 3 µM CHIR99021 (APExBIO, cat. B5779) and 100 U/ml ESGRO LIF (Sigma/Millipore, cat. ESG1106) and adapted to 2i/LIF medium for at least three passages on poly-L-ornithine (Sigma, cat. P3655) and laminin (Corning, cat. 354232) coated 6-well plates. In total, $1.0 × 10^5$ ESCs were washed with 1× PBS and plated on a well of a 12-well plate pre-coated with 16.6 µl/ml fibronectin (Sigma, cat. FC010 and cat. F1141) in EpiLC medium: N2B27 medium supplemented with 20 ng/ml Human Activin A (PeproTech, cat. 120-14), 12 ng/ml Human FGF-basic (Thermo Fisher Scientific, cat. 13256029) and 1% knock-out serum replacement (KOSR; Thermo Fisher Scientific/Gibco, cat. 10828028). Medium was freshly replaced after 24 h. After 44 h in EpiLC medium, cells were harvested with TrypLE Express Enzyme (Thermo Fisher Scientific, cat. 12604021), washed once with 1× PBS containing 1% bovine albumin followed by a second wash in 1× PBS. EpiLCs were then collected for analysis or for PGCLC differentiation.

For PGCLC differentiation, $1.5 × 10^5$ EpiLCs were resuspended in 5 ml ($3 × 10^4$ cells/ml) of GK15 medium (GMEM supplemented with 15% KOSR, 2 mM L-glutamine, 1 mM pyruvate solution, 1× MEM non-essential amino acids, 100 U/ml penicillin/streptomycin, 0.1 mM 2-Mercaptoethanol) supplemented with 50 ng/mL bone morphogenetic protein (BMP) 4 (Qkine, cat. Qk038_BMP4_25 µg), 50 ng/mL BMP8a (R&D Systems, cat. 1073-BP-010), 10 ng/mL stem cell factor (SCF) (R&D Systems, cat. 455-MC-010), 10 ng/mL epidermal growth factor (EGF, R&D System, cat. 2028-EG-200), and 1000 U/mL ESGRO and replated 100 µl per well of a cell repellent U-bottom 96-well plate (Greiner Bio-one, cat. 650970). For cytokine-free differentiation, $1.5 × 10^5$ EpiLCs were resuspended in 5 ml of GK15 medium without cytokines. Cells were collected at day 2 and day 4 for analysis and day 6 for flow cytometry analysis.

$Otx2^{-/-}$::Otx2-ER$^{T2}$ cells were treated with 1 µM of 4-hydroxytamoxifen for the indicated times. To change tamoxifen-containing media, aggregates were collected at day 2, washed, resuspended in GK15 and incubated in 5 mm dishes in rotation until day 4 and day 6 before collection and analysis.

## Flow cytometry

Flow cytometry analysis was performed as previously described (Zhang et al, 2018a, 2018b) with minor changes. Cells were collected, dissociated using EDTA/Trypsin (Thermo Fisher Scientific, cat. 25200072) and neutralised in GK15 medium. Cells were centrifuged, washed with 1× PBS and resuspended in 100 μl 1× PBS/1% KOSR supplemented with Alexa Fluor 647 anti-CD15 (SSEA-1) (Biolegend, cat. 125607) and Phycoerythrin (PE) anti-CD61 (Biolegend, cat. 104307) antibodies diluted 1/200 and 1/600, respectively. Following a 10 min incubation at RT in the dark, cells were washed twice in 1× PBS and resuspended in 250 μl 1× PBS/1% KOSR supplemented with DAPI for live-cell selection. Acquisition was performed on a BD LSR Fortessa instrument and data was analysed using FlowJo v10.

## RT-qPCR analysis

Total RNA was isolated using the Direct-zol RNA MiniPrep kit (Zymo Research, cat. R2025) with in column DNAse treatment, following manufacturer's instructions. The quantity and purity of RNA samples were determined using a micro-volume spectrophotometer (Nanodrop, ND-1000). RNA was reverse-transcribed with High-Capacity cDNA Reverse Transcription kit (Applied Biosystem, cat. 4368814) using random hexamer oligonucleotides, following manufacturer's instructions. Obtained cDNA was diluted to 3 ng/μl in nuclease-free water. Triplicate qPCR reactions were set up with the Takyon SYBR MasterMix (Eurogentec, cat. UF-NSMT-B0701) using 15 ng cDNA for each reaction and analysed using the Roche LightCycler 480 machine. For all qPCR primer pairs melting curves were generated to verify the production of single DNA species. Plotting and statistical analysis (one-way Anova with Tukey's correction for multiple comparisons) were performed using the software GraphPad Prism version 10.5.0.

**List of primers used for RT-qPCR**

| Primer pairs | Forward primer | Reverse primer |
| --- | --- | --- |
| TBP | GGGGAGCTGTGATGTGAAGT | CCAGGAAATAATTCTGGCTCA |
| Fgf5 | TGTGTCTCAGGGGATTGTAGG | AGCTGTTTTCTTGGAATCTCTCC |
| Oct4 | GTTGGAGAAGGTGGAACCAA | CTCCTTCTGCAGGGCTTTC |
| Nanog | AGGATGAAGTGCAAGCGGTG | TGCTGAGCCCTTCTGAATCAG |
| Sox2 | CACAGATGCAACCGATGCA | GGTGCCCTGCTGCGAGTA |
| Tfap2c | ATCCCTCACCTCTCCTCTCC | CCAGATGCGAGTAATGGTCGG |
| Prdm1 | TCCTGGAGAGCTCACAGTGATA | CGCTGTACTCTCTCTTGGGG |
| Prdm14 | CCCTACCTGTGTTCAACCT | GTCTCCAGAGTGGACTCTC |
| Sox17 | CACAACGCAGAGCTAAGCAA | CGCTTCTCTGCCAAGGTC |
| Sox1 | GTGACATCTGCCCCCATC | GAGGCCAGTCTGGTGTCAG |
| Kdr | CCACAGAACAACTCAGGGCTA | GGGAGCAAAGTCTCTGGAAA |
| Dlx5 | CCAAGGCTTATGCCGACTACG | TCCGCCACTTCTTTCTCTGGC |

## CUT&RUN

Cleavage Under Targets & Release Using Nuclease (CUT&RUN) was performed as previously reported (Skene et al, 2018) with no modifications using $5 \times 10^5$ cells per replicate, two biological replicates per sample, 1:100 dilution of anti-OTX2 (R&D Systems,

cat. AF1979) and homemade pAG-MNase. Barcoded libraries were generated with the NEBNext Ultra II DNA Library prep kit (E7645S) and NEBNext Multiplex Oligos for Illumina (E7335S, E7500S, E7710S, E7730S) with modifications. In total, 5 ng of DNA was used per library. End-prep and ligation were performed according to the manufacturer's instructions. In all, 1 μl of USER enzyme was used per sample. For the final PCR step, 1 μl of both universal and indexed primers were used, and the annealing/extension step was decreased to 30 s to reduce amplification of large fragments. Clean-up steps were performed with homemade PCR purification beads. Library quality was assessed with HSD1000 ScreenTapes (Agilent, cat. 5067-5584 and 5067-5585) in a Tapestation 2200 (Agilent). Libraries were sequenced paired-end on Illumina NextSeq 500 and NextSeq 2000.

## CUT&RUN analysis

Quality of reads was assessed by FastQC (v. 0.11.9 (Andrews)), index and adaptor sequences were trimmed using TrimGalore (v. 0.6.6 (Krueger et al, 2021)) and Cutadapt (v. 1.9.1 (Martin, 2011)) followed by a second quality check with FastQC. Sequences were aligned to the mm10 mouse reference genome using Burrows-Wheeler Alignment (BWA, v. 0.7.16 (Li, 2013)) tool with the maximal exact matches (MEM) option. Fragments were quality filtered ( > 10) with Samtools (v. 1.6 (Li et al, 2009)) and PCR duplicates were marked with Picard (v. 2.23.3 (Broad Institute, 2019)) and removed with Samtools. Reads of OTX2 CUT&RUN in ESCs and EpiLCs were normalised to the number of e.coli reads before further analysis.

Peaks were called with Model-Based Analysis of ChIP-Seq (MACS2, v. 2.1.1 (Zhang et al, 2008)) at 5% FDR using parameter -f BAMPE for paired-end input. Blacklisted regions were subtracted from the peak list using BEDTools (v. 2.27.1 (Quinlan and Hall, 2010)) and reproducible peaks between the two biological replicates were identified with IDR (v. 2.0.4.2 (Li et al, 2011)). Bedtools was used to identify ESC-specific, EpiLC-specific and common OTX2-bound regions and following intersections with ATAC-seq data. Bigwig tracks were generated with the bamCoverage option in the python-based DeepTools (v. 3.5.1 (Ramírez et al, 2016)).

## ATAC-seq

ATAC-seq was performed with modifications from the original Buenrostro protocol (Buenrostro et al, 2013, 2015) using $5 \times 10^4$ cells per replicate, two biological replicates per sample. Cells were washed in PBS, incubated for 20 min in lysis buffer (10 mM Tris-HCl pH 7.4, 10 mM NaCl, 3 mM $MgCl_2$, 0.1% (v/v) Igepal) on ice. After centrifugation, nuclei were resuspended in 25 μl of freshly prepared Tagmentation Master Mix containing Assembled Transposomes (Active Motif, cat. 53150). Tagmentation was performed in a thermocycler at 37 °C for 20 min. Tagmented DNA was recovered using the DNA Clean and Concentration-5 kit (Zymo Research, cat. D4013). Libraries were prepared using NEBNext High Fidelity 2x PCR Master Mix and indexed primers reported in (Buenrostro et al, 2013) using the following protocol: 72 °C for 5 min, and 98 °C for 30 s followed by nine cycles of: 98 °C for 10 s, 63 °C for 30 s, 72 °C for 1 min. Library quality was assessed with HSD1000 tapes in a Tapestation Instrument. Libraries were

sequenced paired-end on Illumina NovaSeq 6000 and NextSeq 2000.

## ATAC-seq analysis

ATAC-seq analysis was performed with a pipeline similar to CUT&RUN with the following modifications. After alignment with BWA-MEM, mitochondrial fragments were eliminated with Samtools. Mapped reads were shifted by +4 bp for the forward strand and -5 bp for the reverse strand. ESCs, EpiLCs, PGCLC d2 and somatic cells were normalised by read depth before peak calling. Shifted bam files were transformed into BED files for peak calling using MACS2.

A list of all accessible regions in the six samples analysed in this study was created and used as input for multiCov tool from the deepTools suite together with bam files to retrieve the coverage of each region in each sample. The resulting matrix was used as input for differential accessibility analysis pairwise comparisons with DESeq2 (v. 3.18 (Love et al, 2014)). Volcano plots were generated by DESeq2.

## Peak annotation

Lists of identified regions have been analysed with R package ChIPseeker (v 3.8 (Wang et al, 2022; Yu et al, 2015)) to annotate their position corresponding to the closest gene. The results of ChIPseeker were used to generate pie charts. The list of closest genes was used as input for Gene Ontology analysis.

## Motif and Gene Ontology analyses

Motif analysis was performed with Hypergeometric Optimization of Motif EnRichment (HOMER v. 4.11 (Heinz et al, 2010)) using findMotifsGenome.pl and options -size given -nomotif. Gene ontology was performed with Database for Annotation, Visualization and Integrated Discovery (DAVID) database (Huang et al, 2009; Sherman et al, 2022) using the ENSEMBL gene annotation for the closest gene to each region annotated by ChIPSeeker.

## Data visualization

Average bigwigs between two biological replicates were generated with the deepTools suite (bigwigAverage). The average bigwig file was used to generate heatmaps and average profiles with computeMatrix (reference-point, options --referencePoint center --sortRegions descend --skipZeros --missingDataAsZero), plotHeatmap and plotProfiles of the deepTools suite.

## Data availability

Original ATAC-seq and CUT&RUN data are deposited at the NCBI Gene Expression Omnibus (GEO) with accession numbers GSE289298 (ATAC-seq) and GSE289297 (CUT&RUN). Public ChIP-seq datasets for H3K4me1 and H3K27ac in ESCs and EpiLCs are available through NCBI Gene Expression Omnibus with accession numbers GSE155062.

The source data of this paper are collected in the following database record: biostudies:S-SCDT-10_1038-S44319-025-00622-2.

## Peer review information

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

## Acknowledgements

We thank Claire Cryer and Fiona Rossi for assistance with flow cytometry, Edward Foo for preliminary ATAC-seq analysis, the Genetics Core at the Western General for NGS sequencing, Val Wilson and all members of the Chambers lab for feedback and suggestions. pAG-MNase was a gift from Abdenour Soufi and Burak Ozkan (University of Edinburgh). This work was funded by UK Medical Research Council Grant MR/T003162/1 to IC and by a Marie Sklodowska-Curie fellowship (H2020-MSCAIF-2018/843879) to EB.

## Author contributions

**Elisa Barbieri**: Conceptualization; Formal analysis; Funding acquisition; Investigation; Methodology; Writing—original draft; Writing—review and editing. **Ian Chambers**: Conceptualization; Supervision; Funding acquisition; Writing—original draft; Writing—review and editing.

Source data underlying figure panels in this paper may have individual authorship assigned. Where available, figure panel/source data authorship is listed in the following database record: biostudies:S-SCDT-10_1038-S44319-025-00622-2.

## Disclosure and competing interests statement

The authors declare no competing interests.

# Expanded View Figures

**Figure EV1.   Characterization of OTX2-bound regions.**

(**A**) Tracks of two replicates of OTX2 CUT&RUN at the *Tet2* and *Fgf5* loci. (**B**) Genomic distribution of OTX2-bound regions; promoters are defined as $+/-$ 1 kb from a TSS. (**C**) Heatmap of H3K4me1 (pink) and H3K27ac (blue) signal at ESC-specific, common and EpiLC-specific OTX2-bound distal regions. (**D**) Motif analysis in ESC-specific, common and EpiLC-specific regions. (**E**) Gene ontology of the closest genes to ESC-specific and EpiLC-specific OTX2-bound regions.

▶

**A**

WT ESC rep1 [0 - 100]
WT ESC rep2 [0 - 100]

Tet2

WT EpiLC rep1 [0 - 208]
WT EpiLC rep2 [0 - 208]

Fgf5

**B**

ESC-specific
9%
91%

common
25%
75%

EpiLC-specific
36%
64%

■ promoters
■ distal elements

**C**

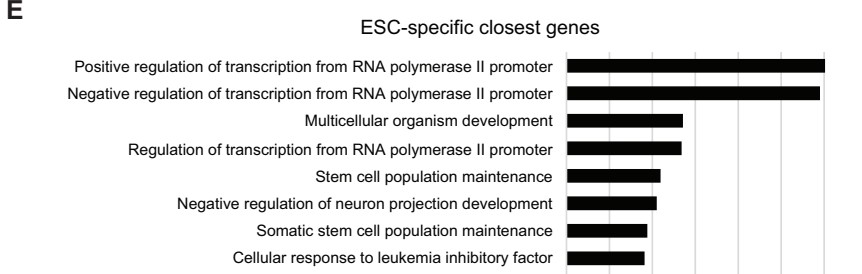

**D**

ESC-specific regions
| | | p-value |
|---|---|---|
| GSC | | 1e-144 |
| Otx2 | | 1e-124 |
| OCT4-SOX2 | | 1e-121 |
| CRX | | 1e-110 |
| Sox2 | | 1e-103 |
| Sox3 | | 1e-98 |
| KLF5 | | 1e-90 |
| Sox15 | | 1e-71 |
| KLF3 | | 1e-71 |
| Sox10 | | 1e-62 |

common regions
| | | p-value |
|---|---|---|
| Otx2 | | 1e-158 |
| GSC | | 1e-117 |
| CRX | | 1e-111 |
| OCT4-SOX2 | | 1e-96 |
| Sox2 | | 1e-76 |
| Sox17 | | 1e-51 |
| Sox3 | | 1e-48 |
| Sox15 | | 1e-42 |
| KLF3 | | 1e-40 |
| Pitx1 | | 1e-37 |

EpiLC-specific regions
| | | p-value |
|---|---|---|
| Otx2 | | 1e-354 |
| GSC | | 1e-305 |
| CRX | | 1e-266 |
| Pitx1 | | 1e-105 |
| OCT4-SOX2 | | 1e-88 |
| Sp5 | | 1e-63 |
| Zic | | 1e-61 |
| Oct4 | | 1e-59 |
| Brn1 | | 1e-57 |
| Zic3 | | 1e-52 |

**E**

ESC-specific closest genes

Positive regulation of transcription from RNA polymerase II promoter
Negative regulation of transcription from RNA polymerase II promoter
Multicellular organism development
Regulation of transcription from RNA polymerase II promoter
Stem cell population maintenance
Negative regulation of neuron projection development
Somatic stem cell population maintenance
Cellular response to leukemia inhibitory factor
Negative regulation of transcription, DNA-templated

$-\log_{10}(FDR)$

EpiLC-specific closest genes

Multicellular organism development
Positive regulation of transcription from RNA polymerase II promoter
Negative regulation of transcription from RNA polymerase II promoter
Axon guidance
Regulation of transcription from RNA polymerase II promoter
Nervous system development
Cell proliferation
Positive regulation of transcription, DNA-templated
Cell differentiation
Cell-cell adhesion
Central nervous system development

$-\log_{10}(FDR)$

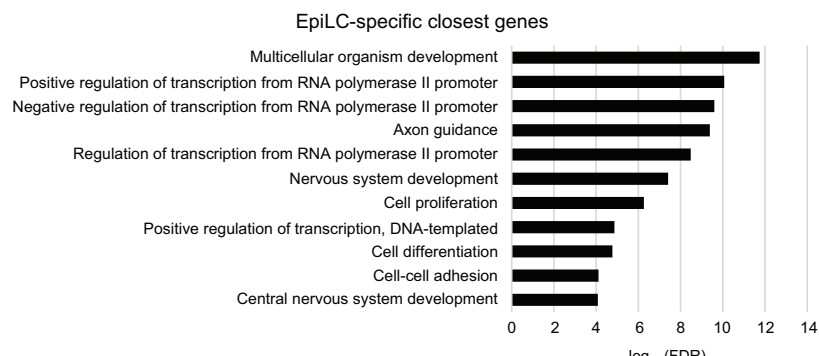

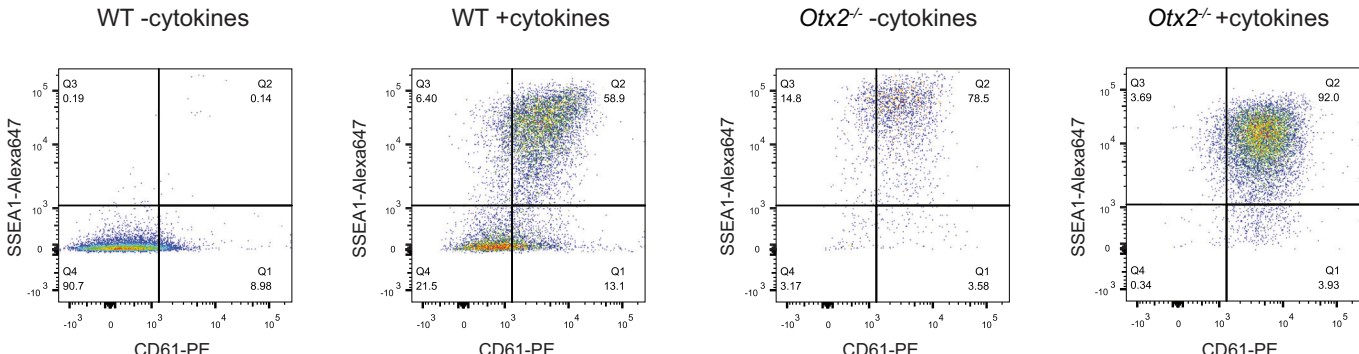

**Figure EV2. CD61 and SSEA1 expression at day 6 of PGCLC differentiation.**

Flow cytometry plots showing surface expression of CD61 and SSEA1 in d6 aggregates from wild-type and *Otx2*$^{-/-}$ cells cultured in the presence or absence of PGCLC-inducing cytokines.

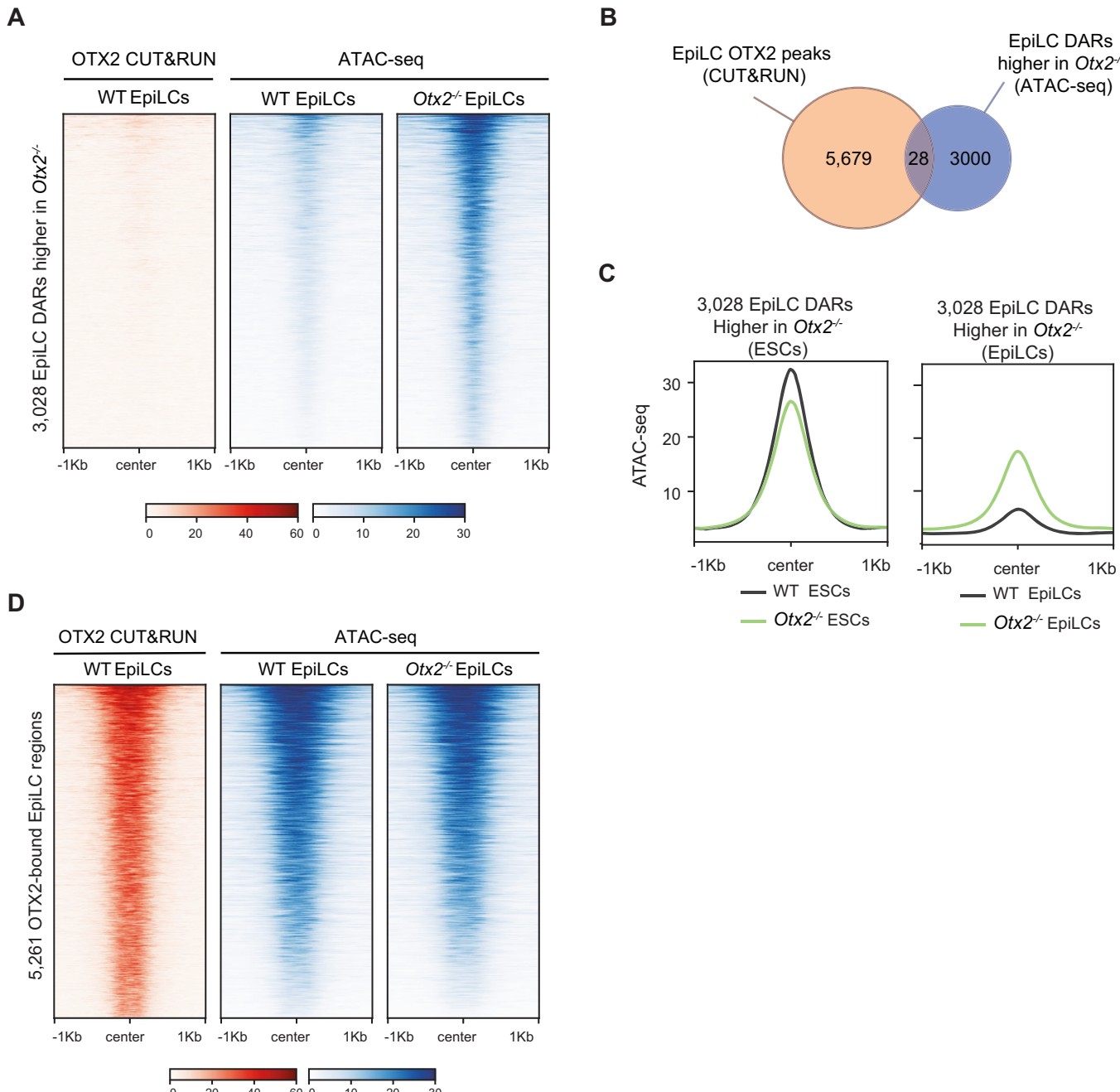

**Figure EV3.  OTX2 indirectly controls closure of chromatin in the ESC to EpiLC transition.**

(A) Heatmap of OTX2 binding (CUT&RUN - red) and accessibility (ATAC-seq - blue) at 3028 EpiLC regions that show increased accessibility in *Otx2*$^{-/-}$ EpiLCs. (B) Venn diagram of the overlap of OTX2-bound regions in EpiLCs (orange) and EpiLC DARs that are more accessible in *Otx2*$^{-/-}$ EpiLCs (blue). (C) Read density profiles of ATAC-seq in wild-type and *Otx2*$^{-/-}$ ESCs (left) and wild-type and *Otx2*$^{-/-}$ EpiLCs (right) at the 3028 EpiLC DARs that are more accessible in *Otx2*$^{-/-}$ EpiLCs. (D) Heatmap of OTX2 binding (CUT&RUN - red) and accessibility (ATAC-seq - blue) at 5261 OTX2-bound EpiLC regions that do not show accessibility changes in *Otx2*$^{-/-}$ EpiLCs.

   

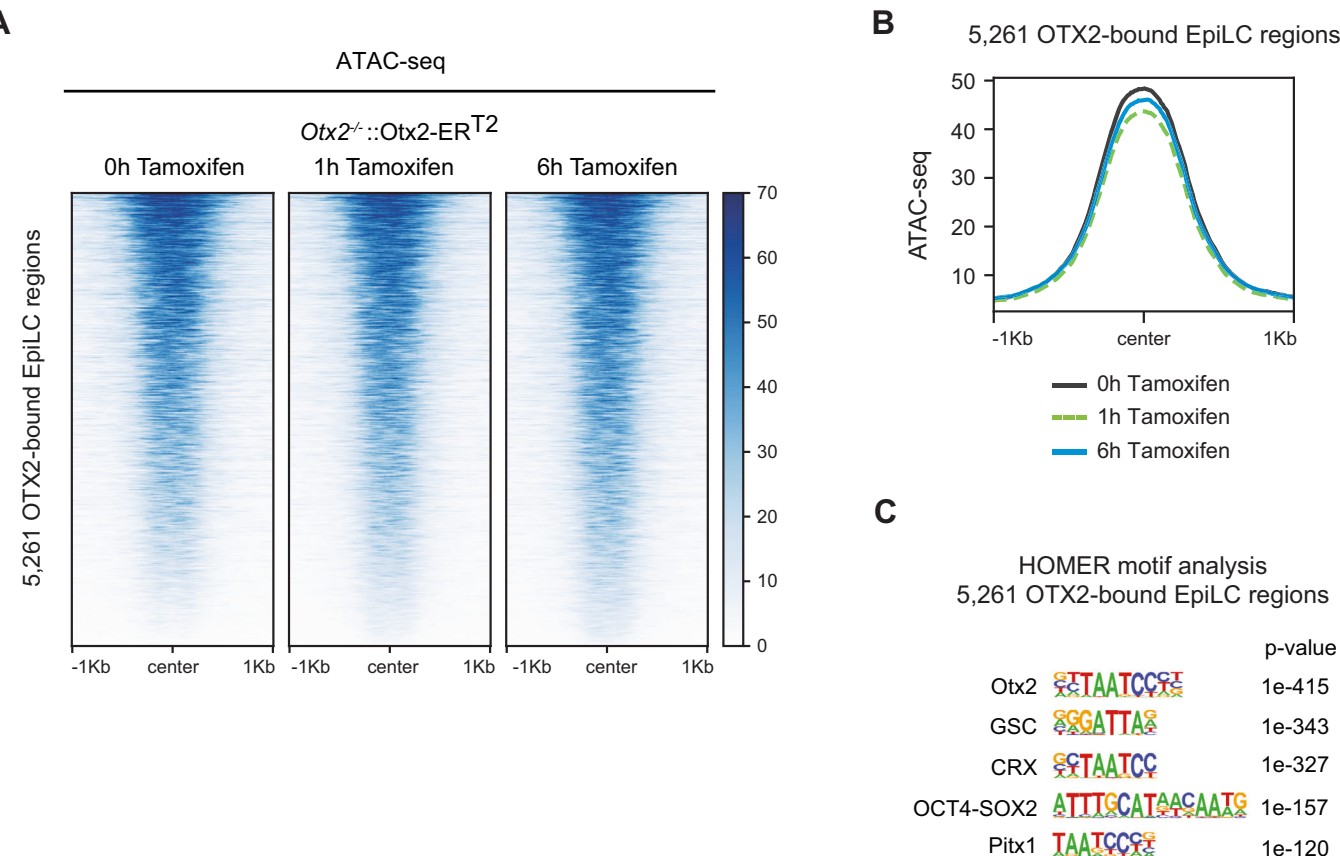

**Figure EV4. Most OTX2-bound EpiLC regions do not change accessibility in *Otx2<sup>−/−</sup>* cells.**

(A) Heatmap of ATAC-seq signal at the 5261 OTX2-bound EpiLC regions that do not change accessibility in *Otx2<sup>−/−</sup>*::Otx2-ER<sup>T2</sup> EpiLCs when treated for 1 or 6 h with tamoxifen. (B) Average read density profile of ATAC-seq signal at the 5261 OTX2-bound EpiLC regions in *Otx2<sup>−/−</sup>*::Otx2-ER<sup>T2</sup> EpiLCs treated for 1 h (dashed green) or 6 h (blue) with tamoxifen compared to untreated cells (black). (C) Motif analysis in the 5261 OTX2-bound EpiLC regions.

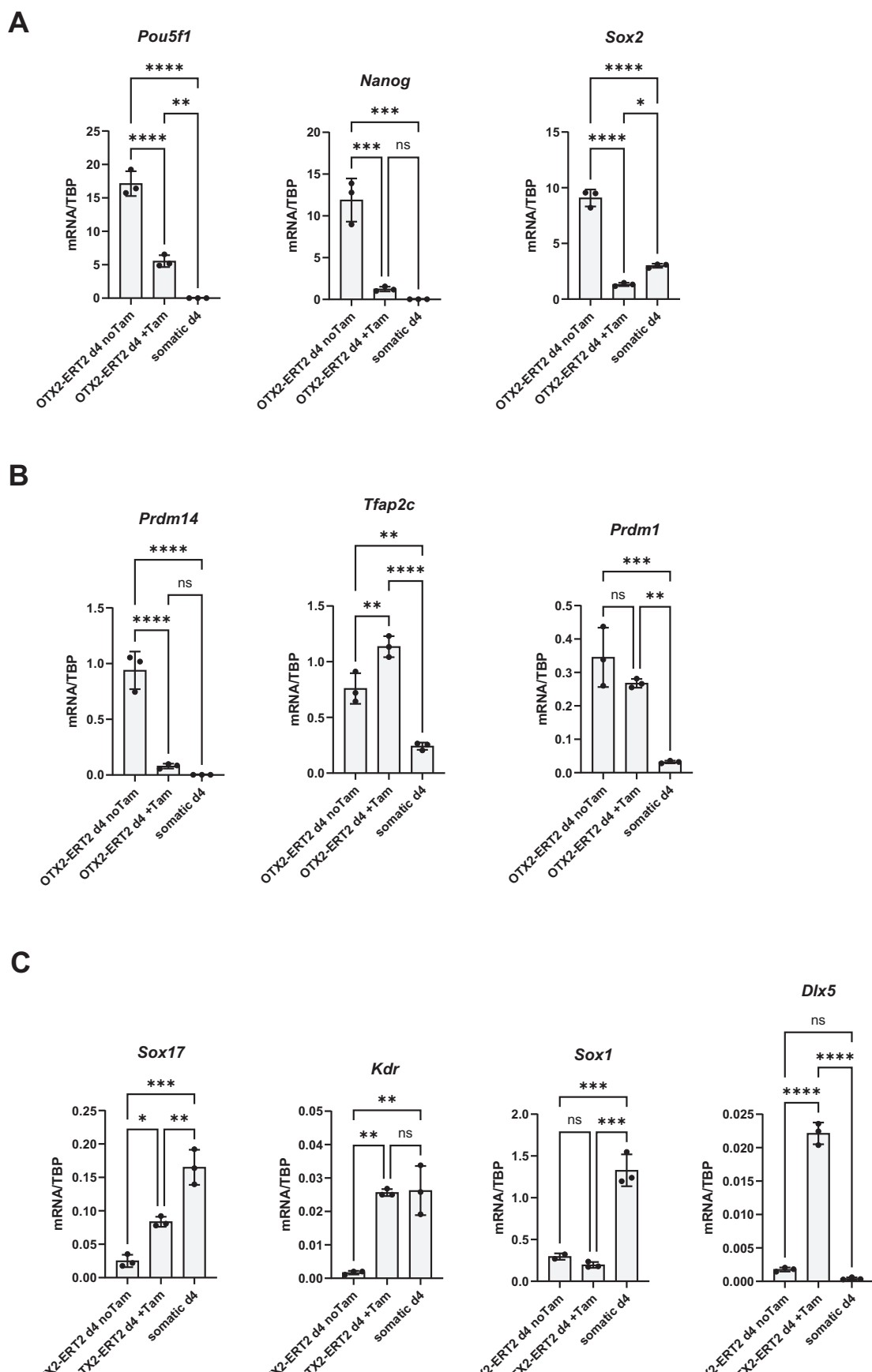

◄

**Figure EV5.   Expression of marker transcripts during differentiation.**

mRNA levels of pluripotency-associated markers (**A**), PGC-specific TF (**B**) and somatic-associated markers (**C**) in tamoxifen-treated and untreated *Otx2*$^{-/-}$::Otx2-ER$^{T2}$ cells differentiated for 2 and 4 days in GK15 in the presence of PGC-inducing cytokines or in somatic cells differentiated in GK15 without PGC-inducing cytokines. mRNA levels were quantified by RT-qPCR and normalised to TBP mRNA levels. Data are from a representative of 2 independent experiments (centre: mean, data points: technical triplicates, error bars: standard deviation). The strength of statistical significance was defined using the following thresholding system: ns (not significant) = $P \geq 0.05$; * = $0.01 < P < 0.05$; ** = $0.001 < P < 0.01$; *** = $0.0001 < P < 0.001$; ****$P < 0.0001$. Statistical comparison for *Pou5f1*: OTX2-ERT2 d4 no Tam vs OTX2-ERT2 d4 +Tam $P < 0.0001$; OTX2-ERT2 d4 no Tam vs somatic d4 $P < 0.0001$; OTX2-ERT2 d4 +Tam vs somatic d4 $P = 0.0031$. Statistical comparison for *Nanog*: OTX2-ERT2 d4 no Tam vs OTX2-ERT2 d4 +Tam $P = 0.0003$; OTX2-ERT2 d4 no Tam vs somatic d4 $P = 0.0002$; OTX2-ERT2 d4 +Tam vs somatic d4 $P = 0.6053$. Statistical comparison for *Sox2*: OTX2-ERT2 d4 no Tam vs OTX2-ERT2 d4 +Tam $P < 0.0001$; OTX2-ERT2 d4 no Tam vs somatic d4 $P < 0.0001$; OTX2-ERT2 d4 +Tam vs somatic d4 $P = 0.0102$. Statistical comparison for *Prdm14*: OTX2-ERT2 d4 no Tam vs OTX2-ERT2 d4 +Tam $P < 0.0001$; OTX2-ERT2 d4 no Tam vs somatic d4 $P < 0.0001$; OTX2-ERT2 d4 +Tam vs somatic d4 $P = 0.6214$. Statistical comparison for *Tfap2c*: OTX2-ERT2 d4 no Tam vs OTX2-ERT2 d4 +Tam $P = 0.0080$; OTX2-ERT2 d4 no Tam vs somatic d4 $P = 0.0016$; OTX2-ERT2 d4 +Tam vs somatic d4 $P < 0.0001$. Statistical comparison for *Blimp1*: OTX2-ERT2 d4 no Tam vs OTX2-ERT2 d4 +Tam $P = 0.2377$; OTX2-ERT2 d4 no Tam vs somatic d4 $P = 0.0008$; OTX2-ERT2 d4 +Tam vs somatic d4 $P = 0.0035$. Statistical comparison for *Sox17*: OTX2-ERT2 d4 no Tam vs OTX2-ERT2 d4 +Tam $P = 0.0118$; OTX2-ERT2 d4 no Tam vs somatic d4 $P = 0.0001$; OTX2-ERT2 d4 +Tam vs somatic d4 $P = 0.0023$. Statistical comparison for *Kdr*: OTX2-ERT2 d4 no Tam vs OTX2-ERT2 d4 +Tam $P = 0.0012$; OTX2-ERT2 d4 no Tam vs somatic d4 $P = 0.0010$; OTX2-ERT2 d4 +Tam vs somatic d4 $P = 0.9831$. Statistical comparison for *Sox1*: OTX2-ERT2 d4 no Tam vs OTX2-ERT2 d4 +Tam $P = 0.6755$; OTX2-ERT2 d4 no Tam vs somatic d4 $P = 0.0006$; OTX2-ERT2 d4 +Tam vs somatic d4 $P = 0.0002$. Statistical comparison for *Dlx5*: OTX2-ERT2 d4 no Tam vs OTX2-ERT2 d4 +Tam $P < 0.0001$; OTX2-ERT2 d4 no Tam vs somatic d4 $P = 0.2643$; OTX2-ERT2 d4 +Tam vs somatic d4 $P < 0.0001$.

**A**

PGCLCs vs somatic

22,499 somatic DARs

24,864 PGCLC DARs

*Padj*<0.05 and FC>2

$-\mathrm{Log}_{10}$ *Padj*

Log$_2$ Fold Change

**B**

*Otx2$^{-/-}$*::Otx2-ER$^{\text{T2}}$ d2 PGCLCs
OTX2 CUT&RUN

no Tam | Tam d0-d2

EpiLC OTX2 peaks

PGCLC DARs

-1Kb  center  1Kb   -1Kb  center  1Kb

**C**

PGCLC DARs (ATAC-seq)

ESC and EpiLC Accessible regions (ATAC-seq)

2,617 | 22,247

**D**

2,617 PGCLC-specific accessible regions

ATAC-seq

— WT ESCs
— *Otx2$^{-/-}$* ESCs
— WT EpiLCs
— *Otx2$^{-/-}$* EpiLCs
— somatic
— PGCLCs

-1Kb  center  1Kb

**E**

2,617 PGCLC-specific accessible regions

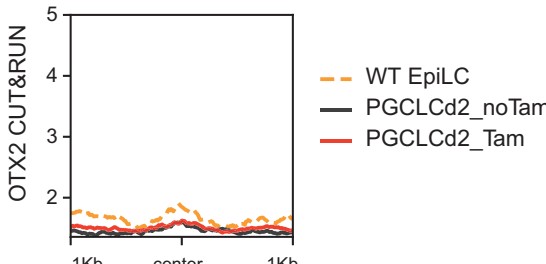

OTX2 CUT&RUN

- - WT EpiLC
— PGCLCd2_noTam
— PGCLCd2_Tam

-1Kb  center  1Kb

**Figure EV6.  OTX2 does not control accessibility of PGCLC-specific accessible regions.**

(A) Volcano plot comparing differentially accessible regions in somatic cells and PGCLCs. Analysis performed and plot generated by DESeq2 using Wald test with Benjamini−Hochberg correction for multiple testing from $n = 2$ biological replicates per condition. (B) Heatmap of OTX2 CUT&RUN signal in tamoxifen-treated *Otx2$^{-/-}$*::Otx2-ER$^{\text{T2}}$ d2 PGCLCs at EpiLC OTX2-bound regions and PGCLC DARs. (C) Venn diagram of the overlap of PGCLC DARs (orange) and ESC+EpiLC accessible regions identifying 2617 PGCLC-specific accessible regions. (D) Average read density profiles of ATAC-seq signal in wild-type and *Otx2$^{-/-}$* ESCs, wild-type and *Otx2$^{-/-}$* EpiLCs, PGCLCs and somatic cells at 2617 PGCLC-specific regions. (E) Average read density profiles of OTX2 CUT&RUN in wild-type EpiLC and tamoxifen-treated *Otx2$^{-/-}$*::Otx2-ER$^{\text{T2}}$ d2 PGCLCs at PGCLC-specific regions.

