## [Peer Review File · EMBO Reports]

OTX2 controls chromatin accessibility to direct somatic versus germline differentiation

Elisa Barbieri and Ian Chambers

Corresponding author(s): Elisa Barbieri (elisa.barbieri@ed.ac.uk)

Review Timeline:

Transfer Date:	12th May 25
Editorial Decision:	22nd May 25
Revision Received:	1st Sep 25
Editorial Decision:	9th Oct 25
Revision Received:	17th Oct 25
Accepted:	21st Oct 25

Editor: Esther Schnapp

Transaction Report: This manuscript was transferred to EMBO reports following peer review at Review Commons.

Review
COMMONS

Review #1

1. Evidence, reproducibility and clarity:

Evidence, reproducibility and clarity (Required)

Summary

OTX2 is a pivotal transcription factor that regulates the fate choice between somatic and primordial germ cell (PGC) lineages in early mouse development. In the current study, the authors use in vitro stem cell models to demonstrate that OTX2 mediates this developmental fate decision through controlling chromatin accessibility, whereby OTX2 helps to activate putative enhancers that are associated with somatic fate. By extension, those somatic-associated regulatory regions therefore become inaccessible in cells adopting PGC identity in which Otx2 is downregulated.

Comments

I enjoyed reading this manuscript. The experiments have been carried out well and for the most part the results provide convincing evidence to support the claims and conclusions in the manuscript. I particularly liked the experiments using the inducible Otx2 transgene to examine the acute changes in chromatin accessibility following restoration of OTX2.

I include some suggestions below to the authors for additional analyses that I feel would further strengthen their study.

I also felt that the authors focus almost exclusively on the subset of OTX2-bound sites that lose accessibility in the absence of OTX2. But, as they show in several figure panels, these sites tend to be the minority and that most OTX2-occupied sites do not lose accessibility in Otx2-null cells (actually, more sites tend to gain accessibility). I encourage the authors to modify the text and some of the analyses to give a better balance to their study.

1. Figure 1: The authors report in the methods that they performed OTX2 CUT&RUN in biological duplicates. It would strengthen their results if they showed in Figure S1 some representative data from each replicate separately to show the consistency.
2. Figure 1: The authors write: "...OTX2 binds mostly to putative enhancers." Whether these distal sites are enhancers is not sufficiently evidenced in the manuscript, but it is important information to collect to support their model of OTX2 function. The authors

should strengthen their analysis by examining whether OTX2 peaks are enriched at previously defined enhancer regions.

3. Figure 2: I think it would be helpful to remind the reader here that Otx2 is normally downregulated in PGCs, and that Otx2 expression is maintained (at least initially) in somatic cells. This would help explain the logic behind the choice of samples that were profiled. That said, I'm still puzzled why the authors did not examine flow-sorted WT+cyto cells?

4. Figure 2D: I appreciate that the highlighted region at the Tet2 locus is a DAR, but from the genome tracks it looks as though the region still has high accessibility. Are there any other examples to exemplify a more obvious DAR? Additionally, since twice as many DARs gain accessibility in Otx2-null ESCs compared to lose accessibility, why not show examples of these as well? The same is true of EpiLCs. (Or alternatively, provide a good explanation for why not to show these other categories)

5. Figure 2: the authors write: "This suggests that OTX2 acts as a pioneer TF...". However, at this point in the manuscript, there is no evidence to support that OTX2 might have pioneer activity. I think this claim would be better suited to later in the manuscript, or in the discussion, following the finding that reintroduction of OTX2 can induce chromatin accessibility at previously closed sites.

6. Figure 3: I would be tempted to put Figure S3A and S3B into Figure 3. It would be better to show all 1246 DARs together, either ordered by OTX2 CT&RUN signal, or presented in two pre-defined groups (OTX2-bound vs unbound). I also suggest that the author show OTX2 signals and ATAC-seq signals for the 3028 DARs that gain accessibility in Otx2-null EpiLCs (this could be added to a supplemental figure).

7. Figure 3: What is special about the 8% of OTX2-bound site that lose accessibility, versus the 92% of sites that do not?

8. Figure 6: Do the PGCLCs with OTX2 expression have chromatin accessibility profiles similar to somatic cells? Consider adding WT somatic cell data to Figure 6A, which could be an interesting comparison with the Tam d0-d2 samples.

9. Figure 6F: If the 4221 sites are split into those bound by OTX2 versus those that are not (related to Figure 6C) then is there a difference? i.e. are the OTX2-bound sites opening up?

10. Is there any evidence that OTX2 binds and compacts PGCLC enhancers in somatic cells? I appreciate this is different to the main thrust of the authors' model, but being able to show that OTX2 does not compact these sites lends further support to their preferred model of OTX2 opening sites of somatic lineages.

11. Discussion: Have prior studies established a connection between OTX2 and chromatin remodellers that can open chromatin? Or, if not, then perhaps this could be proposed as a line of future research.

2. Significance:

Significance (Required)

Strengths

The results presented provide a careful dissection of the role of OTX2 in controlling chromatin accessibility in different stages of pluripotent to somatic and PGC fates. The authors do a good job of revealing the stage-specific differences in OTX2 occupancy and chromatin accessibility as well as the different responses following the acute reintroduction of OTX2.

Limitations

I felt that the authors could present/discuss a bit more on alternative possibilities and models, as it would help the reader to better understand why they favour one model over other ones, and presenting these other possibilities could also provide more support for their preferred model.

Whether OTX2 is binding to putative enhancers is inferred but could be evidenced more strongly, as that is important for their model.

Advance

This study provides key information to understand the mechanisms of OTX2 function in cell fate choice. Similar functions have been shown in other contexts for other transcription factors, but this is a nicely done study and adds to our understanding of how transcription factors function in early development to direct cell-fate decisions.

Expertise

My field of expertise lies in the gene regulatory control of early developmental decisions.

3. How much time do you estimate the authors will need to complete the suggested revisions:

Estimated time to Complete Revisions (Required)

(Decision Recommendation)

Between 1 and 3 months

4. Review Commons values the work of reviewers and encourages them to get credit for their work. Select 'Yes' below to register your reviewing activity at Web of Science Reviewer Recognition Service (formerly Publons); note that the content of your review will not be visible on Web of Science.

No

Review #2

1. Evidence, reproducibility and clarity:

Evidence, reproducibility and clarity (Required)

Barbieri and Chambers explore the role of OTX2 on mouse pluripotency and differentiation. To do so, they examine how the chromatin accessibility and OTX2 binding landscape changes across pluripotency, the exit of pluripotency towards formative and primed states, and through to PGCLC/somatic differentiation. The work mostly represents a resource for the community, with possible implications for our understanding of how OTX2 might mediate the germline-soma switch of fates. While the findings of the work are modest, the results seem solid and the manuscript is clear and well-written. I have some comments as indicated below:

- The comparison between Otx2^{-/-} cells in the presence of PGCLC cytokines compared to WT cells in the absence of cytokines seems like it is missing controls to me. I assume the authors wanted to enable homogeneous populations to facilitate their bulk sequencing methods, but it seems to me like they are comparing apples with oranges. It would have been better to have the reciprocal situations (Otx2^{-/-} cells in basal differentiation medium, and WT cells in PGCLC cytokines) with a sorting strategy to better unpick the differences between the presence and absence of Otx2 in the 2 protocols. Having said that, the authors are careful not to draw many comparisons between those populations so I don't think this omission affects their current claims. They should however clarify whether the flow cytometry (Supp Fig2) was used for sorting cells or if all cells were taken for bulk sequencing.
- The authors focus solely on the activating role of Otx2 in their data, but given the substantial proportion of DARs that decrease following Otx2 depletion, I presume it is possible that it also has a repressive effect? Either way, this should be discussed.

- Throughout the text, the authors subject cells (WT / Otx2^{-/-} /Otx2ER) to different protocols to look at accessibility and Otx2 binding, but with no mention of the cell fate differences that occur in these different conditions. For instance, it is unclear to me to which fate the WT cells without PGCLC cytokines go - I presume this is neural but perhaps this is a mixed fate, given that they are in GK15 rather than N2B27. Likewise, the OTX2ER experiments may promote a mixed population between PGCLC/somatic fates, and this is never described. Ideally transcriptomic data would be collected, but failing that, qPCR data should be obtained to examine this more closely.
- The authors also state that "OTX2 facilitates Fgf5 transcription' (page10) but provide no transcriptional data to substantiate this claim. Again RT-qPCR would help make this point.
- The authors state that d2 PGCLCs "show an intermediate position between ESCs and EpiLCs" based on the PCA location. They should be careful to qualify that this is only in the first 2 principal components, because it may well be the case (and is likely) that in other components the PGCLC population is far removed from the pluripotent states.
- It is unclear to me what the 'increase[d] accessibility' (eg abstract final sentence, Figure 3E) really means at the cellular level. Does this indicate that more cells have this site open, and does this have implications for the heterogeneity of cell fates observed? Since the authors are concerned with fate decisions, this seems like an important consideration that should at least be discussed.

****Minor Suggestions:****

- Presumably the regions bound by OTX2 in Tet2, Mycn and Fgf5 (Fig1E) are called enhancers because these are known from existing literature. It would be helpful to cite the relevant references to this in the text for those unfamiliar with these.
- On page 13, the authors say "To determine whether OTX2 expression is essential to maintain chromatin accessibility in somatic cells..." but this does not seem to be what they test because they are using PGCLC medium. Perhaps I misunderstood, but this could be clarified.
- On page 14 the authors claim, "These results indicate that...the partner proteins that OTX2 act alongside differ...". While this may be the case, their results do not substantiate this, it is just speculation. Should be toned down.
- Page 18, PGCLC differentiation method sections needs to be described as such (ie. Add "For PGCLC differentiation..." before the second paragraph)
- It would be helpful to indicate time on the protocol schematics (eg Fig4A, 5A, 5D etc) as I had to keep checking the methods to find out how long the full differentiation time-course was.
- Since the authors compare between the Tam d0-d2 treatments assessed at d2 versus d4

(Figure5B vs 5E) it would be helpful to make the colourbars the same scale, for both ATAC and Cut&Run datasets.

2. Significance:

Significance (Required)

The study examines the binding of OTX2 and subsequent chromatin accessibility in pluripotent, primed and differentiated (PGCLC/Somatic) cell states, including through Otx2^{-/-} cell lines and temporally-controlled exogenous expression of Otx2. As such, it represents a valuable resource into the potential direct targets of Otx2 and their change in accessibility state across cell types. The work is likely to be of interest to those working on understanding the exit of pluripotency, gene regulatory networks, and chromatin remodelling. My expertise is in cell fate decisions, pluripotency regulation and PGC(LC) differentiation.

3. How much time do you estimate the authors will need to complete the suggested revisions:

Estimated time to Complete Revisions (Required)

(Decision Recommendation)

Between 1 and 3 months

Yes

Review #3

1. Evidence, reproducibility and clarity:

Evidence, reproducibility and clarity (Required)

****Summary:****

In this manuscript, the authors perform OTX2 CUT&RUN and ATAC-seq in Otx2-null and WT ESCs, EpiLCs and PGCLCs to understand whether the role of OTX2 in restricting mouse germline entry that they previously described (Zhang Nature 2018) mechanistically depends on chromatin remodeling. They identify differentially accessible regions (DARs) between Otx2-null and WT cells at different stages of differentiation and show that many of these are OTX2 bound in WT. They then show using cells expressing OTX2-ER^{T2} in Otx2-null Epiblast cells that when OTX2 is moved into the nucleus, the regions that were differentially closed in Otx2-null open within an hour, suggesting chromatin accessibility is directly controlled by OTX2 (rather than indirect effects involving transcription and translation which one would expect to take longer). The scope is narrow, but this is nice work and useful data for the mouse PGC field. However, there are a few places where the data could be strengthened, and the writing is a little confusing in places, for example by stating as fact in early sections what is not proven until later.

****Major Comments:****

1. "we compared Otx2^{-/-} cells cultured in the presence of PGC-promoting cytokines with wild-type cells cultured in the absence of PGC-promoting cytokines. Under these conditions Otx2^{-/-} cells produce an essentially pure (>90%) CD61⁺/SSEA1⁺ population that we refer to as PGCLCs, while wild-type cells yield a cell population from which PGCLCs are absent"

This is not a controlled comparison since one cannot separate the day 2 effect of cytokines from that of the Otx2 knockout. The manuscript would be strengthened if the authors include WT somatic and PGCLCs from the +cytokine conditions, which could be easily sorted out as shown in Supp. Fig. 2. Ideally they would also include Otx2-null somatic cells, although Supp. Fig. 2 shows those are rare under the conditions considered.

As a minor point related to this, the second sentence is confusing since it kind of sounds like Otx2^{-/-} and WT cells are compared under the same conditions unless one carefully reads the previous sentence.

2. "This suggests that OTX2 acts as a pioneer TF to regulate the accessibility of enhancers E1, E2 and E3."

This is from the text corresponding to Fig. 2. That data actually only shows that Otx2-null cells have DARs, so somehow OTX2 affects chromatin accessibility but it could be indirect by controlling transcription of genes that modify chromatin accessibility. It is not until figure 4 that the data suggests that OTX2 directly affects accessibility, perhaps as a pioneer TF.

The authors continue to make many statements about the direct action of OTX2 before the data supporting this is shown, on which I got hung up as a reader. I suggest the authors edit the manuscript to improve this. E.g. "OTX2 may directly control accessibility at these sites (Figure 3E)." and the fact that in 3E and other figure, it says "DARs increased by OTX2 binding" which at that point is not proven, so would better say "Otx2-null vs WT DARs" or something like that.

3. "In ESCs, OTX2 binds <10% (30 out of 375) of DARs that are more accessible in wild-type cells than in Otx2-/- cells (Figure 3A), suggesting that accessibility of ESC DARs is directly due to OTX2 in a small subset of DARs."

When a small number of DARs are OTX2 bound, it does not necessarily suggest that that small set is directly affected by OTX2. It could just mean no DARs are controlled directly by OTX2 and then some are bound by chance by OTX2. Some appropriate statistical null hypotheses about the occurrence of OTX2 motifs might help to see if 10% is more than chance.

4. It would be good if the discussion was broadened to include both human and other transcription factors that are involved. How much of these conclusions could one expect to carry over to human or other mammals? There is some work from the Surani lab considering OTX2 in human. One could even look at published ATAC or OTX2 chip-seq data in hPSCs and potentially learn something interesting. Furthermore, there are studies on other transcription factors modulating chromatin accessibility in the decision between germline and somatic cells, for example PRDM1, PRDM14 (refs in e.g. Tang et al Nat Rev Gen 2016) or TFAP2A (at least in human (Chen et al Cell Rep 2019)). Do these factors affect the same genes? Is a coherent picture emerging of their respective roles in germline entry?

****Minor comments:****

1. "Comparing wild-type and Otx2-/- ESCs identified 375 differentially accessible regions (DARs) with increased accessibility in wild-type cells, and 743 regions with higher accessibility in Otx2-/- ESCs (Figures 2C). An example of ESC DARs where accessibility is increased in cells expressing OTX2 is the intragenic enhancer of Tet2. Tet2 is expressed at high levels in ESCs but at low levels in EpiLCs."

The authors compare Otx2-null and WT ESCs then proceed to give an example comparing ESCs to EpiLCs, instead of Otx2-null vs WT ESCs, which is confusing.

Furthermore, here and in other places the authors describe ESCs as not expressing OTX2. However, they also show CUT&RUN data for OTX2 in ESCs etc, clearly indicating that it is

expressed, just lower (otherwise how could one get anything?).

2. "In contrast, in EpiLCs, OTX2 binds almost 40% (446 out of 1,246) of the DARs that are more accessible in wild-type than in *Otx2*^{-/-} cells (Figure 3B-C). Notably, these regions are mainly located distal to genes (91%, Figure 3D), despite the increased fraction of promoter regions bound by OTX2 in EpiLCs (Figure S1A)."

Are the authors rounding percentages with 2 significant digits, as suggested by the "91%"? If so, 446/1245 ~ 36%, not 40%.

3. The results in Figure 4 are nice and the real meat of the paper.

One suggestion: It would be helpful if Fig. 4B were split up between the 446 and 800 genes instead of showing all 1246, and if the WT control was shown in the same figure as well.

4. "Enforced OTX2 expression opens additional somatic regulatory regions" - it would be clearer to say "OTX2 overexpression opens additional somatic regulatory regions", since this is really about DARs between EpiLCs that already express OTX2 and those forced to express higher than WT endogenous levels by the OTX2-ER system?

2. Significance:

Significance (Required)

Also see summary. Understanding what restricts cells to germline vs somatic lineages is an important question. By providing functional data showing that OTX2 directly controls chromatin accessibility, the authors add an important layer of understanding to their previous finding that OTX2 plays a key role in preventing mouse germline entry. The use of their previously established OTX2-null cells expressing OTX2-ER to rapidly induce nuclear OTX2 in a mutant background or the most part makes their experiments elegant and convincing. In focusing on the role of one gene in one event in one species, it is specialized and narrow in scope and will mostly be of interest to experts in the field, but there is nothing wrong with that.

3. How much time do you estimate the authors will need to complete the suggested revisions:

Estimated time to Complete Revisions (Required)

(Decision Recommendation)

Between 1 and 3 months

4. Review Commons values the work of reviewers and encourages them to get credit for their work. Select 'Yes' below to register your reviewing activity at Web of Science

Reviewer Recognition Service (formerly Publons); note that the content of your review will not be visible on Web of Science.

No

Revision Plan

Manuscript number: RC-2025-02945

Corresponding author(s): Ian, Chambers

1. General Statements [optional]

We thank the reviewers for finding the manuscript enjoyable and well-written, with experiments that were performed well, show solid results and provide useful data for the community. The reviewers have provided meaningful feedback to improve this study. We have addressed the comments point-by-point below. The main text will also be further modified to incorporate new analysis where it has not yet been done.

2. Description of the planned revisions

Reviewer 1:

Summary

OTX2 is a pivotal transcription factor that regulates the fate choice between somatic and primordial germ cell (PGC) lineages in early mouse development. In the current study, the authors use in vitro stem cell models to demonstrate that OTX2 mediates this developmental fate decision through controlling chromatin accessibility, whereby OTX2 helps to activate putative enhancers that are associated with somatic fate. By extension, those somatic-associated regulatory regions therefore become inaccessible in cells adopting PGC identity in which *Otx2* is downregulated.

Comments

I enjoyed reading this manuscript. The experiments have been carried out well and for the most part the results provide convincing evidence to support the claims and conclusions in the manuscript. I particularly liked the experiments using the inducible *Otx2* transgene to examine the acute changes in chromatin accessibility following restoration of OTX2.

I include some suggestions below to the authors for additional analyses that I feel would further strengthen their study.

I also felt that the authors focus almost exclusively on the subset of OTX2-bound sites that lose accessibility in the absence of OTX2. But, as they show in several figure panels, these sites tend to be the minority and that most OTX2-occupied sites do not lose accessibility in *Otx2*-null cells (actually, more sites tend to gain accessibility). I encourage the authors to modify the text and some of the analyses to give a better balance to their study.

We are pleased that this reviewer enjoyed our manuscript. As suggested by the reviewer, we

Revision Plan

included analyses on the regions that are bound by OTX2 but do not show an increase in accessibility (see section 3 reviewer 1 point 6). The text will be expanded to include the new data and to include the description of the subset of OTX2 sites that do not show accessibility changes in the absence of OTX2. We have responded to other points they raised as detailed in the sections below

2. Figure 1: The authors write: "...OTX2 binds mostly to putative enhancers." Whether these distal sites are enhancers is not sufficiently evidenced in the manuscript, but it is important information to collect to support their model of OTX2 function. The authors should strengthen their analysis by examining whether OTX2 peaks are enriched at previously defined enhancer regions.

We plan to compare OTX2 bound regions with defined lists of enhancers identified in ESCs grown in Serum/LIF (e.g. Whyte et al 2013) and, if available, in 2i/LIF and EpiLCs. We will also analyse publicly available datasets for H3K4me1 (enhancer marker) and H3K27ac (marker of active regulatory regions) at the regions bound by OTX2 in ESCs and EpiLCs.

3. Figure 2: I'm still puzzled why the authors did not examine flow-sorted WT+cyto cells?

We agree with the reviewer that it would be interesting to examine flow-sorted WT +cyto PGCLCs. Unfortunately, the expression of CD61 and SSEA1 only becomes visible from day 4 of PGCLC differentiation. Therefore, we were not able to isolate PGCLC at day 2 from WT cells differentiated in the presence of cytokines. We then used OTX2^{-/-} cells at day 2 to model PGCLCs. This is based on the assumption that because day 6 Otx2^{-/-} PGCLCs are transcriptionally similar to sorted day 6 WT cells (Zhang, Zhang et al Nature 2018), the same will be true at day 2. We will modify the text in the final version of this manuscript to clarify this point that has also been raised by reviewers 2 and 3.

6. Figure 3: I would be tempted to put Figure S3A and S3B into Figure 3. It would be better to show all 1246 DARs together, either ordered by OTX2 CT&RUN signal, or presented in two pre-defined groups (OTX2-bound vs unbound). I also suggest that the author show OTX2 signals and ATAC-seq signals for the 3028 DARs that gain accessibility in Otx2-null EpiLCs (this could be added to a supplemental figure).

Although the analysis has been carried out and the figures have been amended, the main text will be modified in a future updated version of the manuscript to incorporate these results.

7. Figure 3: What is special about the 8% of OTX2-bound site that lose accessibility, versus the 92% of sites that do not?

The 8% of the OTX2-bound regions that lose accessibility in the absence of OTX2 appear to be more sensitive to the loss of OTX2. One possible explanation is that the accessibility of the rest of OTX2 bound regions relies on other TFs, such as OCT4, that are expressed in EpiLCs. We will modify the main text to discuss this interesting point raised by the reviewer.

Revision Plan

9. Figure 6F: If the 4221 sites are split into those bound by OTX2 versus those that are not (related to Figure 6C) then is there a difference? i.e. are the OTX2-bound sites opening up?

We separated the 4,221 sites in OTX2 bound and unbound. The result is reported below:

Although there is a slight increase in accessibility in the OTX2 bound subset, the average accessibility reaches less than $\frac{1}{4}$ of the accessibility of these regions when OTX2 is present from day 0 to day 4, while the OTX2 unbound regions do not show an increase in accessibility. Although we can not rule out that a longer treatment with tamoxifen may lead to higher accessibility in the OTX2 bound subset, the dynamics are extremely slower compared to the EpiLC regions where accessibility reaches 50% of the d0-d4 sample in just 1 hour of tamoxifen treatment.

10. Is there any evidence that OTX2 binds and compacts PGCLC enhancers in somatic cells? I appreciate this is different to the main thrust of the authors' model, but being able to show that OTX2 does not compact these sites lends further support to their preferred model of OTX2 opening sites of somatic lineages.

Comparing the ATAC-seq in PGCLCs with ESCs and EpiLCs, we identify a subset of regions that are open in PGCLC only (PGCLC-specific accessible regions, see below). These regions do not show binding of OTX2 in WT EpiLCs or the d0-d2 Tam sample, suggesting that OTX2 does not bind and compact PGCLC-specific enhancers.

PGCLC-specific regions showing high accessibility only in PGCLCs.

PGCLC-specific regions showing high accessibility

Revision Plan

PGCLC-specific accessible regions

OTX2 CUT&RUN signal in WT EpiLC, OTX2-ERT2 PGCLCs in presence or absence of Tamoxifen, showing that OTX2 does not bind PGCLC-specific regions even when it is overexpressed in GK15 medium. These analyses will be incorporated in the manuscript.

11. Discussion: Have prior studies established a connection between OTX2 and chromatin remodellers that can open chromatin? Or, if not, then perhaps this could be proposed as a line of future research.

We thank the reviewer for suggesting to amplify the discussion on the possible connection between OTX2 and chromatin remodellers. Although there is no evidence in the literature of a direct interaction between OTX2 and chromatin remodellers, this can not be excluded. The connection might also be indirect: OTX2 is known to interact with OCT4, which in turn interacts and recruits to chromatin the catalytic subunit of the SWI/SNF complex, BRG1. This point will be discussed in a modified version of the manuscript.

Reviewer 2:

Barbieri and Chambers explore the role of OTX2 on mouse pluripotency and differentiation. To do so, they examine how the chromatin accessibility and OTX2 binding landscape changes across pluripotency, the exit of pluripotency towards formative and primed states, and through to PGCLC/somatic differentiation. The work mostly represents a resource for the community, with possible implications for our understanding of how OTX2 might mediate the germline-soma switch of fates. While the findings of the work are modest, the results seem solid and the manuscript is clear and well-written.

We are pleased that this reviewer found our results solid and the manuscript clear.

I have some comments as indicated below:

1. The comparison between *Otx2*^{-/-} cells in the presence of PGCLC cytokines compared to WT cells in the absence of cytokines seems like it is missing controls to me. I assume the authors wanted to enable homogeneous populations to facilitate their bulk sequencing methods, but it seems to me like they are comparing apples with oranges. It would have been better to have the reciprocal situations (*Otx2*^{-/-} cells in basal differentiation medium, and WT cells in PGCLC cytokines) with a sorting strategy to better unpick the differences between the presence and absence of *Otx2* in the 2 protocols. Having said that, the authors are careful not to draw many comparisons between those populations so I don't think this omission affects their current

Revision Plan

claims. They should however clarify whether the flow cytometry (Supp Fig2) was used for sorting cells or if all cells were taken for bulk sequencing.

We agree with the reviewer that it would be of interest to compare the PGCLC and somatic population derived from the OTX2^{-/-} cells in GK15 without cytokines with the same populations derived from WT cells differentiated in the presence of cytokines. Our work aims to identify what happens at the stages of PGCLC differentiation when cells are still competent for both germline and somatic differentiation. Previous work from the lab showed that this dual competence is lost after day 2, therefore we focus our attention on this time of differentiation. Unfortunately, the two surface markers characteristics of PGCs (CD61 and SSEA1) are not expressed at day2 and, therefore we are not able to sort PGCLCs derived from OTX2^{-/-} cells in GK15 without cytokines or WT cells differentiated in the presence of cytokines. As recognised by this reviewer, we aimed to obtain two homogenous populations that can model PGCLCs and somatic cells. This is based on data obtained at day 6 when Otx2^{-/-} PGCLCs show a similar transcriptome to sorted day 6 WT cells (Zhang, Zhang et al Nature 2018) and the assumption that the same will be true at day 2. We will clarify that the supplementary Figure 2 is not a sorting strategy. As this point has been raised by reviewers 1 and 2 as well, we will modify the text to clarify the choice and the assumption behind using OTX2^{-/-} cells in the presence of cytokines and WT cells in the absence of cytokines to model PGCLCs and somatic cells respectively.

3. Throughout the text, the authors subject cells (WT / Otx2^{-/-} /Otx2ER) to different protocols to look at accessibility and Otx2 binding, but with no mention of the cell fate differences that occur in these different conditions. For instance, it is unclear to me to which fate the WT cells without PGCLC cytokines go - I presume this is neural but perhaps this is a mixed fate, given that they are in GK15 rather than N2B27. Likewise, the OTX2ER experiments may promote a mixed population between PGCLC/somatic fates, and this is never described. Ideally transcriptomic data would be collected, but failing that, qPCR data should be obtained to examine this more closely.

We are planning to generate RT-qPCR data for germ layer markers (ectoderm, endoderm and mesoderm) in WT cells in GK15 without cytokines at day 2, as well as OTX2-ERT2 cells with and without Tamoxifen at day 2 (noTam, d0-d2) and day 4 (no Tam, d0-d4).

4. The authors also state that "OTX2 facilitates Fgf5 transcription" (page10) but provide no transcriptional data to substantiate this claim. Again RT-qPCR would help make this point.

We will analyse the level of Fgf5 by RT-qPCR in OTX2-ERT2 EpiLCs treated for 1 hour and 6 hours with Tamoxifen to show the effect of OTX2 on Fgf5 transcription.

6. It is unclear to me what the 'increase[d] accessibility' (eg abstract final sentence, Figure 3E) really means at the cellular level. Does this indicate that more cells have this site open, and does this have implications for the heterogeneity of cell fates observed? Since the authors are concerned with fate decisions, this seems like an important consideration that should at least be discussed.

Revision Plan

The possibility that the increased accessibility is due to higher heterogeneity in the population is interesting and it will be included in the discussion in a revised version of the manuscript.

Reviewer 3:

In this manuscript, the authors perform OTX2 CUT&RUN and ATAC-seq in Otx2-null and WT ESCs, EpiLCs and PGCLCs to understand whether the role of OTX2 in restricting mouse germline entry that they previously described (Zhang Nature 2018) mechanistically depends on chromatin remodeling. They identify differentially accessible regions (DARs) between Otx2-null and WT cells at different stages of differentiation and show that many of these are OTX2 bound in WT. They then show using cells expressing OTX2-ER^{T2} in Otx2-null Epiblast cells that when OTX2 is moved into the nucleus, the regions that were differentially closed in Otx2-null open within an hour, suggesting chromatin accessibility is directly controlled by OTX2 (rather than indirect effects involving transcription and translation which one would expect to take longer). The scope is narrow, but this is nice work and useful data for the mouse PGC field. However, there are a few places where the data could be strengthened, and the writing is a little confusing in places, for example by stating as fact in early sections what is not proven until later.

We thank the reviewer for finding our work nice and useful for the mouse PGC field, and for the useful comments to improve the manuscript. We have included new analysis and modified the text as suggested to improve the writing, avoiding early statements that were not fully proven until later in the manuscript. We have responded to other points they raised as detailed below and in the next section.

1) "we compared Otx2^{-/-} cells cultured in the presence of PGC-promoting cytokines with wild-type cells cultured in the absence of PGC-promoting cytokines. Under these conditions Otx2^{-/-} cells produce an essentially pure (>90%) CD61⁺/SSEA1⁺ population that we refer to as PGCLCs, while wild-type cells yield a cell population from which PGCLCs are absent"

This is not a controlled comparison since one cannot separate the day 2 effect of cytokines from that of the Otx2 knockout. The manuscript would be strengthened if the authors include WT somatic and PGCLCs from the +cytokine conditions, which could be easily sorted out as shown in Supp. Fig. 2. Ideally they would also include Otx2-null somatic cells, although Supp. Fig. 2 shows those are rare under the conditions considered.

This work aimed to analyse early stages of EpiLC to PGCLC differentiation when cells are still competent for both somatic and germline differentiation. This stage has been described previously to be at day 2 of differentiation in GK15 + cytokines (PGCLC differentiation medium, Zhang, Zhang et al, Nature 2018). Unfortunately, CD61 and SSEA1 are not expressed at day 2 of PGCLC differentiation, and they start to be expressed on the cell surface by day 4. Consequently, it is impossible to sort cells at day 2 using the CD61⁺/SSEA1⁺ strategy. To overcome this problem, we used WT cells grown in GK15 without cytokines to model a population of somatic cells and OTX2^{-/-} cells grown in GK15+ cytokines to model a homogeneous population of PGCLCs. As explained in a similar point raised by reviewers 2 and

Revision Plan

3, we assumed that, as OTX2^{-/-} cells grown in the presence of cytokines are transcriptionally similar to sorted WT cells at day 6 (Zhang, Zhang et al, Nature 2018), OTX2^{-/-} cells at day 2 are similar to their WT counterpart at day 2. The main text will be modified to clarify that we are using homogeneous populations to model both PGCLC and somatic cells and that Figure S2 does not show a sorting strategy.

3) "In ESCs, OTX2 binds <10% (30 out of 375) of DARs that are more accessible in wild-type cells than in Otx2^{-/-} cells (Figure 3A), suggesting that accessibility of ESC DARs is directly due to OTX2 in a small subset of DARs."

When a small number of DARs are OTX2 bound, it does not necessarily suggest that that small set is directly affected by OTX2. It could just mean no DARs are controlled directly by OTX2 and then some are bound by chance by OTX2. Some appropriate statistical null hypotheses about the occurrence of OTX2 motifs might help to see if 10% is more than chance.

We are planning to perform a statistical analysis to ascertain that the small number of DARs bound by OTX2 are or are not bound by chance by OTX2.

4) It would be good if the discussion was broadened to include both human and other transcription factors that are involved. How much of these conclusions could one expect to carry over to human or other mammals? There is some work from the Surani lab considering OTX2 in human. One could even look at published ATAC or OTX2 chip-seq data in hPSCs and potentially learn something interesting. Furthermore, there are studies on other transcription factors modulating chromatin accessibility in the decision between germline and somatic cells, for example PRDM1, PRDM14 (refs in e.g. Tang et al Nat Rev Gen 2016) or TFAP2A (at least in human (Chen et al Cell Rep 2019)). Do these factors affect the same genes? Is a coherent picture emerging of their respective roles in germline entry?

As suggested by the reviewer, we will discuss the role of OTX2 in human PGCLC formation and include studies on PGC-specific transcription factors concerning changes in chromatin accessibility in germline and somatic cells. This will be included in a revised version of the manuscript.

3. Description of the revisions that have already been incorporated in the transferred manuscript

Reviewer 1:

1. Figure 1: The authors report in the methods that they performed OTX2 CUT&RUN in biological duplicates. It would strengthen their results if they showed in Figure S1 some representative data from each replicate separately to show the consistency.

Revision Plan

As suggested by the reviewer to show consistency between replicates, two representative tracks of the two CUT&RUN replicates at the Tet2 (ESCs) and Fgf5 (EpiLCs) loci have been included in Figure S1A. The corresponding tracks of the average bigwig files are reported in Figure 1E. The main text (page 5) and the figure legends have been amended to incorporate the new panels.

3. Figure 2: I think it would be helpful to remind the reader here that Otx2 is normally downregulated in PGCs, and that Otx2 expression is maintained (at least initially) in somatic cells. This would help explain the logic behind the choice of samples that were profiled.

We modified the text with the following sentence, as suggested by the reviewer, emphasising the level of OTX2 in early somatic vs early PGCLCs: “Otx2 expression is rapidly downregulated in the EpiLC to PGCLC transition while its expression is maintained longer in cells entering the somatic lineage [8]” (page 7).

4. Figure 2D: I appreciate that the highlighted region at the Tet2 locus is a DAR, but from the genome tracks it looks as though the region still has high accessibility. Are there any other examples to exemplify a more obvious DAR? Additionally, since twice as many DARs gain accessibility in Otx2-null ESCs compared to lose accessibility, why not show examples of these as well? The same is true of EpiLCs. (Or alternatively, provide a good explanation for why not to show these other categories)

We substituted the Tet2 DAR with a more clear example of ESC DAR located in the Hes1 locus that shows low accessibility in Otx2^{-/-} ESCs versus WT ESCs. Examples of ESC DARs and EpiLC DARs that show higher accessibility in Otx2^{-/-} vs WT cells have been added as new panels 2E (DAR in Pebp4 locus) and 2G (DAR in Tdh locus). We also simplified the panels showing only ATAC-seq tracks in WT and OTX2^{-/-} cells, either ESCs (2D-E) or EpiLCs (G-H). Text and figure legends have been modified to accommodate the changes made in Figure 2.

6. Figure 3: I would be tempted to put Figure S3A and S3B into Figure 3. It would be better to show all 1246 DARs together, either ordered by OTX2 CUT&RUN signal, or presented in two pre-defined groups (OTX2-bound vs unbound). I also suggest that the author show OTX2 signals and ATAC-seq signals for the 3028 DARs that gain accessibility in Otx2-null EpiLCs (this could be added to a supplemental figure).

Figures S3A and S3B have been moved to the main figure. Figure S3A is now part of Figure 3C, where all the 1,246 DARs are shown together, separated into two groups (OTX2-bound and -unbound). Figure S3B is now part of Figure 3F. A new heatmap showing the OTX2 and ATAC-seq signals for the 3028 regions that gain accessibility in Otx2^{-/-} EpiLCs has been added as new Figure S3B. Only 28 out of the 3,028 regions overlap an OTX2 peak as shown in the new Figure S3A. These regions appear to be already open in ESCs (Figure S3C) and they do not fully close when OTX2 is absent. This can be explained by either a) the lack of expression of an OTX2 target gene that represses these regions or b) the continuous expression of a gene that is usually repressed by OTX2 in the transition to EpiLCs. In both cases, OTX2 does not directly repress these regions. Figure legends have been amended to incorporate the new panels. The main text will be modified to incorporate these results.

Revision Plan

8. Figure 6: Do the PGCLCs with OTX2 expression have chromatin accessibility profiles similar to somatic cells? Consider adding WT somatic cell data to Figure 6A, which could be an interesting comparison with the Tam d0-d2 samples.

The heatmap showing the ATAC-seq signal at the additional OTX2-induced regions in somatic cells has been added to Figure 6A. The data show that the regions induced by OTX2 are not open in somatic cells generated in GK15. One possible explanation is the overexpression of OTX2 induces the opening of neural-associated regions, but neural differentiation is not fully supported in GK15 medium (see reviewer 2, point 3). As suggested by reviewer 2, we will perform RT-qPCR of germ layer markers to analyse the identity of somatic cells grown in GK15 (without cytokines) and somatic cells induced by OTX2 overexpression.

Reviewer 2:

2. The authors focus solely on the activating role of Otx2 in their data, but given the substantial proportion of DARs that decrease following Otx2 depletion, I presume it is possible that it also has a repressive effect? Either way, this should be discussed.

As also suggested by reviewer 1 (point 6), we analysed the accessibility level and the OTX2 signal at the 3,028 regions that gain accessibility in Otx2^{-/-} EpiLCs (new Figure S3A-C). These regions show high accessibility in ESCs suggesting that these are ESC regions that do not close properly in the transition to EpiLCs in the absence of OTX2. OTX2 CUT&RUN show a low to absent signal at these regions, with just 28 regions overlapping EpiLCs DARs that show higher accessibility in Otx2^{-/-} cells, suggesting that OTX2 does not have a direct suppressive effect on them.

5. The authors state that d2 PGCLCs "show an intermediate position between ESCs and EpiLCs" based on the PCA location. They should be careful to qualify that this is only in the first 2 principal components, because it may well be the case (and is likely) that in other components the PGCLC population is far removed from the pluripotent states.

The text has been updated as follows: d2 PGCLCs "show an intermediate position between ESCs and EpiLCs on both PC1 and PC2".

Reviewer2 Minor Suggestions:

1. Presumably the regions bound by OTX2 in Tet2, Mycn and Fgf5 (Fig1E) are called enhancers because these are known from existing literature. It would be helpful to cite the relevant references to this in the text for those unfamiliar with these.

References (Whyte et al, Cell, 2018 – Tet2 and Mycn, Buecker et al, Cell Stem Cell, 2013, Thomas et al, Mol Cell 2021 – Fgf5) have been added to the text and the figure legends.

Revision Plan

2. On page 13, the authors say "To determine whether OTX2 expression is essential to maintain chromatin accessibility in somatic cells..." but this does not seem to be what they test because they are using PGCLC medium. Perhaps I misunderstood, but this could be clarified.

Expression of OTX2 during the first 2 days of PGCLC differentiation leads to a block of germline differentiation as previously shown in Zhang, Zhang et al, Nature 2018. After 2 days of tamoxifen treatment, cells have acquired somatic fate and cells will undergo somatic differentiation even after tamoxifen is withdrawn after day 2. Nevertheless, we agree with the reviewer that the sentence is of difficult interpretation and we modified the sentence as shown below and as reported in the updated manuscript: "To determine whether OTX2 expression is essential to maintain chromatin accessibility in cells differentiating in the presence of PGC-inducing cytokines after day 2" (page 12).

3. On page 14 the authors claim, "These results indicate that...the partner proteins that OTX2 act alongside differ...". While this may be the case, their results do not substantiate this, it is just speculation. Should be toned down.

The text has been modified as follows: "These results suggest that...the partner proteins that OTX2 act alongside differ..."

4. Page 18, PGCLC differentiation method sections needs to be described as such (ie. Add "For PGCLC differentiation..." before the second paragraph)

The text "For PGCLC differentiation" has been added at the beginning of the PGCLC differentiation method section.

5. It would be helpful to indicate time on the protocol schematics (eg Fig4A, 5A, 5D etc) as I had to keep checking the methods to find out how long the full differentiation time-course was.

Indication of time has been added to Figures 1, 2, 4, 5 and 6.

6. Since the authors compare between the Tam d0-d2 treatments assessed at d2 versus d4 (Figure5B vs 5E) it would be helpful to make the colourbars the same scale, for both ATAC and Cut&Run datasets.

The heatmap in Figure 5B has been modified. The colourbars of Figure 5B and 5E are now using the same scale.

Reviewer 3:

1) As a minor point related to this, the second sentence is confusing since it kind of sounds like Otx2^{-/-} and WT cells are compared under the same conditions unless one carefully reads the previous sentence.

The text has been modified to clarify the different medium conditions for WT and OTX2^{-/-} cells, as follows: "In the presence of PGC-inducing cytokines, Otx2^{-/-} cells produce an essentially pure (>90%) CD61⁺/SSEA1⁺ population that we refer to as PGCLCs, while wild-type cells differentiated in GK15 medium without cytokines yield a cell population from which PGCLCs are absent" (page 7).

Revision Plan

2) "This suggests that OTX2 acts as a pioneer TF to regulate the accessibility of enhancers E1, E2 and E3."

This is from the text corresponding to Fig. 2. That data actually only shows that Otx2-null cells have DARs, so somehow OTX2 affects chromatin accessibility but it could be indirect by controlling transcription of genes that modify chromatin accessibility. It is not until figure 4 that the data suggests that OTX2 directly affects accessibility, perhaps as a pioneer TF.

The authors continue to make many statements about the direct action of OTX2 before the data supporting this is shown, on which I got hung up as a reader. I suggest the authors edit the manuscript to improve this. E.g. "OTX2 may directly control accessibility at these sites (Figure 3E)." and the fact that in 3E and other figure, it says "DARs increased by OTX2 binding" which at that point is not proven, so would better say "Otx2-null vs WT DARs" or something like that.

The sentence "This suggests that OTX2 acts as a pioneer TF to regulate.." has been removed from the text (page 9). The sentence "OTX2 may directly control accessibility at these sites" has been modified with "suggesting that the presence of OTX2 affects accessibility at these sites" (page 9). The sentence " Together, these results suggest that OTX2 is required to open these chromatin regions" has been modified to "Together, these results suggest that OTX2 is required for the accessibility of these chromatin regions".

The subset of DARs that increase in WT EpiLC and are bound by OTX2 that was called "DARs increased by OTX2 binding" has been renamed as "DARs higher in WT with OTX2 binding". For consistency, the subset of DARs showing increased accessibility in WT EpiLCs that are not bound by OTX2 are now called "DARs higher in WT without OTX2 binding" (Figure 3, Figure 4, main text and figure legends). We will further revise the manuscript to avoid statements or hypotheses that are not yet supported by data throughout the text.

Reviewer 3 – minor comments:

1) "Comparing wild-type and Otx2-/- ESCs identified 375 differentially accessible regions (DARs) with increased accessibility in wild-type cells, and 743 regions with higher accessibility in Otx2-/- ESCs (Figures 2C). An example of ESC DARs where accessibility is increased in cells expressing OTX2 is the intragenic enhancer of Tet2. Tet2 is expressed at high levels in ESCs but at low levels in EpiLCs."

The authors compare Otx2-null and WT ESCs then proceed to give an example comparing ESCs to EpiLCs, instead of Otx2-null vs WT ESCs, which is confusing.

Furthermore, here and in other places the authors describe ESCs as not expressing OTX2. However, they also show CUT&RUN data for OTX2 in ESCs etc, clearly indicating that it is expressed, just lower (otherwise how could one get anything?).

We originally chose Tet2 enhancer as an example of the 375 ESC DAR with higher accessibility in WT vs Otx2-/- ESCs as it shows a slightly decreased level of accessibility and OTX2 binding in ESCs. Therefore, the sentence "where accessibility is increased in cells

Revision Plan

expressing OTX2" refers to WT cells (expressing OTX2) when compared to Otx2^{-/-} cells (OTX2-null). The text has been changed to describe the new panel. The rest of the main text will be checked and modified where appropriate to avoid possible misinterpretations.

We also appreciate that the change in accessibility is not clearly visible in the original Figure 2, as also pointed out by Reviewer 1 (point 6). In the updated Figure 2, we show a region in the Hes1 locus as an example of the 375 ESC DARs. Moreover, we simplified the panels showing ATAC-seq tracks of WT and OTX2^{-/-} ESC (Fig. 2D-E) or EpiLCs (Fig. G-H).

2) "In contrast, in EpiLCs, OTX2 binds almost 40% (446 out of 1,246) of the DARs that are more accessible in wild-type than in Otx2^{-/-} cells (Figure 3B-C). Notably, these regions are mainly located distal to genes (91%, Figure 3D), despite the increased fraction of promoter regions bound by OTX2 in EpiLCs (Figure S1A)."

Are the authors rounding percentages with 2 significant digits, as suggested by the "91%"? If so, 446/1245 ~ 36%, not 40%.

The text has been modified from "OTX2 binds almost 40%" to "OTX2 binds 36%".

3) The results in Figure 4 are nice and the real meat of the paper.

One suggestion: It would be helpful if Fig. 4B were split up between the 446 and 800 genes instead of showing all 1246, and if the WT control was shown in the same figure as well.

Panels with the 446 and 800 regions have been added to Figure 4 instead of the panels with all 1246 regions. WT control has been inserted in Figure 4. The main text and the figure legends have been updated accordingly.

4) "Enforced OTX2 expression opens additional somatic regulatory regions" - it would be clearer to say "OTX2 overexpression opens additional somatic regulatory regions", since this is really about DARs between EpiLCs that already express OTX2 and those forced to express higher than WT endogenous levels by the OTX2-ER system?

We thank the reviewer for their suggestion. The text has been modified (page 12)

4. Description of analyses that authors prefer not to carry out

Dear Ian,

Thank you for the submission of your manuscript with referee reports and revision plan to EMBO reports.

I agree with your proposed revisions and would thus like to invite you to revise your manuscript with the understanding that the referee concerns must be fully addressed and their suggestions taken on board. Please address all referee concerns in a complete point-by-point response. Acceptance of the manuscript will depend on a positive outcome of a second round of review. It is EMBO reports policy to allow a single round of major revision only and acceptance or rejection of the manuscript will therefore depend on the completeness of your responses included in the next, final version of the manuscript.

We realize that it is difficult to revise to a specific deadline. In the interest of protecting the conceptual advance provided by the work, we recommend a revision within 3 months (22nd Aug 2025). Please discuss the revision progress ahead of this time with the editor if you require more time to complete the revisions.

- 1) A data availability section providing access to data deposited in public databases is missing. If you have not deposited any data, please add a sentence to the data availability section that explains that.
- 2) Your manuscript contains statistics and error bars based on $n=2$. Please use scatter blots in these cases. No statistics should be calculated if $n=2$.

3) We replaced Supplementary Information with Expanded View (EV) Figures and Tables that are collapsible/expandable online. A maximum of 8 EV Figures can be typeset. EV Figures should be cited as "Figure EV1, Figure EV2" etc... in the text and their respective legends should be included in the main text after the legends of regular figures.

5) a complete author checklist, which you can download from our author guidelines . Please insert information in the checklist that is also reflected in the manuscript. The completed author checklist will also be part of the RPF.

6) Please note that all corresponding authors are required to supply an ORCID ID for their name upon submission of a revised manuscript (. Please find instructions on how to link your ORCID ID to your account in our manuscript tracking system in our Author guidelines

- the name of the statistical test used to generate error bars and P values,
- the number (n) of independent experiments (please specify technical or biological replicates) underlying each data point,
- the nature of the bars and error bars (s.d., s.e.m.),
- If the data are obtained from n Program fragment delivered error ``Can't locate object method "less" via package "than" (perhaps you forgot to load "than"?) at //ejpvfs23/sites23b/embor_www/letters/embor_decision_rc_revise_and_rereview.txt line 56.' 2, use scatter blots showing the individual data points.

12) All Materials and Methods need to be described in the main text using our 'Structured Methods' format, which is required for all research articles. According to this format, the Methods section includes a Reagents and Tools Table (listing key reagents, experimental models, software and relevant equipment and including their sources and relevant identifiers) followed by a Methods and Protocols section describing the methods using a step-by-step protocol format. The aim is to facilitate adoption of the methodologies across labs. More information on how to adhere to this format as well as a downloadable template (.docx) for the Reagents and Tools Table can be found in our author guidelines:

An example of a Method paper with Structured Methods can be found here: <https://www.embopress.org/doi/full/10.1038/s44320-024-00037-6#sec-4>

I look forward to seeing a revised form of your manuscript when it is ready.

Reviewer #1 (Evidence, reproducibility and clarity (Required)):

Summary

OTX2 is a pivotal transcription factor that regulates the fate choice between somatic and primordial germ cell (PGC) lineages in early mouse development. In the current study, the authors use in vitro stem cell models to demonstrate that OTX2 mediates this developmental fate decision through controlling chromatin accessibility, whereby OTX2 helps to activate putative enhancers that are associated with somatic fate. By extension, those somatic-associated regulatory regions therefore become inaccessible in cells adopting PGC identity in which Otx2 is downregulated.

Comments

I enjoyed reading this manuscript. The experiments have been carried out well and for the most part the results provide convincing evidence to support the claims and conclusions in the manuscript. I particularly liked the experiments using the inducible Otx2 transgene to examine the acute changes in chromatin accessibility following restoration of OTX2.

I include some suggestions below to the authors for additional analyses that I feel would further strengthen their study.

I also felt that the authors focus almost exclusively on the subset of OTX2-bound sites that lose accessibility in the absence of OTX2. But, as they show in several figure panels, these sites tend to be the minority and that most OTX2-occupied sites do not lose accessibility in Otx2-null cells (actually, more sites tend to gain accessibility). I encourage the authors to modify the text and some of the analyses to give a better balance to their study.

We are pleased that this reviewer enjoyed our manuscript.

As suggested by the reviewer, we expanded our analysis including analysis for the 3,028 DARs that gain accessibility in Otx2-null EpiLCs (Expanded View Figure EV3 discussed under point 6, below).

We have responded to this reviewer's other points as detailed below.

1. Figure 1: The authors report in the methods that they performed OTX2 CUT&RUN in biological duplicates. It would strengthen their results if they showed in Figure S1 some representative data from each replicate separately to show the consistency.

As suggested by the reviewer to show consistency between replicates, two representative tracks of the two CUT&RUN replicates at the Tet2 (ESCs) and Fgf5 (EpiLCs) loci have been included in Expanded View Figure EV1A. The corresponding tracks of the average bigwig files are reported in Figure 1E.

2. Figure 1: The authors write: "...OTX2 binds mostly to putative enhancers." Whether these distal sites are enhancers is not sufficiently evidenced in the manuscript, but it is important information to collect to support their model of OTX2 function. The authors should strengthen their analysis by examining whether OTX2 peaks are enriched at previously defined enhancer regions.

We referred purposefully to these regions as "putative enhancers" because not all 'enhancer' lists are composed of regions with confirmed enhancer activity. Nevertheless, we agree with the reviewer that more details would be helpful to the reader. Therefore, we compared the OTX2 peaks to previously defined 'enhancer' regions. First, we removed OTX2 peaks within 1Kb of each TSS from each list of OTX2 peaks (ESC-specific, EpiLC-specific and common) to exclude promoters and therefore to obtain OTX2 bound ESC-specific, common and EpiLC-specific distal regions. These regions were then compared to two published datasets of 'enhancer' regions. The first dataset of enhancers was described in Whyte et al, where enhancers were defined in mESCs cultured in Serum/LIF by the co-localization of OCT4, SOX2 and NANOG, as co-localization of these three transcription factors is highly predictive of ESC enhancer function according to (Whyte et al, 2013; Chen et al, 2008). 49.6% of OTX2-bound common distal regions and 53.4% ESC-specific OTX2 bound distal regions overlap the list of enhancers published by Whyte et al.

To analyse the enrichment of OTX2 peaks at putative EpiLC enhancers, we compared our CUT&RUN data to the list of enhancers published by Christa Buecker (Buecker et al, 2014) and used in (Sankar et al, 2022) (GSE160974). In this case, enhancers are defined by the presence of H3K27ac, H3K4me1 and p300. 77.6% of common distal regions and 52.1% EpiLC-specific distal regions overlap with the enhancers defined by Buecker/Sankar.

From these comparisons, we concluded that at least half of the OTX2 peaks overlap previously reported enhancers, as characterised by co-localization of OCT4-SOX2-NANOG or the presence of H3K27ac-H3K4me1-p300. The results of the comparisons between OTX2-bound regions and Whyte and Buecker enhancer lists have been added to the main text (page 7).

As the lists of published enhancers may overlook regions that are not co-bound by the three TFs in ESCs or do not have H3K27ac marker in EpiLCs, we compared the OTX2 peaks with recently published ChIP-seq datasets for H3K4me1 (a marker of enhancer regions) and H3K27ac (a marker of active chromatin regions) generated in 2i/LIF ESCs and EpiLCs (Bleckwehl et al, 2021) (GSE155062). Active enhancers are defined by the presence of both H3K4me1 and H3K27ac, while primed enhancers are characterised by H3K4me1 only. OTX2-bound common regions showed H3K4me1 and H3K27ac in both ESCs and EpiLCs suggesting they are putative enhancers active in both cell types. OTX2-bound ESC-specific regions show high H3K4me1 and H3K27ac in ESCs, suggesting they are active enhancers in ESCs (see panel below). At these regions, both H3K4me1 and H3K27ac are strongly reduced or absent in EpiLCs, suggesting these regions are decommissioned during the transition from ESCs to EpiLCs. Conversely, OTX2 bound EpiLC-specific regions show low level of H3K4me1 and H3K27ac in ESCs, and high H3K4me1 in EpiLCs (see panel below), suggesting they are putative enhancers in EpiLCs. As H3K27ac is present in about 1/3 of OTX2-bound EpiLC-specific regions in EpiLCs, this suggests that this subset of OTX2-bound EpiLC-specific regions are active enhancers in EpiLCs, while the remaining regions are primed enhancers (H3K4me1 only) that may activate in later stages of somatic differentiation.

The heatmaps of H3K4me1 and H3K27ac signals generated with the data from (Bleckwehl et al, 2021) have been added to Expanded View Figure EV1 and are described in the text (page 7).

3. Figure 2: I think it would be helpful to remind the reader here that *Otx2* is normally downregulated in PGCs, and that *Otx2* expression is maintained (at least initially) in somatic cells. This would help explain the logic behind the choice of samples that were profiled. That said, I'm still puzzled why the authors did not examine flow-sorted WT+cyto cells?

We modified the text in the Introduction with the following sentence, as suggested by the reviewer, emphasizing the level of OTX2 in early somatic vs early PGCLCs: "Subsequently, Otx2 is rapidly downregulated as EpiLCs transition to PGCLCs, but is maintained initially during somatic differentiation (Zhang et al, 2018)" (page 4).

*We agree with the reviewer that it would be interesting to examine flow-sorted WT +cyto PGCLCs. Unfortunately, the expression of CD61 and SSEA1 only becomes visible from day 4 of PGCLC differentiation. Therefore, we were not able to isolate PGCLC at day 2 from WT cells differentiated in the presence of cytokines. It was previously shown that day 6 *Otx2*^{-/-} PGCLCs are transcriptionally similar to sorted day 6 WT cells (Zhang et al, 2018). Therefore, we used OTX2^{-/-} cells at day 2 to model PGCLCs. We modified the text (pages 8-9) to clarify this point (also raised by reviewers 2 and 3).*

4. Figure 2D: I appreciate that the highlighted region at the *Tet2* locus is a DAR, but from the genome tracks it looks as though the region still has high accessibility. Are there any other examples to exemplify a more obvious DAR? Additionally, since twice as many DARs gain accessibility in *Otx2*-null ESCs compared to lose accessibility, why not show examples of these as well? The same is true of EpiLCs. (Or alternatively, provide a good explanation for why not to show these other categories)

*The reviewer makes a good point, for which we are thankful. Therefore we substituted the *Tet2* DAR with a clearer example of an ESC DAR (located in the *Hes1* locus) that shows low accessibility in *Otx2*^{-/-} ESCs versus WT ESCs. Examples of ESC DARs and EpiLC DARs that show higher accessibility in *Otx2*^{-/-} vs WT cells have been added as new panels 2E (DAR in *Pebp4* locus) and 2H (DAR in *Tdh* locus). We also simplified the panels showing only ATAC-seq tracks in WT and OTX2^{-/-}*

cells, either ESCs (2D-E) or EpiLCs (G-H). The text (page10) and figure legends have been modified to accommodate the changes made in Figure 2.

5. Figure 2: the authors write: "This suggests that OTX2 acts as a pioneer TF...". However, at this point in the manuscript, there is no evidence to support that OTX2 might have pioneer activity. I think this claim would be better suited to later in the manuscript, or in the discussion, following the finding that reintroduction of OTX2 can induce chromatin accessibility at previously closed sites.

The text in page 9 has been modified and the sentence "This suggests that OTX2 acts as a pioneer TF..." has been removed. As suggested, we now discuss the ability of OTX2 to act as a pioneer factor in the discussion (pages 19-20).

6. Figure 3: I would be tempted to put Figure S3A and S3B into Figure 3. It would be better to show all 1246 DARs together, either ordered by OTX2 CT&RUN signal, or presented in two pre-defined groups (OTX2-bound vs unbound). I also suggest that the author show OTX2 signals and ATAC-seq signals for the 3028 DARs that gain accessibility in Otx2-null EpiLCs (this could be added to a supplemental figure).

Figures S3A and S3B have been moved to the main figure. Figure S3A is now part of Figure 3C, where all the 1,246 DARs are shown together, separated into two groups (OTX2-bound and -unbound).

New Figure 3C:

Figure S3B is now part of Figure 3F:

F

A new heatmap showing the OTX2 and ATAC-seq signals for the 3,028 regions that gain accessibility in *Otx2*^{-/-} EpiLCs has been added as new Expanded View Figure EV3A. Only 28 out of the 3,028 regions overlap an OTX2 peak as shown in the new Expanded View Figure EV3B. These regions appear to be already open in ESCs (Expanded View Figure EV3C) and they do not fully close when OTX2 is absent in EpiLCs. This can be explained by either a) the lack of expression of an OTX2 target gene that represses these regions or b) the continuous expression of a gene that is usually repressed by OTX2 in the transition to EpiLCs. In both cases, OTX2 does not directly repress these regions.

A

B

C

Figure legends have been amended to incorporate the new panels. The description of the data has been incorporated in the main text (pages 11-12).

7. Figure 3: What is special about the 8% of OTX2-bound site that lose accessibility, versus the 92% of sites that do not?

To identify what is special about the 8% of OTX2-bound sites that lose accessibility, we performed motif analysis in the 5,261 regions that do not lose accessibility (see below, panel C) and compared the results with the motif analysis in the 446 regions (8%) that lose accessibility (Figure 3F). OTX2 (and OTX2-like) motifs are the most enriched in both groups of regions, as expected as OTX2 binds to these regions. Interestingly, the OCT4-SOX2 motif is enriched in the sites that do not lose accessibility but it is not enriched in the sites that lose accessibility in the absence of OTX2 (ie, the 8%). Both OCT4 and SOX2 interact with OTX2 (Buecker et al, 2014) and are able to open chromatin (Soufi et al, 2015). OTX2 may not be required for chromatin accessibility at the 5,261 regions that do not lose accessibility without OTX2 and that contain OCT4-SOX2 motifs. At these regions accessibility may be controlled by OCT4 or SOX2. Moreover, expression of OTX2 in *Otx2*^{-/-} OTX2-ERT2 EpiLCs does not induce an increase in chromatin accessibility at the 5,261 (see below, panels A and B). In contrast, expression of OTX2 in *Otx2*^{-/-} OTX2-ERT2 EpiLCs does induce increased chromatin accessibility in the 8% of the OTX2-bound sites that lose accessibility without OTX2 (Figure 3B, D). These results suggest that the higher sensitivity to the absence of OTX2 shown by the 8% of OTX2-bound regions may result from the lack of an OCT4-SOX2 motif rendering these 8% dependent on OTX2 to maintain an open chromatin state.

The panels above have been added as new Figure EV4.

The following heatmap of OTX2 CUT&RUN and ATAC-seq in the 5,261 OTX2-bound sites that do not change accessibility in the absence of OTX2 showing no changes in accessibility at these regions has been added to Figure EV3D.

The text (page 13) was modified to incorporate these new data.

8. Figure 6: Do the PGCLCs with OTX2 expression have chromatin accessibility profiles similar to somatic cells? Consider adding WT somatic cell data to Figure 6A, which could be an interesting comparison with the Tam d0-d2 samples.

The heatmap showing the ATAC-seq signal at the additional OTX2 induced regions in somatic cells has been added to Figure 6A (see below).

The data show that the regions induced by OTX2 are not open in somatic cells generated in GK15.

9. Figure 6F: If the 4221 sites are split into those bound by OTX2 versus those that are not (related to Figure 6C) then is there a difference? i.e. are the OTX2-bound sites opening up?

We separated the 4,221 sites in OTX2 bound and unbound. The result is reported below:

Although there is a slight increase in accessibility in the OTX2 bound subset at 1-6 hours, the average accessibility at 6h is less than ¼ of the accessibility of these regions when OTX2 is present from day 0 to day 4, while the OTX2 unbound regions do not show an increase in accessibility. These dynamics are extremely slow compared to the 1,246 DARs that are more accessible in WT EpiLCs where accessibility reaches 50% of the d0-d4 sample in just 1 hour of tamoxifen treatment. This indicates that these 4,221 regions are not as responsive to the presence of OTX2 as the 1,246 EpiLC regions and suggest that they may require additional factors for the establishment of their accessibility.

10. Is there any evidence that OTX2 binds and compacts PGCLC enhancers in somatic cells? I appreciate this is different to the main thrust of the authors' model, but being able to show that OTX2 does not compact these sites lends further support to their preferred model of OTX2 opening sites of somatic lineages.

Comparing the ATAC-seq in PGCLCs with ESCs and EpiLCs, we identify a subset of regions that are open only in PGCLCs (PGCLC-specific accessible regions, see below:

We analysed OTX2 CUT&RUN in WT EpiLC, OTX2-ERT2 PGCLCs in presence or absence of Tamoxifen at these regions and we did not detect OTX2 binding in any of the analysed samples, suggesting that OTX2 does not bind and compact PGCLC-specific enhancers in WT cells or when OTX2 expression is enforced in GK15 medium. See below:

These analyses have been incorporated in the revised version of the manuscript (Expanded View Figure EV6 and pages 15-16) .

11. Discussion: Have prior studies established a connection between OTX2 and chromatin remodellers that can open chromatin? Or, if not, then perhaps this could be proposed as a line of future research.

We thank the reviewer for suggesting to amplify the discussion on the possible connection between OTX2 and chromatin remodellers. Although there is no evidence in literature of a direct interaction between OTX2 and chromatin remodellers, this cannot be excluded. The connection might also be indirect: OTX2 is known to interact with OCT4, which in turn interacts and recruits to chromatin the catalytic subunit of the SWI/SNF complex, BRG1. We have revised the discussion to incorporate these points (pages 19-20).

Reviewer #1 (Significance (Required)):

Strengths

The results presented provide a careful dissection of the role of OTX2 in controlling chromatin accessibility in different stages of pluripotent to somatic and PGC fates. The authors do a good job of revealing the stage-specific differences in OTX2 occupancy and chromatin accessibility as well as the different responses following the acute reintroduction of OTX2.

Limitations

I felt that the authors could present/discuss a bit more on alternative possibilities and models, as it would help the reader to better understand why they favour one model over other ones, and presenting these other possibilities could also provide more support for their preferred model.

We added a new paragraph in the discussion (pages 22-23), also based on the new results we discuss in the response to points 7 and 10 .

Whether OTX2 is binding to putative enhancers is inferred but could be evidenced more strongly, as that is important for their model.

We addressed this point in the response to point 2 with the analysis of histone modifications at OTX2-bound sites.

Advance

This study provides key information to understand the mechanisms of OTX2 function in cell fate choice. Similar functions have been shown in other contexts for other transcription factors, but this is a nicely done study and adds to our understanding of how transcription factors function in early development to direct cell-fate decisions.

Expertise

My field of expertise lies in the gene regulatory control of early developmental decisions.

Reviewer #2 (Evidence, reproducibility and clarity (Required)):

Barbieri and Chambers explore the role of OTX2 on mouse pluripotency and differentiation. To do so, they examine how the chromatin accessibility and OTX2 binding landscape changes across pluripotency, the exit of pluripotency towards formative and primed states, and through to PGCLC/somatic differentiation. The work mostly represents a resource for the community, with possible implications for our understanding of how OTX2 might mediate the germline-soma switch of fates. While the findings of the work are modest, the results seem solid and the manuscript is clear and well-written. I have some comments as indicated below:

We are pleased that this reviewer found our results solid and the manuscript clear.

1. The comparison between *Otx2*^{-/-} cells in the presence of PGCLC cytokines compared to WT cells in the absence of cytokines seems like it is missing controls to me. I assume the authors wanted to enable homogeneous populations to facilitate their bulk sequencing methods, but it seems to me like they are comparing apples with oranges. It would have been better to have the reciprocal situations (*Otx2*^{-/-} cells in basal differentiation medium, and WT cells in PGCLC cytokines) with a sorting strategy to better unpick the differences between the presence and absence of *Otx2* in the 2 protocols. Having said that, the authors are careful not to draw many comparisons between those populations so I don't think this omission affects their current claims. They should however clarify whether the flow cytometry (Supp Fig2) was used for sorting cells or if all cells were taken for bulk sequencing.

*We agree it would be of interest to compare the PGCLC and somatic population derived from the OTX2^{-/-} cells in GK15 without cytokines with the same populations derived from WT cells differentiated in the presence of cytokines. The aim of this work is to identify how OTX2 works to inhibit germline differentiation. Previous work from our lab showed that cells lose dual competence for both germline and somatic differentiation after day 2, when PGC are committed and when enforced expression of OTX2 cannot block germline differentiation. Therefore, we examined cells at day 0 and day 2 to compare cells committed to the PGC fate with cells committed to the somatic fate. Unfortunately, the two surface markers characteristics of PGCs (CD61 and SSEA1) are not expressed at day 2 and, therefore we are not able to use CD61+SSEA1+ to sort PGCLCs derived from mixed populations of cells such as OTX2^{-/-} cells differentiated in GK15 without cytokines or WT cells differentiated in the presence of cytokines. As recognised by this reviewer, we aimed to obtain two homogenous populations that can model PGCLCs and somatic cells, based on data obtained at day 6 (Zhang et al, 2018). At this time, *Otx2*^{-/-} PGCLCs show a similar transcriptome to PGCLCs sorted at day 6 from WT cells differentiated with PGC-promoting cytokines. Also at day 6 WT cells differentiated without PGC-promoting cytokines completely lack PGCs (Zhang et al, 2018). As this important point has been raised by reviewers 1 and 2 as well, we modified the text (pages 8-9) to clarify the choice behind using OTX2^{-/-} cells in presence of cytokines and WT cells in the absence of cytokines to model PGCLCs and somatic cells respectively. The flow cytometry plots in Figure EV2 generated at day 6 of differentiation in GK15 with or without cytokines are not a sorting strategy but illustrate the different yields of PGCLCs obtained from *Otx2*-null and WT cells.*

2. The authors focus solely on the activating role of *Otx2* in their data, but given the substantial proportion of DARs that decrease following *Otx2* depletion, I presume it is possible that it also has a repressive effect? Either way, this should be discussed.

A similar point was raised by reviewer 1 and our response to that is given above. In this manuscript we identified 1,246 regions with decreased accessibility in the absence of OTX2, showing that accessibility is rescued by the nuclear relocation of OTX2-ERT2 after tamoxifen treatment,

demonstrating an activating role for OTX2 in EpiLCs. As pointed out by the reviewer, a larger proportion of DARs showed increased accessibility in the absence of OTX2, suggesting a possible repressive effect for OTX2. To answer this question, we analysed the accessibility level and the OTX2 signal at the 3,028 regions that gain accessibility in *Otx2*^{-/-} EpiLCs (new Expanded View Figure EV3A-C).

*These regions show high accessibility in ESCs suggesting that these are ESC regions that do not close properly in the transition to EpiLCs in the absence of OTX2. OTX2 CUT&RUN show a low to absent signal at these regions, with just 28 regions overlapping EpiLCs DARs that show higher accessibility in *Otx2*^{-/-} cells. This suggests that OTX2 does not have a direct suppressive effect on them. The description of the data has been incorporated in the main text (pages 11-12) and figure legends have been amended.*

3. Throughout the text, the authors subject cells (WT / *Otx2*^{-/-} /*Otx2*ER) to different protocols to look at accessibility and *Otx2* binding, but with no mention of the cell fate differences that occur in these different conditions. For instance, it is unclear to me to which fate the WT cells without PGCLC cytokines go - I presume this is neural but perhaps this is a mixed fate, given that they are in GK15 rather than N2B27. Likewise, the OTX2ER experiments may promote a mixed population between PGCLC/somatic fates, and this is never described. Ideally transcriptomic data would be collected, but failing that, qPCR data should be obtained to examine this more closely.

As suggested by the reviewer, we generated RT-qPCR data to analyse the cell fate of WT cells differentiated in the absence of cytokines as well as OTX2-ERT2 cells treated with Tamoxifen.

*We first analysed pluripotency associated markers (*Pou5f1*-*Oct4*, *Nanog*, *Sox2*) in these cells (see below). While *Otx2*^{-/-} OTX2-ERT2 PGCLCs showed high expression of *Pou5f1*, *Nanog* and *Sox2*, somatic cells and Tamoxifen-treated *Otx2*^{-/-} *Otx2*-ERT2 cells showed a reduced level of all three pluripotent markers at day 4, suggesting that differentiation in GK15 without cytokines or overexpression of OTX2 in cells differentiating in GK15 causes a loss of pluripotency.*

We then analysed the PGC-specific markers *Prdm14*, *Tfap2c* (AP2 γ) and *Prdm1* (Blimp1) (see below). As expected, *Tfap2c*, *Prdm1* and *Prdm14* are expressed in untreated *Otx2*^{-/-} OTX2-ERT2 PGCLCs. Expression of all 3 is reduced by day 4 in somatic cells. Enforced OTX2 expression in Tamoxifen treated OTX2-ERT2 cells cultured in the presence of PGC-inducing cytokines inhibits expression of *Prdm14*. Both *Tfap2c* and *Prdm1* expression is similar between treated and untreated OTX2-ERT2 cells.

We also analysed the level of somatic markers: *Sox17* (endoderm), *Kdr* (mesoderm), *Sox1* (neural ectoderm) and *Dlx5* for surface ectoderm (see below). Somatic cells generated in GK15 without PGC-promoting cytokines showed expression of neural ectoderm (*Sox1*) and lower expression of mesoderm and endoderm markers (*Kdr* and *Sox17*). As expected, untreated *Otx2*^{-/-} OTX2-ERT2 cells did not express somatic markers, as they differentiate towards the germline (see PGC markers in panel A above). Tamoxifen treated OTX2-ERT2 cells showed expression of endoderm (*Sox17*), mesoderm (*Kdr*) and surface ectoderm (*Dlx5*) markers.

C

Overall, we can conclude that wildtype cells differentiated in GK15 in the absence of cytokines differentiate towards a mixed population of cells containing neural ectoderm and mesendoderm. Cells cultured in GK15 + PGC-promoting cytokines express PGC and pluripotency TFs with minimal expression of somatic markers. Cells cultured similarly, but with enforced OTX2 expression shut down pluripotency TF expression. This is also the case for *Prdm14*, but not for *Prdm1* and *Tfap2c*. In addition, enforced OTX2 expression activates expression of *Sox17* and *Kdr* but not the neural ectoderm marker *Sox1*. This can be explained by the presence of BMP4 in the PGC-promoting cocktail of cytokines used to induced PGCLCs, as BMP4 inhibits neural differentiation (Di-Gregorio et al, 2007; Ying et al, 2003). Interestingly, OTX2 overexpressing cells induced *Dlx5* expression, suggesting that in the presence of BMP4, OTX2 may direct the cells toward a surface ectoderm rather than neural ectoderm fate.

These data have been included in the manuscript as Expanded View Figure EV5 and are discussed in the text in page15.

4. The authors also state that "OTX2 facilitates *Fgf5* transcription" (page10) but provide no transcriptional data to substantiate this claim. Again RT-qPCR would help make this point.

We analysed the level of *Fgf5* by RT-qPCR in OTX2-ERT2 EpiLCs treated for 1 hour and 6 hours with Tamoxifen to show the effect of OTX2 on *Fgf5* transcription. As shown in new Figure 4F, the relocalization of OTX2-ERT2 into the nucleus following the addition of Tamoxifen leads to an increased level of *Fgf5* that reaches its expression level shown in wild-type EpiLCs after 6h of Tamoxifen treatment. The main text (page 10) and the figure legends have been modified to accommodate the new data.

5. The authors state that d2 PGCLCs "show an intermediate position between ESCs and EpiLCs" based on the PCA location. They should be careful to qualify that this is only in the first 2 principal components, because it may well be the case (and is likely) that in other components the PGCLC population is far removed from the pluripotent states.

The text has been updated as follows: d2 PGCLCs "show an intermediate position between ESCs and EpiLCs on both PC1 and PC2" (page 9).

6. It is unclear to me what the 'increase[d] accessibility' (eg abstract final sentence, Figure 3E) really means at the cellular level. Does this indicate that more cells have this site open, and does this have implications for the heterogeneity of cell fates observed? Since the authors are concerned with fate decisions, this seems like an important consideration that should at least be discussed.

We agree with the reviewer and have therefore revised the discussion on pages 20-21 to include the possibility that the increased accessibility is due to higher heterogeneity in the population.

Minor Suggestions:

1. Presumably the regions bound by OTX2 in Tet2, Mycn and Fgf5 (Fig1E) are called enhancers because these are known from existing literature. It would be helpful to cite the relevant references to this in the text for those unfamiliar with these.

As discussed in the reply to reviewer 1, the distal regions in the Tet2 and Mycn loci have been identified previously as enhancers based on the co-occupancy of the transcription factors (TFs) OCT4, NANOG and SOX2 (Whyte et al, 2013) and a previous paper (Chen et al, 2008) reported that co-occupancy of these TFs is predictive of enhancer activity.

Fgf5 enhancers have been firstly described in (Buecker et al, 2014) as bound by OCT4 and characterized by p300, H3K27ac and H3K4me1 ChIP-seq signals, and further characterized for their enhancer activity in (Thomas et al, 2021). These references have been added to the text and the figure legends.

2. On page 13, the authors say "To determine whether OTX2 expression is essential to maintain chromatin accessibility in somatic cells..." but this does not seem to be what they test because they are using PGCLC medium. Perhaps I misunderstood, but this could be clarified.

Thanks for spotting this. We removed "in somatic cells" The sentence now reads: "To determine whether OTX2 expression is essential to maintain chromatin accessibility in cells differentiating in the presence of PGC-inducing cytokines after day 2" (page 14).

3. On page 14 the authors claim, "These results indicate that...the partner proteins that OTX2 act alongside differ...". While this may be the case, their results do not substantiate this, it is just speculation. Should be toned down.

The text has been modified as follow: "Therefore, while OTX2 is expressed in both ESCs and EpiLCs, we speculate that the partner proteins that OTX2 may act alongside differ in naïve and formative pluripotent states" (page 18).

4. Page 18, PGCLC differentiation method sections needs to be described as such (ie. Add "For PGCLC differentiation..." before the second paragraph)

The text "For PGCLC differentiation" has been added at the beginning of the PGCLC differentiation method section (page 25).

5. It would be helpful to indicate time on the protocol schematics (eg Fig4A, 5A, 5D etc) as I had to keep checking the methods to find out how long the full differentiation time-course was.

Indication of time has been added to Figure 1, 2, 4, 5 and 6.

6. Since the authors compare between the Tam d0-d2 treatments assessed at d2 versus d4 (Figure5B vs 5E) it would be helpful to make the colourbars the same scale, for both ATAC and Cut&Run datasets.

The heatmap of the Tamoxifen treatment analysed at day 2 (Figure 5B) has been modified as requested.

Reviewer #2 (Significance (Required)):

The study examines the binding of OTX2 and subsequent chromatin accessibility in pluripotent, primed and differentiated (PGCLC/Somatic) cell states, including through Otx2^{-/-} cell lines and temporally-controlled exogenous expression of Otx2. As such, it represents a valuable resource into the potential direct targets of Otx2 and their change in accessibility state across cell types. The work is likely to be of interest to those working on understanding the exit of pluripotency, gene regulatory networks, and chromatin remodelling. My expertise is in cell fate decisions, pluripotency regulation and PGC(LC) differentiation.

Reviewer #3 (Evidence, reproducibility and clarity (Required)):

Summary:

In this manuscript, the authors perform OTX2 CUT&RUN and ATAC-seq in Otx2-null and WT ESCs, EpiLCs and PGCLCs to understand whether the role of OTX2 in restricting mouse germline entry that they previously described (Zhang Nature 2018) mechanistically depends on chromatin remodeling. They identify differentially accessible regions (DARs) between Otx2-null and WT cells at different stages of differentiation and show that many of these are OTX2 bound in WT. They then show using cells expressing OTX2-ER^{T2} in Otx2-null Epiblast cells that when OTX2 is moved into the nucleus, the regions that were differentially closed in Otx2-null open within an hour, suggesting chromatin accessibility is directly controlled by OTX2 (rather than indirect effects involving transcription and translation which one would expect to take longer). The scope is narrow, but this is nice work and useful data for the mouse PGC field. However, there are a few places where the data could be strengthened, and the writing is a little confusing in places, for example by stating as fact in early sections what is not proven until later.

We are pleased that the reviewer considers our work useful for the mouse PGC field, and grateful for their suggestions to improve the manuscript. We have included new analysis and modified the text as suggested to improve the writing, and to avoid early statements that were not fully proven until later in the manuscript. We have responded to other points they raised as detailed below.

Major Comments:

1) "we compared Otx2^{-/-} cells cultured in the presence of PGC-promoting cytokines with wild-type cells cultured in the absence of PGC-promoting cytokines. Under these conditions Otx2^{-/-} cells produce an essentially pure (>90%) CD61⁺/SSEA1⁺ population that we refer to as PGCLCs, while wild-type cells yield a cell population from which PGCLCs are absent"

This is not a controlled comparison since one cannot separate the day 2 effect of cytokines from that of the Otx2 knockout. The manuscript would be strengthened if the authors include WT somatic and PGCLCs from the +cytokine conditions, which could be easily sorted out as shown in Supp. Fig. 2. Ideally they would also include Otx2-null somatic cells, although Supp. Fig. 2 shows those are rare under the conditions considered.

We agree with the reviewer that it would be of interest to compare the PGCLC and somatic population derived from the OTX2^{-/-} cells in GK15 without cytokines with the same populations derived from WT cells differentiated in the presence of cytokines. As discussed in the response to reviewer 1 point 1, the aim of this work is to identify how OTX2 works to block germline differentiation. Previous work from the lab show that enforced OTX2 expression cannot block PGCLC differentiation after day 2, suggesting that at that time, cells are committed to the germline. Therefore, we have focussed on this time of differentiation (day 2) to compare cells committed to the PGC fate with cells committed to the somatic fate. Unfortunately, the two surface markers characteristics of PGCs (CD61 and SSEA1) are not expressed at day 2 and, therefore we are not able to sort PGCLCs derived from OTX2^{-/-} cells in GK15 without cytokines or WT cells differentiated in the presence of cytokines. As recognised by this reviewer, we aimed to obtain two homogenous populations that can model PGCLCs and somatic cells. This is based on data obtained at day 6 when Otx2^{-/-} PGCLCs show a similar transcriptome to sorted day 6 WT cells (Zhang et al, 2018). We have clarified that the supplementary Figure 2 is not a sorting strategy. As this point has been raised by reviewers 1 and 2 as well, we modified the text (pages 8-9) to clarify the use of OTX2^{-/-} cells in the

presence of cytokines and WT cells in the absence of cytokines to model PGCLCs and somatic cells respectively.

As a minor point related to this, the second sentence is confusing since it kind of sounds like Otx2^{-/-} and WT cells are compared under the same conditions unless one carefully reads the previous sentence.

The text has been modified to clarify the different medium conditions for WT and OTX2^{-/-} cells, as follows: "To circumvent this problem, we compared Otx2^{-/-} cells cultured in the presence of PGC-promoting cytokines with wild-type cells cultured in the absence of PGC-promoting cytokines. Under these conditions Otx2^{-/-} cells produce an essentially pure (>90%) CD61⁺/SSEA1⁺ population (Zhang et al, 2018a; Hayashi et al, 2011; Hayashi & Saitou, 2013), while wild-type cells yield a cell population from which PGCLCs are absent (Figures 2A and EV2). Therefore, we used Otx2^{-/-} cells cultured in the presence of PGC-promoting cytokines to model PGCLCs and wild-type cells cultured in the absence of PGC-promoting cytokines to model somatic cells." (pages 8-9).

2) "This suggests that OTX2 acts as a pioneer TF to regulate the accessibility of enhancers E1, E2 and E3."

This is from the text corresponding to Fig. 2. That data actually only shows that Otx2-null cells have DARs, so somehow OTX2 affects chromatin accessibility but it could be indirect by controlling transcription of genes that modify chromatin accessibility. It is not until figure 4 that the data suggests that OTX2 directly affects accessibility, perhaps as a pioneer TF.

The authors continue to make many statements about the direct action of OTX2 before the data supporting this is shown, on which I got hung up as a reader. I suggest the authors edit the manuscript to improve this. E.g. "OTX2 may directly control accessibility at these sites (Figure 3E)." and the fact that in 3E and other figure, it says "DARs increased by OTX2 binding" which at that point is not proven, so would better say "Otx2-null vs WT DARs" or something like that.

The sentences "This suggests that OTX2 acts as a pioneer TF to regulate..", "OTX2 may directly control accessibility at these sites" and "Together, these results suggest that OTX2 is required to open these chromatin regions" have been removed from the text.

The subset of DARs that increase in WT EpiLC and are bound by OTX2 that was called "DARs increased by OTX2 binding" has been renamed as "DARs higher in WT with OTX2 binding". For consistency, the subset of DARs showing increased accessibility in WT EpiLCs that are not bound by OTX2 are now called "DARs higher in WT without OTX2 binding" (Figure 3, Figure 4, main text and figure legends). We further revised the manuscript to avoid statements or hypotheses that are not yet supported by data throughout the text.

3) "In ESCs, OTX2 binds <10% (30 out of 375) of DARs that are more accessible in wild-type cells than in Otx2^{-/-} cells (Figure 3A), suggesting that accessibility of ESC DARs is directly due to OTX2 in a small subset of DARs."

When a small number of DARs are OTX2 bound, it does not necessarily suggest that that small set is directly affected by OTX2. It could just mean no DARs are controlled directly by OTX2 and then some are bound by chance by OTX2. Some appropriate statistical null hypotheses about the occurrence of OTX2 motifs might help to see if 10% is more than chance.

Motif analysis with HOMER in the 30 DARs that overlap an OTX2 peak show enrichment for OTX2 (p -value = $1e-16$) and OTX2-like (GSC, p -value = $1e-14$) motifs using all the accessible regions in ESCs as background. This suggests that OTX2 binds to the regions that have OTX2 motif and not by chance.

Total Target Sequences = 30, Total Background Sequences = 97714

Rank	Motif	Name	P-value
1		Otx2(Homeobox)/EpiLC-Otx2-ChIP-Seq(GSE56098)/Homer	1e-16
2		GSC(Homeobox)/FrogEmbryos-GSC-ChIP-Seq(DRA000576)/Homer	1e-14
3		CRX(Homeobox)/Retina-Crx-ChIP-Seq(GSE20012)/Homer	1e-10
4		RARg(NR)/ES-RARg-ChIP-Seq(GSE30538)/Homer	1e-8
5		RAR:RXR(NR),DR0/ES-RAR-ChIP-Seq(GSE56893)/Homer	1e-8
6		Nr5a2(NR)/mES-Nr5a2-ChIP-Seq(GSE19019)/Homer	1e-5

We also analysed the occurrence of OTX2 motifs in the 375 ESC DARs more accessible in the wild-type than $Otx2^{-/-}$ cells using FIMO, part of the MEME suite, and bedtools. We identified 100 DARs that contain at least one OTX2 motif. As control we used the list of all the accessible regions in ESCs, both wild-type and $Otx2^{-/-}$ and we found 10,146 regions containing at least one OTX2 motif in 116,970 ESC accessible regions. We then applied a binomial test using R as following:

Observed number of regions with motifs

k <- 100

Total number of sequences

n <- 375

Expected probability under null hypothesis

p0 <- (10146/116970)

One-sided binomial test: is observed frequency > chance?

binom.test(k, n, p = p0, alternative = "greater")

We obtain the following result:

Exact binomial test

data: k and n

number of successes = 100, number of trials = 375, p-value < 2.2e-16

alternative hypothesis: true probability of success is greater than 0.08674019

95 percent confidence interval:

0.2292419 1.0000000

sample estimates:

probability of success
0.2666667

This shows that the occurrence of OTX2 motif in the 375 DARs is higher than by chance (0.0867). We included the results of the motif analysis in page 10. We have not included the results of the binomial test on OTX2 motif occurrence in the 375 regions, as this may result in cumbersome writing. However, we can do so if required.

4) It would be good if the discussion was broadened to include both human and other transcription factors that are involved. How much of these conclusions could one expect to carry over to human or other mammals? There is some work from the Surani lab considering OTX2 in human. One could even look at published ATAC or OTX2 chip-seq data in hPSCs and potentially learn something interesting. Furthermore, there are studies on other transcription factors modulating chromatin accessibility in the decision between germline and somatic cells, for example PRDM1, PRDM14 (refs in e.g. Tang et al Nat Rev Gen 2016) or TFAP2A (at least in human (Chen et al Cell Rep 2019)). Do these factors affect the same genes? Is a coherent picture emerging of their respective roles in germline entry?

As suggested by the reviewer, we included a section to discuss the role of OTX2 in human PGCLC formation (pages 21-22). We included as well a paragraph discussing roles of PRDM1, PRDM14, TFAP2A and TFAP2C in controlling chromatin accessibility in hESCs and hPGCLCs (pages 21-22).

Minor comments:

1) "Comparing wild-type and Otx2^{-/-} ESCs identified 375 differentially accessible regions (DARs) with increased accessibility in wild-type cells, and 743 regions with higher accessibility in Otx2^{-/-} ESCs (Figures 2C). An example of ESC DARs where accessibility is increased in cells expressing OTX2 is the intragenic enhancer of Tet2. Tet2 is expressed at high levels in ESCs but at low levels in EpiLCs."

The authors compare Otx2-null and WT ESCs then proceed to give an example comparing ESCs to EpiLCs, instead of Otx2-null vs WT ESCs, which is confusing.

Furthermore, here and in other places the authors describe ESCs as not expressing OTX2. However, they also show CUT&RUN data for OTX2 in ESCs etc, clearly indicating that it is expressed, just lower (otherwise how could one get anything?).

As also suggested by reviewer 1 (point 4), we modified the panel in Figure 2 and substituted the Tet2 enhancers with an open regions in the Hes1 locus as an example of the 375 ESC DARs.

We also revised the text to avoid possible misinterpretation when referring to OTX2 expression in ESCs. For example the sentence highlighted by the reviewer has been modified as follows: Examples of DARs with increased accessibility in wild-type or Otx2^{-/-} ESCs are present in the Hes1 and Pebp4 loci, respectively (page 10)

2) "In contrast, in EpiLCs, OTX2 binds almost 40% (446 out of 1,246) of the DARs that are more accessible in wild-type than in *Otx2*^{-/-} cells (Figure 3B-C). Notably, these regions are mainly located distal to genes (91%, Figure 3D), despite the increased fraction of promoter regions bound by OTX2 in EpiLCs (Figure S1A)."

Are the authors rounding percentages with 2 significant digits, as suggested by the "91%"? If so, 446/1245 ~ 36%, not 40%.

The text has been modified from "OTX2 binds almost 40%" to "OTX2 binds 36%" (page 10).

3) The results in Figure 4 are nice and the real meat of the paper.

One suggestion: It would be helpful if Fig. 4B were split up between the 446 and 800 genes instead of showing all 1246, and if the WT control was shown in the same figure as well.

Panels with the 446 and 800 regions have been added to figure 4 instead of the panels with all 1246 regions. WT control has been inserted in Figure 4.

4) "Enforced OTX2 expression opens additional somatic regulatory regions" - it would be clearer to say "OTX2 overexpression opens additional somatic regulatory regions", since this is really about DARs between EpiLCs that already express OTX2 and those forced to express higher than WT endogenous levels by the OTX2-ER system?

The relevant text has been modified (page 16)

Reviewer #3 (Significance (Required)):

Also see summary. Understanding what restricts cells to germline vs somatic lineages is an important question. By providing functional data showing that OTX2 directly controls chromatin accessibility, the authors add an important layer of understanding to their previous finding that OTX2 plays a key role in preventing mouse germline entry. The use of their previously established OTX2-null cells expressing OTX2-ER to rapidly induce nuclear OTX2 in a mutant background or the most part makes their experiments elegant and convincing. In focusing on the role of one gene in one event in one species, it is specialized and narrow in scope and will mostly be of interest to experts in the field, but there is nothing wrong with that.

References cited in this rebuttal letter

- Bleckwehl T, Crispatzu G, Schaaf K, Respuela P, Bartusel M, Benson L, Clark SJ, Dorigi KM, Barral A, Laugsch M, *et al* (2021) Enhancer-associated H3K4 methylation safeguards in vitro germline competence. *Nat Commun* 12: 5771
- Buecker C, Srinivasan R, Wu Z, Calo E, Acampora D, Faial T, Simeone A, Tan M, Swigut T & Wysocka J (2014) Reorganization of Enhancer Patterns in Transition from Naive to Primed Pluripotency. *Cell Stem Cell* 14: 838–853
- Chen X, Xu H, Yuan P, Fang F, Huss M, Vega VB, Wong E, Orlov YL, Zhang W, Jiang J, *et al* (2008) Integration of External Signaling Pathways with the Core Transcriptional Network in Embryonic Stem Cells. *Cell* 133: 1106–1117
- Di-Gregorio A, Sancho M, Stuckey DW, Crompton LA, Godwin J, Mishina Y & Rodriguez TA (2007) BMP signalling inhibits premature neural differentiation in the mouse embryo. *Development* 134: 3359–3369
- Hayashi K, Ohta H, Kurimoto K, Aramaki S & Saitou M (2011) Reconstitution of the Mouse Germ Cell Specification Pathway in Culture by Pluripotent Stem Cells. *Cell* 146: 519–532
- Hayashi K & Saitou M (2013) Generation of eggs from mouse embryonic stem cells and induced pluripotent stem cells. *Nat Protoc* 8: 1513–1524
- Sankar A, Mohammad F, Sundaramurthy AK, Wang H, Lerdrup M, Tatar T & Helin K (2022) Histone editing elucidates the functional roles of H3K27 methylation and acetylation in mammals. *Nat Genet* 54: 754–760
- Soufi A, Garcia MF, Jaroszewicz A, Osman N, Pellegrini M & Zaret KS (2015) Pioneer Transcription Factors Target Partial DNA Motifs on Nucleosomes to Initiate Reprogramming. *Cell* 161: 555–568
- Thomas HF, Kotova E, Jayaram S, Pilz A, Romeike M, Lackner A, Penz T, Bock C, Leeb M, Halbritter F, *et al* (2021) Temporal dissection of an enhancer cluster reveals distinct temporal and functional contributions of individual elements. *Molecular Cell* 81: 969-982.e13
- Whyte WA, Orlando DA, Hnisz D, Abraham BJ, Lin CY, Kagey MH, Rahl PB, Lee TI & Young RA (2013) Master Transcription Factors and Mediator Establish Super-Enhancers at Key Cell Identity Genes. *Cell* 153: 307–319
- Ying Q-L, Nichols J, Chambers I & Smith A (2003) BMP Induction of Id Proteins Suppresses Differentiation and Sustains Embryonic Stem Cell Self-Renewal in Collaboration with STAT3. *Cell* 115: 281–292
- Zhang J, Zhang M, Acampora D, Vojtek M, Yuan D, Simeone A & Chambers I (2018) OTX2 restricts entry to the mouse germline. *Nature* 562: 595–599

Dear Ian,

Thank you for the submission of your revised manuscript. We have now received the enclosed reports from the referees and I am happy to say that all support its publication now. Please address the last comment by referee 2. A few editorial requests will also need to be addressed before we can proceed with the official acceptance of your manuscript:

- Please add up to 5 keywords to the ms file.
- Please correct the conflict of interest subheading to "Disclosure and Competing Interests Statement"
- The author credits need to be removed from the ms file. All credits need to be entered during online ms submission.
- The Methods section should include a separate Reagents and Tools Table file (listing key reagents, experimental models, software and relevant equipment and including their sources and relevant identifiers) and a Methods and Protocols section in which authors should describe their methods using a step-by-step protocol format with bullet points, to facilitate the adoption of the methodologies across labs. More information on how to adhere to this format as well as downloadable templates (.docx) for the Reagents and Tools Table can be found in our author guidelines: <<https://www.embopress.org/page/journal/14693178/authorguide#manuscriptpreparation>>.
- The source data for Fig 4 does not have folders labeled with the panels (4BCDEF). If possible, please upload the SD for Fig 4 with subfolders for each figure panel.
- Summary should be Abstract
- Materials and Methods should be Methods
- Nomenclature of EV figure legends and individual files is not correct: it should be Figure EV1, etc. instead of Expanded View Figure EV1, etc. Please correct.
- The specific URLs for GSE155062, GSE289297, GSE289297GSE289298 datasets are not provided in the data availability statement, please add the URLs to the Data Availability Section.

Figure Legends - Comments

- Please define the annotated p values ****/***/**/* as well as provide the exact p-values for the same in the legend of figure 4F, EV5 A-C as appropriate and reasonable.
- Please indicate the statistical test used for data analysis in the legends of figures 2C, F; 4F, EV6 A
- Please note that information related to n is missing in the legends of figures 2C, F; EV6 A

EMBO press papers are accompanied online by A) a short (1-2 sentences) summary of the findings and their significance, B) 2-3 bullet points highlighting key results and C) a synopsis image that is exactly 550 pixels wide and 200-600 pixels high (the height is variable). The synopsis image should provide a sketch of the major findings, like a graphical abstract. Please note that text needs to be readable at the final size. Please send us this information along with the final manuscript.

Referee #1:

The authors have fully addressed my comments. The study reveals detailed molecular insights into the function of OTX2 in directing early cell fate decisions. The manuscript is strong and complete and I support publication.

Referee #2:

The authors have adequately addressed all my previous comments and I find the manuscript substantially improved.

One small correction: in the discussion it says "PGCLC differentiation from primed human ESCs via intermediate mesoderm-like cells". I believe the authors mean "incipient mesoderm-like cells", not to be confused with intermediate mesoderm (which gives rise to the kidney etc)

Referee #3:

The authors have adequately addressed my concerns. Their paper now gives a useful resource for the role of OTX2 binding during cell fate decisions.

All editorial and formatting issues were resolved by the authors.

Dr. Elisa Barbieri
University of Edinburgh
Institute for Stem Cell Research
5 Little France Drive
Edinburgh EH164UU
United Kingdom

Dear Dr. Barbieri,

I am very pleased to accept your manuscript for publication in the next available issue of EMBO reports. Thank you for your contribution to our journal.

Yours sincerely,
